# Understanding atmospheric aerosol particles with improved particle identification and quantification by single particle mass spectrometry

Xiaoli Shen[1,2], Harald Saathoff[1,*], Wei Huang[1,2], Claudia Mohr[1,3], Ramakrishna Ramisetty[1,4], Thomas Leisner[1,5]

[1]Institute of Meteorology and Climate Research, Karlsruhe Institute of Technology, Hermann-von-Helmholtz-Platz 1, 76344 Eggenstein-Leopoldshafen, Germany
[2]Institute of Geography and Geoecology, Working Group for Environmental Mineralogy and Environmental System Analysis, Karlsruhe Institute of Technology, Kaiserstr.12, 76131 Karlsruhe, Germany
[3]Department of Environmental Science and Analytical Chemistry, Stockholm University, Stockholm, 11418, Sweden
[4]Now at: TSI Instruments India Private Limited, Bangalore, 560102, India
[5]Institute of Environmental Physics, University Heidelberg, In Neuenheimer Feld 229, 69120 Heidelberg, Germany

*Correspondence to: Harald Saathoff (harald.saathoff@kit.edu)

**Abstract.** Single particle mass spectrometry (SPMS) is a widely used tool to determine chemical composition and mixing state of aerosol particles in the atmosphere. During a six-week field campaign in summer 2016 at a rural site in the upper Rhine valley near Karlsruhe city in southwest Germany, ~$3.7 \times 10^5$ single particles were analysed by a laser ablation aerosol particle time-of-flight mass spectrometer (LAAPTOF). Combining fuzzy classification, marker peaks, typical peak ratios, and laboratory-based reference spectra, seven major particle classes were identified. With the precise particle identification and well characterized laboratory-derived overall detection efficiency (ODE) for this instrument, particle similarity can be transferred into corrected number and mass fractions without a need of a reference instrument in the field. Considering the entire measurement period, "Aged biomass burning and soil dust like particles" dominated the particle number (45.0% number fraction) and mass (31.8% mass fraction); "Sodium salts like particles" were the second lowest in number (3.4%), but the second dominating class in terms of particle mass (30.1%). This difference demonstrates the crucial role of particle number counts correction for mass quantification using SPMS data. Using corrections for size and chemically-resolved ODE, the total mass of the particles measured by LAAPTOF accounts for 23–68% of the total mass measured by an aerosol mass spectrometer (AMS) depending on the measurement periods. These two mass spectrometers show a good correlation (correlation coefficient $\gamma > 0.6$) regarding total mass for more than 85% of the measurement time, indicating non-refractory species measured by AMS may originate from particles consisting of internally mixed non-refractory and refractory components. In addition, specific relationships of LAAPTOF ion intensities and AMS mass concentrations for non-refractory compounds were found for specific measurement periods, especially for the fraction of org/(org+nitrate). Furthermore, our approach allows assigning the non-refractory compounds measured by AMS to different particle classes. Overall AMS-nitrate was mainly arising from sodium salts like particles, while aged biomass burning particles were dominant during events with high organic aerosol particle concentrations.

## 1 Introduction

Life times of ambient aerosol particles range from hours to several days, except for newly formed particles (~3 to 5 nm), which have a lifetime in the order of seconds (Pöschl, 2005). The atmospheric evolution of aerosol particles can alter their internal and external mixing states, as well as their chemical and physical properties on timescales of several hours, e.g., they can acquire coatings of secondary inorganic (e.g. sulfates, nitrates, and ammonium) and secondary organic compounds (Fuzzi et al., 2015). Hence, most aerosol particles are relatively complex mixtures, not easy to distinguish and to trace to their primary source and/or secondary formation pathway. Single particle mass spectrometry (SPMS) has the capability of measuring most components of the particles in real time, thus it has been a widely used technique to investigate mixing state and aging of

aerosol particles for many years (Murphy, 2007; Noble and Prather, 2000; Pratt and Prather, 2012). However, there are still challenging issues related to large amounts of SPMS data analysis.

Particle type identification, i.e., the assignment of every detected particle to one out of a set of particle types, which are either predefined or deduced from the experimental data, is perhaps one of the most critical issues. Different data classification methods, e.g., fuzzy k-means clustering algorithm, fuzzy c-means (modification of k-means), ART-2a neural network, hierarchical clustering algorithms, and machine learning algorithms are applied to reduce the complexity and highlight the core information of mass spectrometric data (Reitz, et al., 2016; Christopoulos et al., 2018). Reitz et al. (2016) reviewed commonly used data classification methods in SPMS studies and pointed out the advantage of the fuzzy c-means clustering approach, which allows individual particle to belong to different particle classes according to spectral similarities. One recent classification approach applied machine learning algorithms and successfully distinguished SOA, mineral and soil dust, as well as biological aerosols based on a known a priori data set (Christopoulos et al., 2018). In this study we used the fuzzy c-means clustering approach which is embedded in the data analysis Igor software for our laser ablation aerosol particle time-of-flight mass spectrometer (LAAPTOF, AeroMegt GmbH). Based on the data classification, averaged or representative mass spectra of different particle classes can be obtained.

Due to the relatively complex laser desorption and ionization (LDI) mechanisms, including charge and proton transfer, as well as ion-molecule reactions that may occur in the plume with many collisions (Murphy, 2007; Reilly et al., 2000; Reinard and Johnston, 2008; Zenobi and Knochenmuss, 1998), some mass spectroscopic signature peaks do not necessarily reflect the primary composition of the particles. Gallavardin et al. (2008) used a pair of peak area ratios, such as $Ca_2O^+/Ca^+$ vs $CaO^+/Ca^+$ and $SiO^-/SiO_2^-$ vs $SiO_3^-/SiO_2^-$, to differentiate calcium/silicon containing mineral dust. Normalized histograms of $PO_3^-/PO_2^-$ and $CN^-/CNO^-$ ratios were used to identify primary biological aerosol particles (Zawadowicz et al., 2017). Setting thresholds for marker peak signals can also help to classify and further identify specific particles (Köllner et al., 2017). Lu et al. (2018) used natural silicon isotopic signatures to study the sources of airborne fine particulate matter ($PM_{2.5}$), which shows how useful isotopic signatures can be for particle identification. A combination of peak area and peak shift ratio, based on subtle changes in ion arrival times in the mass spectrometer, was introduced by Marsden et al. (2018) for the differentiation of mineral phases in silicates. Ternary sub-composition systems, such as $(Al+Si)^+–K^+–Na^+$ and $Cl^-–(CN+CNO)^-–SO_4^-$, were used to identify mineralogy and internal mixing state of ambient particles (Marsden et al., 2019). In our previous study (Shen et al., 2018), laboratory-based reference spectra were suggested to be a useful tool for particle identification. These methods guide the way for improving the techniques to identify particle type and further identify individual aerosol particles.

An even more challenging issue is the quantitative analysis of individual particles' mass and chemical composition, which cannot be directly provided by SPMS measurements, because laser ablation only allows an a priori unknown fraction (neutral species) of the single particle to be vaporized/desorbed and then ionized (Murphy, 2007; Reinard and Johnston, 2008). In addition, matrix effects may obscure the particle composition (Gemayel et al., 2017; Gross et al., 2000; Hatch et al., 2014). Our previous laboratory SPMS study also verified the difficulty of particle quantification due to incomplete ionization, which could not be improved significantly by replacing the originally used nanosecond excimer laser with a femtosecond laser with higher laser power density and shorter laser pulse length (Ramisetty et al., 2018). In the last two decades, great effort has been put into solving such quantification issues by using specific scaling or normalization methods. Allen et al. (2006) developed an explicit scaling method to quantify SPMS data, based on comparison with co-located more quantitative particle measurement. This approach has been widely used to obtain continuous aerosol mass concentrations as a function of particle size (Allen et al., 2006; Bein et al., 2006; Fergenson et al., 2001) and has been improved by a hit rate correction (Qin et al., 2006; Wenzel et al., 2003). Recently, composition-dependent density corrections were applied to such scaling approaches to obtain chemically-resolved mass concentrations (Gunsch et al., 2018; May et al., 2018; Qin et al., 2006; Qin et al., 2012). In these studies, the scaled SPMS data showed good agreement with the results from reference instruments, e.g., micro-orifice uniform deposition impactors (MOUDI), scanning mobility particle sizer (SMPS), aerodynamic particle sizer (APS), and other

independent quantitative aerosol particle measurements, e.g., by a high-resolution time-of-flight aerosol mass spectrometer (HR-ToF-AMS). With respect to particulate chemical compounds, Gross et al. (2000) reported relative sensitivity factors (RSF) for ammonium and alkali metal cations in a single particle mass spectrometer to corresponding bulk concentrations and accurately determined the relative amounts of $Na^+$ and $K^+$ in sea-salt particles. Jeong et al. (2011) developed a method to

quantify ambient particulate species from scaled single particle analysis. Healy et al. (2013) quantitatively determined the mass contribution of different carbonaceous particle classes to total mass and estimated the mass fractions of different chemical species, i.e., sulphate, nitrate, ammonium, OC, EC, and potassium determined for each particle class, by using RSF. The resulting SPMS-derived mass concentrations of these particulate species were comparable with the reference bulk data. Similar methodologies have been used in other SPMS studies (Gemayel et al., 2017; Zhou et al., 2016). It should be noted that these

field-based scaling approaches (field-based overall detection efficiency, ODE) rely on the availability of a reference instrument and their corrections are mainly class independent.

    Many previous studies have also compared single particle classes and bulk species (Dall'Osto et al., 2016; Dall'Osto et al., 2012; Dall'Osto and Harrison, 2012; Dall'Osto et al., 2009; Dall'Osto et al., 2013; Decesari et al., 2014; Decesari et al., 2011; Drewnick et al., 2008; Gunsch et al., 2018; Pratt et al., 2010; Pratt et al., 2011; Pratt and Prather, 2012). Some studies compared

ion intensities from single particle data (Bhave et al., 2002) or specific ion ratios, such as nitrate/sulfate (Middlebrook et al., 2003), OC/EC (Spencer and Prather, 2006), and EC/(EC+OC) (Ferge et al., 2006), carbonaceous/(carbonaceous+sulfate) (Murphy et al., 2006) with the other bulk data. Hatch et al. (2014) used m/z 36 $C_3^+$ as a pseudo-internal standard to normalize the secondary inorganic and organic peak areas in organic rich particles, resulting in good correlation with the independent AMS measurements. Similarly, Ahern et al. (2016) used the peak area ratio of organic matter marker at m/z 28 $CO^+$ to EC

markers ($C_{2-5}^+$) to account for laser shot-to-shot variability, and demonstrated a linear relationship between normalized organic intensity and secondary organic aerosol (SOA) coating thickness on soot particles. A normalized or relative peak areas (RPAs) method was suggested by Hatch et al. (2014) to account for shot-to-shot variability of laser intensities. Although the LDI matrix effects cannot be completely overcome by the aforementioned method, some examples for good comparisons between single particle and bulk measurements were shown.

In this study we aim to quantify mass contributions of different particle classes based on single particle measurements only by employing overall detection efficiencies determined in systematic laboratory studies. As a test case ambient aerosol particles were analysed in summer 2016 at a rural site in the upper Rhine valley of Germany, using and LAAPTOF and HR-ToF-AMS. Seven major particle classes were identified by a fuzzy c-means analysis among a total of $\sim 3.7 \times 10^5$ measured single particles. Based on laboratory determined size dependent overall detection efficiencies (ODEs) of LAAPTOF for different reference

particle types, mass contributions for individual aerosol particles could be estimated. Aerosol particle mass concentrations determined independently by LAAPTOF and AMS are compared and potentially useful relationships of specific ion intensity ratios of LAAPTOF and AMS are discussed.

## 2 Methods

### 2.1 Measurement location and instrumentation

The measurements were made as part of the TRAM01 campaign at a rural site in the upper Rhine valley from July 15[th] to September 1[st], 2016 next to the tram line north of the village of Leopoldshafen, Germany (49°6'10.54" N, 8°24'26.07" E). This location is about 12 km north of the city of Karlsruhe with 300 000 inhabitants and significant industry including a power plant and refineries (Hagemann et al., 2014). Ambient particles were sampled for mass spectroscopic analysis with a flow rate of 1 $m^3$ $h^{-1}$ through a $PM_{2.5}$ inlet (SH 2.5 - 16, Comde-Derenda GmbH) and vertical stainless steel tubes. A total suspended

particulates (TSP) inlet (Comde-Derenda GmbH) was used for instruments for particle physical characterisation. Trace gases were sampled via an 8 mm PFA sampling tube. All sampling inlets were positioned 1.5 m above a measurement container and

3.7 m above ground level. To study the nature and to identify possible sources of the particles in this area, their number, size, chemical composition, and associated trace gases, as well as meteorological conditions were measured using the following instruments: Condensation particle counters (CPC3022A, CPC3772, TSI Inc.), optical particle counter (FIDAS, PALAS GmbH), aethalometer (AE33-7, Magee Scientific), ozone monitor (O341M, Environment SA), $SO_2$ monitor (AF22M Environment SA), $NO_2$ monitor (AS32M, Environment SA), $CO_2$ monitor (NGA2000, Rosemont Inc.), and meteorology sensors (WS700 & WS300, Lufft GmbH). From July 26[th] to August 31[st], the following mass spectrometers were in operation, e.g., a high resolution time-of-flight aerosol mass spectrometer (HR-ToF-AMS, Aerodyne Inc.), and a laser ablation aerosol particle time-of-flight mass spectrometer (LAAPTOF, AeroMegt GmbH), providing real time information on size and mass spectral patterns for bulk samples and individual particles, respectively.

The HR-ToF-AMS yields quantitative information (mass concentration) on size resolved particle bulk chemical composition with high time resolution and high sensitivity (DeCarlo et al., 2006). Briefly, aerosols are sampled with a flowrate of ~84 cm$^3$ min$^{-1}$ via an aerodynamic lens, which focuses particles with sizes of 70 to 2500 nm (vacuum aerodynamic diameter, $d_{va}$) into a narrow beam. The particle beam passes through a sizing chamber where the particles' size is determined. Afterwards, particles encounter a 600 °C heater that vaporises the non-refractory species. The vapours are ionized by electron impact (electron energy: 70 eV). The generated positive ions are analysed by a time-of-flight mass spectrometer. Particles can bounce off the heater/vaporizer, leading to an underestimation of ambient mass concentrations measured by AMS. Collection efficiencies (CE) are used to correct for this (CE, the product of net particle transmission and detection efficiency) (Canagaratna et al., 2007). It is important to note that the CE can vary depending on composition and phase of the particles (Bahreini et al., 2005). In this study, we applied a CE value of 0.5. This is in agreement with previous studies (Canagaratna et al., 2007; Middlebrook et al., 2012), and close to a composition dependent CE calculated for this measurement campaign by Huang et al. (2019).

The LAAPTOF is a commercially available SPMS and has been described elsewhere (Ahern et al., 2016; Gemayel et al., 2016; Marsden et al., 2016; Ramisetty et al., 2018; Reitz et al., 2016; Shen et al., 2018; Wonaschuetz et al., 2017). In brief, aerosols are sampled with a flowrate of ~80 cm$^3$ min$^{-1}$ via an aerodynamic lens, focusing and accelerating particles in a size range between 70 nm and 2500 nm $d_{va}$. Afterwards, they pass through the detection chamber with two diode laser beams ($\lambda$ = 405 nm). Particles smaller than 200 nm and larger than 2 μm are difficult to detect, due to weak light scattering by the smaller particles and due to a larger particle beam divergence for the larger particles. Once a single particle is detected successively by both of the detection lasers, its aerodynamic size is determined and recorded based on its time of flight, and an excimer laser pulse ($\lambda$ = 193 nm) is fired for a one step desorption/ionization of the refractory and non-refractory species of the particle. The resulting cations and anions are analysed by a bipolar time-of-flight mass spectrometer resulting it mass spectra with unit mass resolution. Thus, for each individual particle its size and a pair of positive and negative mass spectra are measured.

**2.2 Single particle identification and quantification methods for LAAPTOF data**

The general data analysis procedures for particle spectral and size information were described in full detail in our previous study (Shen et al., 2018). In brief, spectral data is classified by a fuzzy *c*-means clustering algorithm embedded in the LAAPTOF Data Analysis Igor software (Version 1.0.2, AeroMegt GmbH) to find the major particle classes. Afterwards, we can obtain particle class resolved size ($d_{va}$) distribution and the representative spectra, which will be correlated with laboratory-based reference spectra. The resulting correlations together with marker peaks (characteristic peaks arising from the corresponding species and some typical peak ratios (e.g., isotopic ratio of potassium) are used to identify the particle classes. Here, we extend this approach to quantify particle class mass contributions using a large ambient sample as test case.

The fuzzy *c*-means clustering approach has the advantage of allowing particles to belong to multiple classes based on the similarity of the mass spectra (Reitz et al., 2016), namely attributing one spectrum (particle) to multiple clusters (particle classes). The similarity metric is Euclidian distance between the spectral data vectors and a cluster centre (Hinz et al., 1999;

Reitz et al., 2016). In our study, fuzzy clustering derived fraction for each particle class is the degree of similarity between aerosol particles in one particular class, rather than a number percentage. Thus, we can obtain similarity information for the whole data set rather than a single particle (Hinz et al., 1999; Reitz et al., 2016). One drawback is that the individual particles are not directly assigned to individual particle classes, which hinders a direct class-dependent quantification of particle mass.

In order to quantify particle mass, we first need to assign a particle class to every individual particle, which is achieved by correlating the individual bipolar mass spectra with the representative fuzzy class spectra using Pearson's correlation coefficient ($\gamma$). Since the positive LAAPTOF spectra are more characteristic than the negative ones (Shen et al., 2018), the threshold value for the positive spectra correlation was set to $\gamma_{pos} \geq 0.6$, while for the negative spectra $\gamma_{neg}$ was tuned with values ranging from 0.3 to 0.8 (cf. Table S1). Individual particles are assigned to the class for which the corresponding

correlation coefficients for both spectra exceed the threshold values. All corresponding correlation coefficients ($\gamma_{pos}$ and $\gamma_{neg}$) are listed in Table S1. This way, we can obtain time series of particle counts, which have good ($\gamma > 0.6$)/strong correlation ($\gamma > 0.8$) with the fuzzy results. The corresponding correlation coefficients are also listed in Table S1 and typical examples are shown in Fig. S1. With this method, we were able to successfully classify 96% of the measured particles. Once the class information for individual particles has been determined, we are able to calculate single particle geometric size, volume and

mass as described in the following.

    For simplicity, we assume the particles are spherical with a shape factor ($\chi$) of 1, thus particle geometric diameter ($d_p$), volume ($V_p$) and mass ($m_p$) can be obtained from the following equations:

$$d_p = d_m = \frac{d_{va}}{\rho_{eff}} \times \rho_0 \ \left( \chi = 1; \rho_p = \rho_{eff} \right) \text{ (DeCarlo et al., 2004)} \tag{1}$$

$$V_p = \frac{1}{6} \times \pi \times d_p^{\ 3} \tag{2}$$

$$m_p = V_p \times \rho_{eff} \tag{3}$$

where $d_m$ is the electrical mobility diameter, $d_{va}$ is the vacuum aerodynamic diameter measured by LAAPTOF, $\rho_0$ is the standard density (1g cm$^{-3}$), $\rho_p$ is the particle density, and $\rho_{eff}$ is the effective density. It should be noted that in some previous studies, the particle shapes were also assumed as spherical and uniform particle densities ranging from ~1.2 to 1.9 g cm$^{-3}$ were applied for total aerosol particle mass quantification (Allen et al., 2006; Allen et al., 2000; Ault et al., 2009; Gemayel et al.,

2017; Healy et al., 2013; Healy et al., 2012; Jeong et al., 2011; Wenzel et al., 2003; Zhou et al., 2016). In our study, we have determined an average density of $1.5 \pm 0.3$ g cm$^{-3}$ for all ambient particles, based on a comparison between $d_{va}$ measured by AMS and $d_m$ measured by SMPS. However, the density for different types of ambient particles varies, especially for fresh ones (Qin et al., 2006). Particle densities varied during the campaign (Fig. S2) and the representative mass spectra of different particle classes indicate chemical inhomogeneity. In order to reduce the uncertainty induced by the assumption of a uniform

density, we assigned specific effective densities (derived from $d_{va}/d_m$) from literature data to each particle class. A density of 2.2 g cm$^{-3}$ was used for calcium nitrate rich particles (Zelenyuk et al., 2005), 1.25 g cm$^{-3}$ for aged soot rich in ECOC-sulfate (Moffet et al., 2008b; Spencer et al., 2007) , 2.1 g cm$^{-3}$ for sodium salts (Moffet et al., 2008b; Zelenyuk et al., 2005), 1.7 g cm$^{-3}$ for secondary inorganic rich particles (Zelenyuk et al., 2005; Zelenyuk et al., 2008), 2.0 g cm$^{-3}$ for aged biomass burning particles (Moffet et al., 2008b), 2.6 g cm$^{-3}$ for dust like particles (Bergametti and Forêt, 2014; Hill et al., 2016). These densities

were used for the individual particles of each class without size dependence. Similar chemically-resolved densities have also been used in some previous studies (Gunsch et al., 2018; May et al., 2018; Qin et al., 2006; Qin et al., 2012).

    Furthermore, the single particle identification allows for correcting the particle number counts by using the overall detection efficiency (ODE), which depends strongly on particle size and type (Allen et al., 2000; Dall'Osto et al., 2006; Qin et al., 2006; Shen et al., 2018). In a previous publication, we defined ODE as the number of bipolar mass spectra obtained from

the total number of particles in the sampled air, described how to generate the laboratory-derived ODE and discussed the factors influencing ODE in detail (Shen et al., 2018). Our ODE accounts for both physical and chemical factors (e.g., particle size and types shown in Fig. 1). However, we did not determine relative sensitivity factors for individual chemical compounds.

As shown in Fig. 1, we have determined ODEs for several particle types, from particles consisting of pure compounds to the more realistic ones including major ambient particles (cf. Fig. 1). For simplicity and in order to account for different types of ambient particles, we averaged the ODE determined for ammonium nitrate, sodium chloride, PSL particles, and some other particles, e.g., agricultural soil dust, sea salt, organic acids, as well as secondary organic aerosol particles measured in the lab. The mean ODE with uncertainties as a function of particle size ($d_m$) are shown in Fig. 1. However, using a mean ODE will obviously lead to some bias. For example, if we apply ODE mean values to all the ambient particles, the number of ammonium nitrate rich particles will be overestimated due to the higher ODE of ammonium nitrate, while the ammonium sulfate rich, sea salt particles, and some organic rich particles will be underestimated. Therefore, we used reference particle ODE values to estimate the size dependent ODE values for the particle classes observed in the field as follows. ODE values for ammonium nitrate and sodium chloride were used to fit ODE curves for secondary inorganic rich and sodium salt like particles, respectively. The mean ODE values from all reference particles was used for the class of aged soot particles since it showed best agreement with the reference soot particles (cf. Fig. 1). For the same reason, the minimum ODE curve from all reference particles was used for all dust like particle classes It should be noted that, dust like particles were often mixed with other species such as organics (e.g. biomass burning-soil particles; cf. section 3.1), and that they likely have dust-core shell structures (Goschnick et al., 1994). We assume that their detection is dominated by the dust core as it significantly influences the light scattering (size) and the particle beam divergence (shape).

The chemically-resolved ODE could also bring some bias due to complex particle matrix. For instance, if ammonium sulfate is internally mixed with ammonium nitrate, LAAPTOF can detect both of them with good efficiency. This has been verified in our laboratory and the matrix effect has been discussed in our previous study (Shen et al., 2018). As shown in Fig. 1, ODEs for ammonium nitrate are at a higher level, while ODEs for sodium chloride are relatively low. This could lead to an underestimation and overestimation of secondary inorganic rich and sodium salts particles, respectively. ODEs from reference particles with low detection efficiency were applied to dust like particles. This may lead to an overestimation of their concentration if they are mixed with better detectable species. Mean ODEs values were applied to soot particles which may lead to an overestimation if they were e.g. coated. This is because even non-absorbing species e.g., organics can refract light towards the absorbing black carbon core, increasing light absorption (Ackerman and Toon, 1981). Since most of the particle classes consist of mixtures of the poorly detectable types with better detectable types this seems to partially compensate for the limitation of LAAPTOF to detected certain particle types as evident by comparison with the AMS mass concentrations (cf. section 3.2).

As shown Fig. 1, we determined ODE values for mobility equivalent particle sizes ($d_m$) ranging from 300 nm to 1 μm. The ODE decreases significantly for larger particles, because of increasing particle beam divergence. We assume ODEs for supermicron particles to follow the decreasing trend illustrated in Fig. 1. Please note that LAAPTOF cannot measure particles larger than a $d_{va}$ of 2.5 μm, which corresponds to a $d_m$ of 1.0 to 1.5 µm assuming effective particle densities of 1.7 to 2.6 g cm$^{-3}$ for different ambient particle classes, respectively. Hence, a large fraction of the ambient particles measured by LAAPTOF could be number-corrected by using our laboratory-derived ODEs.

The equations for correction and calculation of mass concentration are as follows:

$$counts_{corrected} = 1/ODE_{size\ and\ chemically-resolved} \qquad (4)$$

$$mass_{corrected} = counts_{corrected} \times m_p \qquad (5)$$

$$mass\ concentration = Total\ mass/(sample\ flowrate \times time) \qquad (6)$$

where $ODE_{d_m}$ is the mean ODE that depends on $d_m$; $counts_{corrected}$ and $mass_{corrected}$ are the corrected particle number counts and mass at each time point; the sample flowrate is ~80 cm$^{-3}$ min$^{-1}$. Using Eq. (4) to (6) we can calculate the corrected number and mass fractions.

The aforementioned assumptions and the related uncertainties in particle mass are summarised as follows: 1) ambient particles are spherical with a shape factor $\chi$=1. However, several ambient particle types are non-spherical with a shape factor

$\chi$ not equal to 1, e.g., $\chi_{NaCl} = 1.02-1.26$ (Wang et al., 2010) and $\chi_{NH_4NO_3} = 0.8$ (Williams et al., 2013). This can cause uncertainties of 26% and 20% for the particle diameter and 100% and 50% for the particle mass of sodium chloride like and ammonium nitrate like particles, respectively. For soot like particles, the shape caused uncertainty could be even larger, due to their aggregate structures. Such an uncertainty is difficult to reduce, since we do not have particle shape information for individual particles. However, using effective densities may at least partially compensate some of the particle shape related uncertainties. 2) Particles in the same class have the same density, which is likely to vary and lead to an uncertainty hard to estimate. 3) The variability of the ODE values (cf. Fig. 1) depends on particle size and type. It reaches values ranging from $\pm100\%$ for 200 nm particles to $\pm170\%$ for 800 nm size particles.

Hence, the overall uncertainty in particle mass according to the assumptions is ~300% with the ODE caused uncertainty being dominant. This is because: 1) the aforementioned particle matrix effects may cause higher or lower ODEs than their surrogates generated in the laboratory. In addition, the more complex morphology and various optical properties of ambient particles can have a strong impact on their ODE (Shen et al., 2018); 2) instrumental aspects such as alignment and variance in particle-laser interaction lead to uncertainty in ODE. They are included in the uncertainties given in Fig. 1 for which repeated measurements after various alignments were used. The fluctuations of particle-laser interactions can be reduced by using a homogeneous laser desorption and ionization beam (Wenzel and Prather, 2004) or delayed ion extraction (Li et al., 2018; Vera et al., 2005; Wiley and Mclaren, 1955). Note that we used the same sizing laser and desorption/ionization laser pulse energy (4 mJ) in the field as those used for generating ODE, and aligned the instrument in the field with the similar procedures as we did in the lab. During our field measurements we did calibrations of the LAAPTOF with PSL particles of 400, 500, 700, and 800 nm $d_m$ resulting in ODE values with no significant difference compared to the ODE values determined in the laboratory. This finding reflects the good stability of the LAAPTOF performance in the temperature controlled container. Actually, once the LAAPTOF adjustments were optimized after transport no further adjustments were necessary during the 6 weeks of the campaign. Moreover, it is important to note that the ODE curve applied herein should not be extrapolated to other LAAPTOF or SPMS instruments without a standard check against e.g. PSL particles. In order to evaluate our quantification approach, we will compare the particle mass estimated based on single particle measurements with AMS total mass in section 3.2.

Noteworthy that the major difference between our quantification method and previous SPMS studies is that our ODE is based on elaborate laboratory work, while previous studies typically used field-based scaling approaches (field-derived ODE).

# 3 Results and Discussion

## 3.1 Identification of particle classes and the internal mixing

During the six-week measurement campaign, we obtained ~$3.7 \times 10^5$ bipolar LAAPTOF spectra for single particles. Seven major particle classes were found using fuzzy c-means classification. The corresponding representative spectra with marker peaks assignment are shown in Fig. 2. Considering some weak but characteristic peaks, we show the spectra with a logarithmic scale. The linearly scaled spectra (cf. Fig. S3) are provided for comparison in the supporting information. Furthermore, Fig. 3 shows the size resolved number fraction for the seven particle classes measured during the field campaign TRAM01, based on fuzzy classification according to fuzzy c-means clustering algorithm as well as the overall size distribution for all particles measured by LAAPTOF during the campaign. Signatures for organic and secondary inorganic compounds can be observed in each class, i.e., for organics m/z 24 $C_2^-$, 25 $C_2H^-$, 26 $C_2H_2/CN^-$, and 42 $C_2H_2O/CNO^-$, for sulfate 32 $S^-$, 64 $SO_2^-$, 80 $SO_3^-$, 81 $HSO_3^-$, 97 $HSO_4^-$, 177 $SO_3HSO_4^-$ and 195 $HSO_4H_2SO_4^-$, for nitrate 30 $NO^+$, 46 $NO_2^-$, and 62 $NO_3^-$, and for ammonium 18 $NH_4^+$ and 30 $NO^+$. Similar species were previously identified off-line in the same region (Faude and Goschnick, 1997; Goschnick et al., 1994). Note that 30 $NO^+$ can not only originate from nitrate (majority), but also from ammonium (Murphy et al., 2006; Shen et al., 2018). Besides, m/z 24 $C_2^-$ could also be related to elemental carbon (EC). In this case, m/z 24$^-$ should actually show a higher intensity than m/z 26$^-$, and further EC markers ($C_n^{\pm}$) should show up as well. Although different particle classes

have similar fragments, they show characteristic patterns with several intensive marker peaks in the corresponding spectra, which can also be identified using reference spectra (Shen et al., 2018).

After fuzzy classification each particle was tested for its similarity to the different particle classes. Although a similarity is not equal to the number fraction, they are related. A higher similarity of the total aerosol particles to one class indicates that a bigger number fraction of this class may be expected once the individual particles are assigned to it. As shown in Fig. 4a, the highest similarity (43.5% of all particles) is found to class 3, which is named "Sodium salts" due to its strong correlation ($\gamma \geq$ 0.8) with Na salts (cf. Fig. 5). The spectra of this class in our study feature marker peaks arising from $NaNO_3$ (m/z 115 $Na(NO_2)_2^-$, 131 $NaNO_2NO_3^-$, and 147 $Na(NO_3)_2^-$), $Na_2SO_4$ (m/z 165 $Na_3SO_4^+$), and NaCl (m/z 81/83 $Na_2Cl^+$, 139/141 $Na_3Cl_2^+$, 35/37 $Cl^-$, and 93/95 $NaCl_2^-$) (cf. Fig. 2). These signature peaks were also observed for Na related particle types such as aged sea salt, Na-containing dust, and Na/K-sulfate rich particles in the other SPMS studies (Gard et al., 1998; Gaston et al., 2011; Jeong et al., 2011; May et al., 2018; Middlebrook et al., 2003; Schmidt et al., 2017). In the positive spectra of class 3, there is a nitrogen containing organic compound marker at m/z 129 $C_5H_7NO^+$, which could originate from the OH oxidation of volatile organic compounds (VOCs) in the presence of $NO_x$ on the seed particles, since the same peak was observed during simulation chamber studies with OH radicals reacting with α-pinene and/or toluene in the presence of $NO_x$. Besides, peaks at m/z 149 $C_4H_7O_2NO_3^+$ and 181 $C_4H_7O_4NO_3^+$ are associated with organonitrates that can form from the oxidation of VOCs in the presence of $NO_x$ (Perring et al., 2013) and are expected to increase the light absorbing capability of the particles (Canagaratna et al., 2007). Huang et al. (2019) showed that organonitrates contributed to particle growth during night time at this location. This class accounts for the largest fraction in the size range from 1000 to 2500 nm $d_{va}$ (cf. Fig. 3). The size distribution of class 3 particles was dominated by two modes centred at about 1400 and 2000 nm $d_{va}$, indicating two sub-particle populations in this class. Goschnick et al. (1994) did off-line depth-resolved analysis of the aerosol particles collected north of Karlsruhe in the upper Rhine valley, and observed sodium chloride in both fine and coarse particles, while sodium nitrate was mainly enriched in the coarse mode. This hints to possible sub-classes assignments, which are likely to be fresh and aged sea salts. However, the measurement site is relatively far away from the sea (e.g., North Atlantic Ocean is ~800 km away). Therefore, we need more evidence, such as back trajectory analysis or other transport modelling, to prove that this class is really fresh and/or aged sea salt. This will be discussed in a separate study.

20.8% of the total particle population belongs to class 4 ("Secondary inorganics-Amine"). This class has the most prominent secondary inorganic signature and strongest correlation with the reference spectra for homogeneous mixtures of $NH_4NO_3$ and $(NH_4)_2SO_4$. In addition, it features marker peaks for amines at m/z 58 $C_2H_5NHCH_2^+$, 59 $(CH_3)_3N^+$, 86 $(C_2H_5)_2NCH_2^+$, 88 $(C_2H_5)_2NO/C_3H_6NO_2^+$, 118 $(C_2H_5)_2NCH_2^+$, which were also identified by SPMS in the other field and lab studies (Angelino et al., 2001; Dall'Osto et al., 2016; Healy et al., 2013; Jeong et al., 2011; Köllner et al., 2017; Lin et al., 2017; Pratt et al., 2009; Roth et al., 2016; Schmidt et al., 2017). Among all the representative mass spectra for the seven particle classes, class 4 is relatively "clean" with the fewest peaks (cf. Fig. 2 and Fig. S3), indicating that these particles did not have had the time to uptake other components. Hence, most likely they were formed not very long ago by conversion of their precursors. The secondary inorganic amine particles have a rather narrow size distribution in the range between 500 and 1000 nm $d_{va}$ (cf. Fig. 3).

Aged biomass burning and soil dust like particles (class 5: Biomass burning – Soil) comprise 16.1% of all particles according to similarity of the mass spectra. It has the most prominent peak at m/z 39 $K/C_3H_3^+$, aromatic marker peaks at 50 $C_4H_2^+$, 63 $C_5H_3^+$, 77 $C_6H_5^+$, 85 $C_7H^+$, 91 $C_7H_7^+$, 95 $C_7H_{11}^+$, 104 $C_8H_8^+$, 115 $C_9H_7^+$. The ratio of m/z $39^+/41^+$ is ~11.6, which is similar to the value of (13.5 ± 0.9) measured for pure potassium containing inorganic particles (e.g. $K_2SO_4$) by our LAAPTOF in the laboratory. The contribution of organic fragments is likely the reason for the slightly lower value, as this ratio was determined to ~8 for humic acid and ~1.1 for α-pinene SOA (Shen et al., 2018). Hence, we assign the signal at m/z $39^+$ mainly to potassium. The aromatic signature was observed by the other SPMS (Dall'Osto and Harrison, 2012; Schmidt et al., 2017; Silva and Prather, 2000). As suggested by previous studies, such potassium rich particles can originate from biomass burning

and are often mixed with sulfate and/ nitrate (Gaston et al., 2013; Lin et al., 2017; Middlebrook et al., 2003; Moffet et al., 2008a; Pratt et al., 2010; Qin et al., 2012; Roth et al., 2016; Schmidt et al., 2017). This is also the case for class 5 particles that exhibit a characteristic peak at m/z 213 $K_3SO_4^+$. Note that we also attributed this class as soil dust like based on the correlation diagram (Fig. 5), although there are no obvious marker ions visible. It is correlated well ($\gamma \geq 0.6$) with reference spectra of dust particles, especially agricultural soil dust. The weak spectral signal might due to a core-shell structure of the particles (Pratt and Prather, 2009). In fact, previous studies identified soil dust as the particle type dominating the coarse particles sampled in the same region (Faude and Goschnick, 1997; Goschnick et al., 1994). Goschnick et al. (1994) found a core-shell structure in both submicron and coarse particles collected north of the Karlsruhe city in the upper Rhine valley. This supports our hypothesis. In addition, similar as class 3, class 5 also has two modes in its size distribution centred at about 500 and 800 nm $d_{va}$. Such potential sub-classes will be further analysed in the future.

Particle class 6 contains 5.7% of all particles and they have sizes ranging from 400 to 1000 nm $d_{va}$. This class is named "Biomass burning-Organosulfate" short for "Aged biomass burning and Organosulfate containing particles". It also shows biomass burning markers such as m/z 213 $K_3SO_4^+$, and features organosulfates at m/z 141 $C_2H_5OSO_4^-$, 155 $C_2H_3O_2SO_4^-$, and 215 $C_5H_{11}OSO_4^-$, which are consistent with signals from sulfate esters of glycolaldehyde/methylglyoxal, glyoxal/glycolic acid, and isoprene epoxydiols (IEPOX), respectively, observed by other SPMS in field measurements (Froyd et al., 2010; Hatch et al., 2011a, b). Unfortunately, we don't have laboratory based reference spectra for organosulfate particles. Such reference could be very useful for a further analysis. The ratio of m/z $39^+/41^+$ is ~6.7 is closer to organics rather than to potassium. However, we cannot rule out a significant potassium contribution. In addition, this class features a specific pattern of m/z $39^+$, $41^+$, and $43^+$ (which have much higher intensities than their interstitial peaks at m/z $40^+$ and $42^+$), hydrocarbon and oxygenated organic fragments at m/z $53^+$, $55^+$, $63^+$, $65^+$, $67^+$, $69^+$, $71^+$, $73^+$, $81^+$, $83^+$, $85^+$, $95^+$, $97^+$, and $99^+$ likely from organic acids and biogenic SOA (Shen et al., 2018).

Class 1 (5.0% of all particles) is identified as "Calcium-Soil" short for "Calcium rich and soil dust like particles". It contains calcium related signatures at m/z 40 $Ca^+$, 56 $CaO^+/Fe^+$, 57 $CaOH^+$, 75 $CaCl^+$, 96 $Ca_2O^+$, and 112 $(CaO)_2^+$, as well as some other metals related signatures including m/z 23 $Na^+$, 64/66 $Zn^+$, 65 $Cu^+$, 138 $Ba^+$, 154 $BaO^+$ and 206–208 $Pb^+$. Most of the signature peaks for calcium related particles, such as Ca rich soil dust, engine exhaust, and lake spray aerosols were also identified by other SPMS studies (Dall'Osto et al., 2016; May et al., 2018; Roth et al., 2016). This class shows a strong correlation with nitrate and correlates well with all reference spectra of dust samples, especially soil dust (cf. Fig. 5). Class 2 (4.3% of all particles), "Aged soot", is predominantly located in the small size range (200 to 600 nm $d_{va}$) and exhibits prominent EC patterns in mass spectra (characteristic $C_n^\pm$ progressions with up to n = 12) and mixed with sulfate and nitrate. Such soot signatures are normally found in SMPS studies (Ault et al., 2010; Dall'Osto et al., 2016; Gaston et al., 2013; Middlebrook et al., 2003; Spencer and Prather, 2006). These mass spectra show a strong correlation to the reference spectra of soot particles, especially diesel soot ($\gamma = $ ~1). Class 7 (4.6% of all particles) is identified as "Mixed/aged-Dust", which contains no obvious characteristic features, and is correlated with most of the reference spectra. It has a relatively even and broad size distribution covering the whole size range that LAAPTOF is able to measure.

We observe intensive signals at m/z 138 $Ba^+$ and 154 $BaO^+$ in class 1, 5, 6 and 7, indicating a similar source of these particle types, which all have a good correlation with mineral and soil dust particles (Fig. 5). Prominent lead markers at m/z $206^+$ to $208^+$ can be found in each class, except class 4, which is further evidence for these particles to be relatively young. The marker peaks for lead appear broader because at higher m/z, we observe larger peak shifts that cannot be completely corrected with the existing LAAPTOF software. Note that even though we did not obtain spectra for pure ammonium sulfate or pure biogenic SOA particles in ambient air, it is still possible for such particles to be present. However, laboratory measurements show a very low sensitivity of the LAAPTOF to these types of particles, potentially due to their low absorbance at 193 nm. Due to this low instrument sensitivity for these types of particles is very difficult to achieve reasonable quantitative estimates about their abundance based on LAAPTOF measurements alone.

The aforementioned full and short names for seven classes, as well as their signature ion peaks are listed in Table 1. We emphasize here that the expression "rich" as used in this study only indicates a strong signal in the mass-spectra rather than a large fraction in mass, since there is no well-defined relationship between LAAPTOF spectral signal and the corresponding quantity. The sensitivities of this instrument to different species have to be established in the future.

All the laboratory-based reference spectra used in this study are publicly available via the EUROCHAMP-2020 data base (www.eurochamp.org). Information on newly added reference spectra is given in Table S2.

## 3.2 Quantification of single particle mass and the external mixing

In this section, we estimate mass concentrations of the particle classes observed in the field. This is based on the particle identification discussed above as well as the assignment of appropriate ODE values of surrogate reference particles and on

several assumptions on particle density and shape (cf. Sect. 2.2). Please note that both AMS and LAAPTOF cannot measure particles larger than 2.5 μm, which can be analysed by FIDAS. FIDAS data showed that PM$_{2.5}$ accounted for majority mass of the total aerosol particles sampled through TSP inlet (PM$_{2.5}$ = 73% of PM$_{10}$ and 64% of PM$_{total}$, respectively). In this study, we only focus on PM$_{2.5}$ particles. The fuzzy classification derived similarity (Fig. 4a) can be transferred into corrected number fractions using size and chemically-resolved ODE (Fig. 4b) and further transferred into mass fractions (Fig. 4c) of the seven

particle classes. The corresponding time series of chemically-resolved number and mass concentrations can be found in Fig S4. Please note that the aged soot particles (class 2), which dominate the number fraction for particles below 400 nm in the fuzzy c-means analysis comprise only a minor fraction of the total number counts in Figure 4 because the total particle number is dominated by particles larger than 500 nm (cf. Figure 3b). Significant changes can be observed between the similarity number fraction, the corrected number fraction, and the resulting mass fractions (cf. Fig. 4a to b to c). Compared to the

similarity fraction, the number fractions of class 3 "Sodium salts" and class 4 "Secondary inorganics-Amine" decrease dramatically: "Sodium salts" particles changed from 43.5% (similarity) to 3.4% (corrected number fraction) and "Secondary inorganics-Amine" dramatically decreased from 20.8% to 2.4%, while those of the other classes increase. This is because classes 3 and 4 comprise mainly of larger particles (class 3: d$_{va}$ peaks at ~1400 and 2000 nm corresponding to d$_p$ ~700 and 1000 nm; class 4 peaks at ~680 nm d$_{va}$ and 400 nm d$_p$) which have the highest ODE values. In contrast the other classes

comprise mainly smaller particles (d$_{va}$ < 500 nm; d$_p$ < 300 nm) (cf. Fig. 3), which have a lower ODE (cf. Fig. 1). Class 5 "Biomass burning-Soil" accounts for the second highest number fraction of the smaller particles and has a relatively high effective density. After correction, the number fraction of particles attributed to this class has increased from 16.1% to 45.0%, corresponding to 31.8% mass fraction, and it becomes the dominating class with respect to particle number and mass. "Sodium salts" is another dominating class with respect to mass (30.1% mass fraction) due to their relatively large size. These

observations demonstrate the crucial role of the corrections applied for particle mass quantification for SPMS data. Note that we can obtain similarly corrected number and mass fractions by using minimum, mean, and maximum ODE, respectively (Table S3). The observed external mixing of aerosol particles varied significantly with time, e.g., class 6 "Biomass burning-Organosulfate" dominated both particle number and mass at the beginning of the measurements until August 1$^{st}$, while class 3 dominated the mass for August 5$^{th}$ to 10$^{th}$, 21$^{st}$ to 24$^{th}$, and 29$^{th}$ to 30$^{th}$, and class 4 particles peaked twice on August 11$^{th}$ and

19$^{th}$ (cf. Fig. 4).

As discussed above, raw LAAPTOF data overestimate the particles with higher ODE, while the ones with lower ODE will be underestimated. After correction of the number counts and estimation of the mass concentrations, we can compare the LAAPTOF result with the quantitative instruments such as AMS in the overlapping size range of 200 to 2500 nm d$_{va}$. A correction for the particles in the size range between 70–200 nm considering mass concentrations may be negligible since they

typically contribute only a minor mass fraction. It turns out that the total mass of the particles measured by LAAPTOF is 7±3% (with maximum ODE), 16±6% (mean ODE), 60±24% (minimum ODE) and 45±16% (23–68% with chemically-resolved ODE) of the total AMS mass depending on the measurement periods. Two criteria were used to select characteristic time periods: a

period should have a stable correlation between LAAPTOF and AMS total mass; and a period should contain special events or dominating particle classes observed by LAAPTOF and/or AMS (cf. Fig 4 c and Fig. 6). Despite of the relative large differences in the average mass concentrations of LAAPTOF and AMS they show much better agreement in total mass and also good correlations during specific periods (P), such as P1, 2, 4, and 5 (cf. Fig. 6 and Fig. S5), covering ~85% of the measurement time. Hence, the large differences in the average mass concentrations are caused by larger deviations during some relatively short periods or events. Considering that AMS can only measure non-refractory compounds, the good correlation between AMS and LAAPTOF gives us a hint that the species measured by AMS may mainly originate from the particles of complex mixtures of both refractory and non-refractory species. It is worth noting that weakest correlation ($\gamma$=-0.1) is observed in P6 when LAAPTOF measured the highest fraction of sodium salts particles (especially sodium chloride) on August 29th, while AMS is unable to measure refractory species such as sodium chloride. Specifically, from 9:00 to 23:53 on August 29th, LAAPTOF and AMS tended to be slightly anti-correlated ($\gamma$=-0.3).

As shown in Fig. 6 (a), the mass ratio of LAAPTOF to AMS has its lower values in P3 and P5 when the AMS organic mass concentration is higher than in most of the other periods. Although LAAPTOF data shows a good correlation with the AMS data e.g. for period P5, it obviously misses a large mass fraction of most likely smaller organic particles. The corresponding chemically-resolved size distributions of particles measured by AMS are given in Fig. S6. This may be due to an insufficient representation of this kind of organic rich particles in the particles classes identified initially. Even using reference spectra of organic particles it was not possible to identify a number of those particles sufficient to close this gap. In addition, during the whole campaign the sulfate mass fraction measured by AMS is largest in P3 (cf. Fig. 6c). However, the LAAPTOF is not sensitive to some sulfate salts, e.g., pure ammonium sulfate (Shen et al., 2018), thus it is likely that such particles were dominating in P3, which resulted in a weaker correlation between these two instruments. Relatively pure ammonium sulfate was also suggested to be a "missing" particle type in the other SPMS field studies (Erisman et al., 2001; Stolzenburg and Hering, 2000; Wenzel et al., 2003) and (Thomson et al., 1997) showed in a laboratory study that pure ammonium sulfate particles were difficult to measure using LDI at various wavelengths.

**3.3 Correlation of AMS and LAAPTOF results for non-refractory compounds**

Considering the different capabilities of LAAPTOF and AMS, we did not apply the relative sensitivity factors (RSF) method (Healy et al., 2013; Jeong et al., 2011). We analysed our LAAPTOF and AMS data independently and compared them thereafter. For LAAPTOF data, we used relative ion intensities (each ion peak intensity is normalised to the sum of all or selected ion signals. Positive and negative ions were analysed separately), similar to the relative peak area (RPA) method suggested by Hatch et al. (2014). As shown in Fig. S7 (a), m/z 30 $NO^+$ measured by LAAPTOF has a good correlation ($\gamma$ = 0.6) with ammonium measured by AMS, but LAAPTOF m/z 18 $NH_4^+$ doesn't show this ($\gamma$ = 0.3, not shown in the figure). This was also found by Murphy et al. (2006) for another single particle mass spectrometer, PALMS, which also uses an excimer laser with the same wavelength for ionization as that in the LAAPTOF. For nitrate (panel b: sum of the marker peaks at m/z 46 $NO_2^-$ and 62 $NO_3^-$), sulfate (panel c: sum of m/z 32 $S^-$, 64 $SO^-$, 80 $SO_3^-$, 81$HSO_3^-$, 96 $SO_4^-$, 97 $HSO_4^-$, 177 $SO_3HSO_4^-$, 195 $H_2SO_4HSO_4^-$), and organics (cations in panel d: sum of m/z 43 $C_3H_7/C_2H_3O/CHNO^+$, 58 $C_2H_5NHCH_2^+$, 59 $(CH_3)_3N^+$, 88 $(C_2H_5)_2NO/C_3H_6NO_2^+$, 95 $C_7H_{11}^+$, 104 $C_8H_8^+$, 115 $C_9H_7^+$, 129 $C_5H_7NO^+$, and anions in panel e: sum of m/z 24 $C_2^-$, 25 $C_2H^-$, 26 $C_2H_2/CN^-$, 42 $C_2H_2O/CNO^-$, 45 $COOH^-$, 59 $CH_2COOH^-$, 71 $CCH_2COOH^-$, 73 $C_2H_4COOH^-$, 85 $C_3H_4COO^-$, 89 $(CO)_2OOH^-$), there is a poor correlation ($\gamma \leq 0.4$) between these two instruments if we consider the entire measurement period. However, the fraction of LAAPTOF organic cations to the sum of ammonium and organic cations, org/(org+ammonium), anion fraction of org/(org+sulfate), and org/(org+nitrate), show better correlations between these two instruments (Fig. 7), especially for org/(org+nitrate). As shown in Fig. 7b, a scatter plot of org/(org+nitrate) measured by LAAPTOF and AMS shows an exponential trend. A similar trend for the ratio carbonaceous/(carbonaceous+sulfate) was observed by PALMS compared to AMS results for free tropospheric aerosol particles measured by Murphy et al. (2006).

Note that the aforementioned comparisons in this section are for the entire measurement period and demonstrate general correlations between these two instruments. Considering different time periods, the correlations vary (Fig. 7). All corresponding Pearson's correlation coefficient (γ) values for the comparisons of compounds measured by LAAPTOF and AMS are summarized in Table S4. During period 4, most of the γ values are above 0.6, suggesting good correlation, which is comparable with the mass comparison results discussed in Sect. 3.2.2. In particular, for the comparison of the org/(org+nitrate) ratio, LAAPTOF and AMS show good/strong correlations for almost the complete measurement time. The corresponding scatter plots are shown in Fig. 7 (b1-b6). Periods 2 and 4, covering more than 50% of the measurement time, show similar exponential trends as the general fit in Fig. 7b, while periods 1, 3, and 5 show a linear correlation (especially in periods 3 and 5). This implies different dominant particle types. Consistent with the observations shown in Fig. 4c, period 2 and 4 are dominated by "Sodium salts" and there are two "Secondary inorganics-Amine" burst events, while period 3 and 5 are dominated by "Biomass burning-Soil" particles containing more organics, which can also be validated by AMS results as shown in Fig. 6. Therefore, we conclude that the relationship between LAAPTOF-org/(org+nitrate) and AMS-org/(org+nitrate) varies due to changing particle types.

Taken together, the correlations shown in Fig. 7 and Fig. S7 may be used to estimate the mass concentrations of non-refractory compounds for LAAPTOF measurements without AMS in rural locations: ammonium mass concentrations can be estimated from Fig. S7(a), afterwards organic mass concentrations can be estimated by using Fig. 7(a), and then nitrate can be estimated from Fig. 7(b) and/or Fig. 7(b1 to b6) once the dominating particle types are determined, and finally the sulfate mass can be estimated from Fig. 7(c).

### 3.4 Particle sources of non-refractory components

The AMS can quantify the bulk particle mass of non-refractory species such as ammonium, nitrate, sulfate, and organics. LAAPTOF measurements suggest that ambient aerosol particles at this location are often internal mixtures of ammonium, nitrate, sulfate, organics, and other characteristic species such as metals. In order to find out the dominant particle class/classes contributing to/donating a certain non-refractory compound measured by AMS (namely compound-donor particle class/classes), we also need the class information of the single particles, which can be achieved by the single particle identification method described in Sect. 2.2, and assume that LAAPTOF has a similar sensitivity to the same components of different particle classes. For nitrate measured by AMS, the dominating nitrate-donor particles with marker peaks at m/z 46 $NO_2^-$ and 62 $NO_3^-$ in LAAPTOF varied in different periods (Fig. 8): "Sodium salts" was the dominating class for the whole measurement campaign, but "Secondary inorganics-Amine" was dominant in its burst events (August 11[th] and 19[th]), while "Biomass burning-Soil" was dominant from August 25[th] to 29[th]. For ammonium measured by AMS, we have observed a similar trend as for "Secondary inorganics-Amine" particles, indicating that the ammonium AMS measured mainly originated from this class. This can be reinforced by comparing with the time series of LAAPTOF marker peaks for ammonium and amine at m/z 18 $NH_4^+$, 30 $NO^+$, 58 $C_2H_5NHCH_2^+$, 59 $(CH_3)_3N^+$, and 88 $(C_2H_5)_2NO/C_3H_6NO_2^+$ (Fig. S8). For sulfate measured by AMS, we cannot infer the dominating donor class, since there is no comparable LAAPTOF class and fragments. This indicates again that this instrument has a low sensitivity to some sulfate containing particles, such as pure ammonium sulfate. For organic compounds measured by AMS, it is also hard to find the comparable class and marker peaks in LAAPTOF data, probably due to two reasons: one is the same as that for sulfate containing particles, and another one is that compared with AMS there are more fragments (cations and anions) arising from organics in LAAPTOF mass spectra. Nevertheless, we have found that peaks at m/z 129 $C_5H_7NO_3^+$ (arising from organonitrates) and 73 $C_2H_4COO^-$ (from organic acids) have a similar trend as the organics measured by AMS (Fig. 9 b and c). At the beginning of the LAAPTOF measurements, the dominating organic-donor class is class 6 "Biomass burning-Organosulfate" (mainly contributing organic acids to be measured by AMS), while at the end of the measurement period this changed to "Sodium salts" rich particles and "Biomass burning-Soil" (mainly contributing organonitrate and organic acids, respectively). Apart from that, aromatic compounds mainly in "Biomass burning-Soil" could

also contribute to the organic mass fraction measured by AMS, especially for the strongest organic burst event towards the end of the measurement period (cf. Fig. 9 d and e).

Although, the LDI matrix effects cannot be completely overcome by using relative ion intensities, the time series of the corresponding maker peaks (Fig. 8, Fig. 9, and Fig. S8) can still be used for preliminary assignments of the bulk species to different particle types.

## 4 Conclusions and atmospheric implications

In this study, we used a combination of representative spectra obtained by fuzzy classification, laboratory based-reference spectra, marker peaks, and typical peak ratios for the improved single aerosol particle identification at a rural site in the upper Rhine valley near the city of Karlsruhe, Germany. Seven major particle classes were identified among a total of $\sim 3.7 \times 10^5$ single particles: "Calcium-Soil"; "Aged soot"; "Sodium salts"; "Secondary inorganics-Amine"; "Biomass burning-Soil"; "Biomass burning-Organosulfate"; and "Mixed/aged-Dust". All particles were internally mixed with organic and secondary inorganic compounds, i.e., ammonium, sulfate, and nitrate. According to our observation, these particles are expected to show a significant hygroscopicity due to their secondary inorganic contents (Fuzzi et al., 2015), as well as the presence of organosulfates (Thalman et al., 2017). The light absorption of soot particles is expected to be enhanced by mixing with non-absorbing species such as organic compounds (Bond et al., 2013). Organonitrates signatures found on "Sodium salts" particles are also expected to increase their light absorbing capability (Canagaratna et al., 2007) and to assist nocturnal particle growth. The good correlation of most of the particle classes and dust signatures suggests that condensation processes and heterogeneous chemistry have modified the dust particles during their transportation. For example, organosulfates coated dust could form from heterogeneous reactions of volatile organic compounds (VOCs), such as glyoxal, on mineral dust particles aged by reaction with e.g. $SO_2$ (Shen et al., 2016). Since organosulfates can form by heterogeneous reactions of IEPOX on acidic particles at low $NO_x$ level (Froyd et al., 2010; Surratt et al., 2010), it is likely that they form also on acidified dust particles at similar conditions. Our general observation of dominating aged and mixed aerosol particles is expected at a location about 2 hours downwind of nearest major emission sources (12 km distance to Karlsruhe at an average daytime wind speed of 1.7 m/s).

Based on the precise identification for particle classes and individual particles, we applied a quantification method for single particles, employing size and particle class/chemically-resolved overall detection efficiencies (ODEs) for this instrument. In contrast to methods used in previous SPMS studies, our approach is laboratory-based and doses not rely on the availability of a reference instrument in the field. The corresponding "corrections" to the standard similarity classification result in substantial changes in the particle class abundancies: "Sodium salts" particles changed from 43.5% (similarity) to 3.4% (corrected number fraction) corresponding to a mass fraction of 30.1%, becoming the second dominating class in mass; "Secondary inorganics-Amine" dramatically decreased from 20.8% to 2.4% corresponding to a mass fraction of 3.6%; becoming the second least abundant class; "Biomass burning-Soil" changed from 16.1% to 45.0% corresponding to a mass fraction of 31.8%, becoming the dominating class in number and mass. The big difference between number-based and mass-based SPMS results has enforced the importance of particle mass quantification. Noteworthy, our quantification approach requires several assumptions mainly regarding particle shape and density, which results in potential uncertainties of up to ~300% with the dominant source still the ODE values. Despite this large uncertainty, the resulting total particle mass show good agreement with the total mass of non-refractory compounds measured by AMS in different periods, covering ~85% of the measurement time. However, some discrepancies still remain most likely due to the low sensitivity of LAAPTOF for small particles as well as ammonium sulfate and organic rich particles. Furthermore, we have found specific relationships of LAAPTOF ion intensities ratios and AMS mass concentration results for non-refractory compounds, especially for the fraction of org/(org+nitrate). This will be applied for source apportionment in an upcoming publication. The corresponding scatter plots may be used to estimate mass concentrations in future SPMS studies as well.

We have shown how particle size, density, morphology (shape), and chemical composition have impact the ODE of the LAAPTOF. Therefore, these factors need to be taken into account for a reasonable quantitative interpretation of SPMS data. Considering reduced quantification uncertainties, systematic measurements on different types of standard samples, as well as real ambient samples (size selected) under controlled environmental conditions (temperature and relative humidity) are still needed to obtain more comprehensive sensitivities for LAAPTOF.

Employing particle class information for individual particles and specific marker peaks with relative ion intensities, this study is able to assign non-refractory compounds measured by AMS to different classes of particles measured by SPMS. It turns out that nitrate measured by AMS was mainly from sodium salts like particles. Ammonium measured by AMS was mainly arising from secondary inorganics-amine particles. However, the dominating donor particle classes varied in different time periods during the measurements. Organic compounds measured by AMS were from organic acids (mainly on aged biomass burning particles), organonitrates (from sodium salts), and aromatic compounds (from aged biomass burning particles). During the entire measurement campaign, the dominating particle classes changed with respect to particle number and mass, and the donor classes for non-refractory compounds also varied substantially indicating changes of particles sources.

In spite of significant uncertainties stemming from several assumptions and instrumental aspects, our study provides a good example for identification and quantitative interpretation of single particle data. Together with the complimentary results from bulk measurements by AMS, we have shown how a better understanding of the internal and external mixing state of ambient aerosol particles can be achieved.

**Data availability**

LAAPTOF reference spectra are available upon request to the corresponding author and are available in electronic format via the EUROCHAMP DATA CENTER – Library of Analytical Resources of the EU project EUROCHAMP-2020 (https://data.eurochamp.org/, EUROCHAMP, 2019).

**Author contributions**

X.S. operated LAAPTOF and AMS during the whole field campaign, did the LAAPTOF data analysis, produced all figures, and wrote the manuscript. H.S. organized the campaign, provided suggestions for the data analysis, interpretation, and discussion. W.H. operated AMS during the whole campaign and did AMS data analysis. C.M. helped to operate the instruments, provided suggestions for the data analysis, interpretation, and discussion. R.R. helped to operate LAAPTOF. T.L. gave general advices and comments for this paper. All authors contributed to the final text.

**Competing interests**

The authors declare no conflict of interest.

**Acknowledgements**

The authors gratefully thank the AIDA staff at KIT for helpful discussions and technical support, and the China Scholarship Council (CSC) for financial support of Xiaoli Shen and Wei Huang. Special thanks go to Daniel Cziczo for discussions about particle identification and quantification methods, to Nsikanabasi Umo for discussions about the coal fly ash sample, and to the Albtal-Verkehrs-Gesellschaft (AVG) for providing power and the measurement location near the tram line.

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

**Table 1: Particle class numbers, names, labels and corresponding signature ion peaks.**

| Class number: name (label) | Signature ion peaks (cations and anions are marked in red and blue, respectively) |
| --- | --- |
| **Class 1:** Calcium rich and soil dust like particles (Calcium-Soil) | 23 $Na^+$, 40 $Ca^+$, 56 $CaO/Fe^+$, 57 $CaOH^+$, 64/66 $Zn^+$, 65 $Cu^+$, 75 $CaCl^+$, 96 $Ca_2O^+$, 112 $(CaO)_2^+$, 138 $Ba^+$, 154 $BaO^+$, 206-208 $Pb^+$ |
| **Class 2:** Aged soot like particles (Aged soot) | 12n $Cn^+$, 206-208 $Pb^+$ 12n $Cn^-$; sulfate (32 $S^-$, 64 $SO_2^-$, 80 $SO_3^-$, 81 $HSO_3^-$, 97 $HSO_4^-$, 177 $SO_3HSO_4^-$, 195 $HSO_4H_2SO_4^-$) |
| **Class 3:** Sodium salts like particles (Sodium salts) | 23 $Na^+$, 39 $NaO/K^+$, 40 $Ca^+$, 46 $Na_2^+$, 62 $Na_2O^+$, 63 $Na_2OH^+$, 81/83 $Na_2Cl^+$, 92 $Na_2NO_2^+$, 108 $Na_2NO_3^+$, 129 $C_5H_7NO_3^+$, 141 $Na_3Cl_2^+$, 149 $C_4H_7O_2NO_3^+$, 165 $Na_3SO_4^+$, 181 $C_4H_7O_4NO_3^+$, 206-208 $Pb^+$ 35/37 $Cl^-$, 93/95 $NaCl_2^-$, 104 $NaClNO_2^-$, 111 $NaCl_2H_2O^-$, 115 $Na(NO_2)_2^-$, 119 $NaSO_4/AlSiO_4^-$, 120 $NaClNO_3^-$, 131 $NaNO_2NO_3^-$, 147 $Na(NO_3)_2^-$, 151/153 $Na_2Cl_3^-$, 177 $NaClNaSO_4/SO_3HSO_4^-$ |
| **Class 4:** Secondary inorganics rich and amine containing particles (Secondary inorganics-Amine) | ammonium and amine (18 $NH_4^+$, 27 $C_2H_3/CHN^+$, 28 $CO/CH_2N^+$, 30 $NO^+$, 43 $C_3H_7/C_2H_3O/CHNO^+$, 58 $C_2H_5NHCH_2^+$, Amine 59 $(CH_3)_3N^+$, 86 $(C_2H_5)_2NCH_2^+$, 88 $(C_2H_5)_2NO/C_3H_6NO_2^+$, 118 $(C_2H_5)_2NCH_2^+$) nitrate (46 $NO_2^-$, 62 $NO_3^-$); sulfate |
| **Class 5:** Aged biomass burning and soil dust like particles (Biomass burning-Soil) | 39 $K/C_3H_3^+$, 41 $K/C_3H_5^+$, 43 $C_3H_7/C_2H_3O^+$, 50 $C_4H_2^+$, 53 $C_4H_5^+$, 55 $C_4H_4/C_3H_3O^+$, 63 $C_5H_3^+$, 77 $C_6H_5^+$, 85 $C_7H^+$, 91 $C_7H_7^+$, 95 $C_7H_{11}^+$, 104 $C_8H_8^+$, 115 $C_9H_7^+$, 138 $Ba^+$, 154 $BaO^+$, 175 $K_2HSO_4^+$, 206-208 $Pb^+$, 213 $K_3SO_4^+$ sulfate |
| **Class 6:** Aged biomass burning and organosulfate containing particles (Biomass burning-Organosulfate) | positive signature peaks feature biomass-burning very similar as given for class 5 organosulfate (141 $C_2H_5O_8O_4^-$, 155 $C_2H_3O_2SO_4^-$, 215 $C_5H_{11}O_3SO_4^-$) |
| **Class 7:** Mixed/aged and dust like particles (Mixed/aged-Dust) | contains almost all the signature peaks from the other classes |

Note that "rich" used in the names stands for the strong spectral signal rather than the real mass fraction.

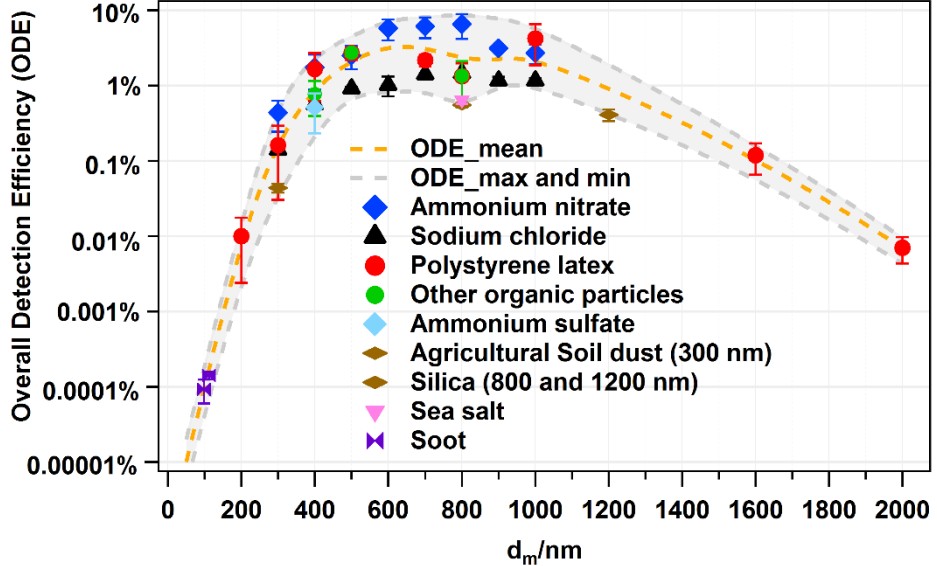

**Figure 1:** Overall detection efficiency of LAAPTOF for different types of particles as a function of the mobility diameter ($d_m$), adapted from Shen et al. (2018) and extended. Dashed lines are fitting curves for maximum, mean and minimum values of ODE. For other organic particles (green), ODE at 400 nm is the data from secondary organic aerosol (SOA) particles from α-pinene ozonolysis, ODE at 500 nm is the data from humic acid, and ODE at 800 nm is data from humic acid ($1.9 \pm 0.3\%$), oxalic acid ($0.3 \pm 0.1\%$), pinic acid ($1.6 \pm 0.1\%$), and cis-pinonic acid ($1.9 \pm 0.7\%$). SOA particles were formed in the Aerosol Preparation and Characterization (APC) chamber and then transferred into the AIDA chamber. Agricultural soil dust (brown symbol) were dispersed by a rotating brush generator and injected via cyclones into the AIDA chamber. Sea salt particles (purple) were also sampled from the AIDA chamber. Soot particles from incomplete combustion of propane were generated with a propane burner (RSG miniCAST; Jing Ltd.), and then injected into and sampled from a stainless steel cylinder of ~0.2 m³ volume. SiO₂ particles were directly sampled from the headspace of their reservoirs. The other aerosol particles shown in this figure were generated from a nebulizer and size-selected by a DMA. Note that there is uncertainty with respect to particle size due to the particle generation method. The nebulized and DMA sized samples have relative smaller standard deviation (SD) from Gaussian fitting to the measured particle sizes. PSL size has the smallest size SD (averaged value is 20 nm) and the corresponding relative SD (RSD = SD divided by the corresponding size) is ~6%, since the original samples are with certain sizes. The other nebulized samples have standard deviations ranging from 70 to 120 nm SD and 3 to 23% RSD. Particles sampled from AIDA chamber have much bigger size SD: ~70 nm for SOA (17% RSD), ~100 nm for agricultural soil dust (~83% RSD) and ~180 nm for sea salt particles (~34% RSD). Considering this uncertainty, we have chosen size segment of 100 nm ($\pm$50 nm) for correction, e.g., particles with size of 450 to 550 nm will use the ODE at 500 nm particle number correction.

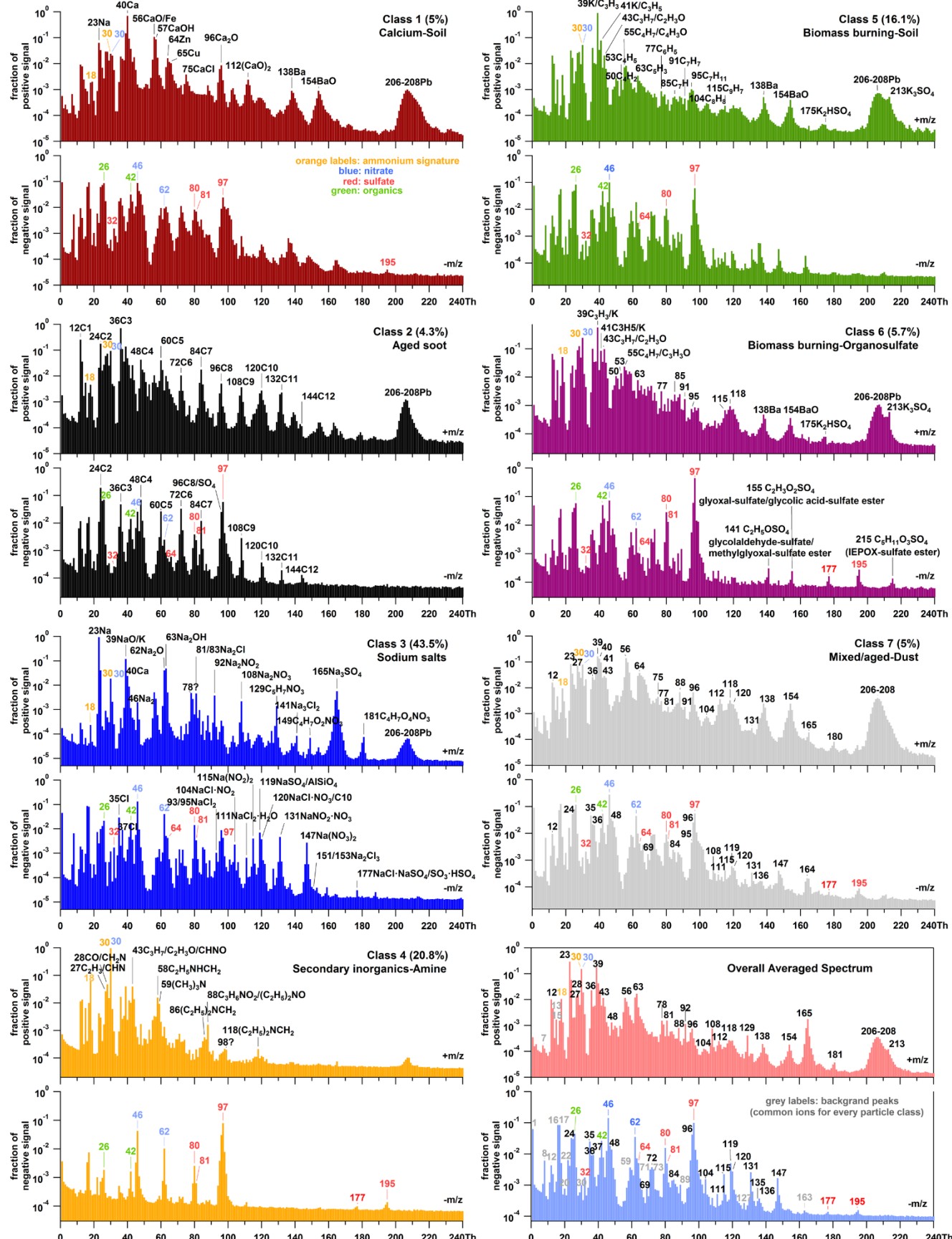

**Figure 2: Representative mass spectra of seven particle classes measured during the field campaign TRAM01, based on fuzzy classification according to fuzzy *c*-means clustering algorithm, and averaged spectrum of total ~3.7 × 10⁵ single particles measured. The percentage in each pair of spectra gives us information about the similarity of the total aerosol particles to different classes. Black labels represent the ions characteristic for different classes. The red, blue, and orange labels represent the signatures for sulfate (32 S⁻, 64 SO₂⁻, 80 SO₃⁻, 81 HSO₃⁻, 97 HSO₄⁻, 177 SO₃HSO₄⁻, and 195 HSO₄H₂SO₄⁻), nitrate (30 NO⁺, 46 NO₂⁻, and 62 NO₃⁻) and ammonium (18 NH₄⁺ and 30 NO⁺). The green labels represent the organic compounds (26 C₂H₂/CN⁻ and 42 C₂H₂O/CNO⁻). In the overall averaged spectrum, grey labels represent the background fragments (common ions) that exist for every particle class.**

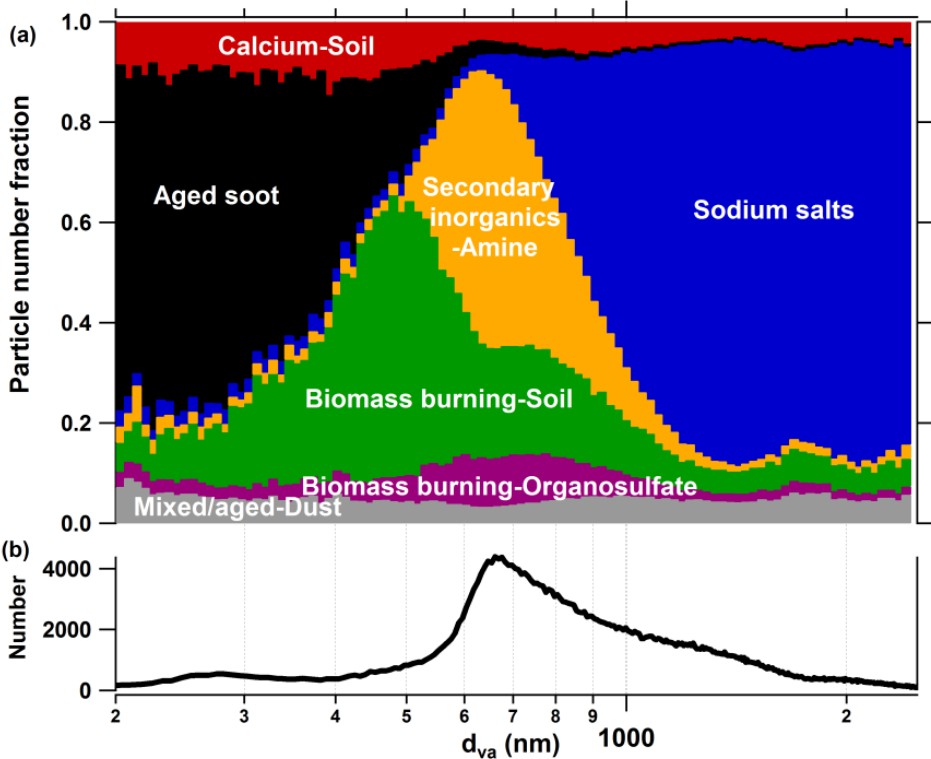

**Figure 3: (a) Size resolved number fraction for seven particle classes measured during the field campaign TRAM01, based on fuzzy classification according to fuzzy c-means clustering algorithm. (b) Overall size distribution for the particles measured by LAAPTOF during the whole campaign.**

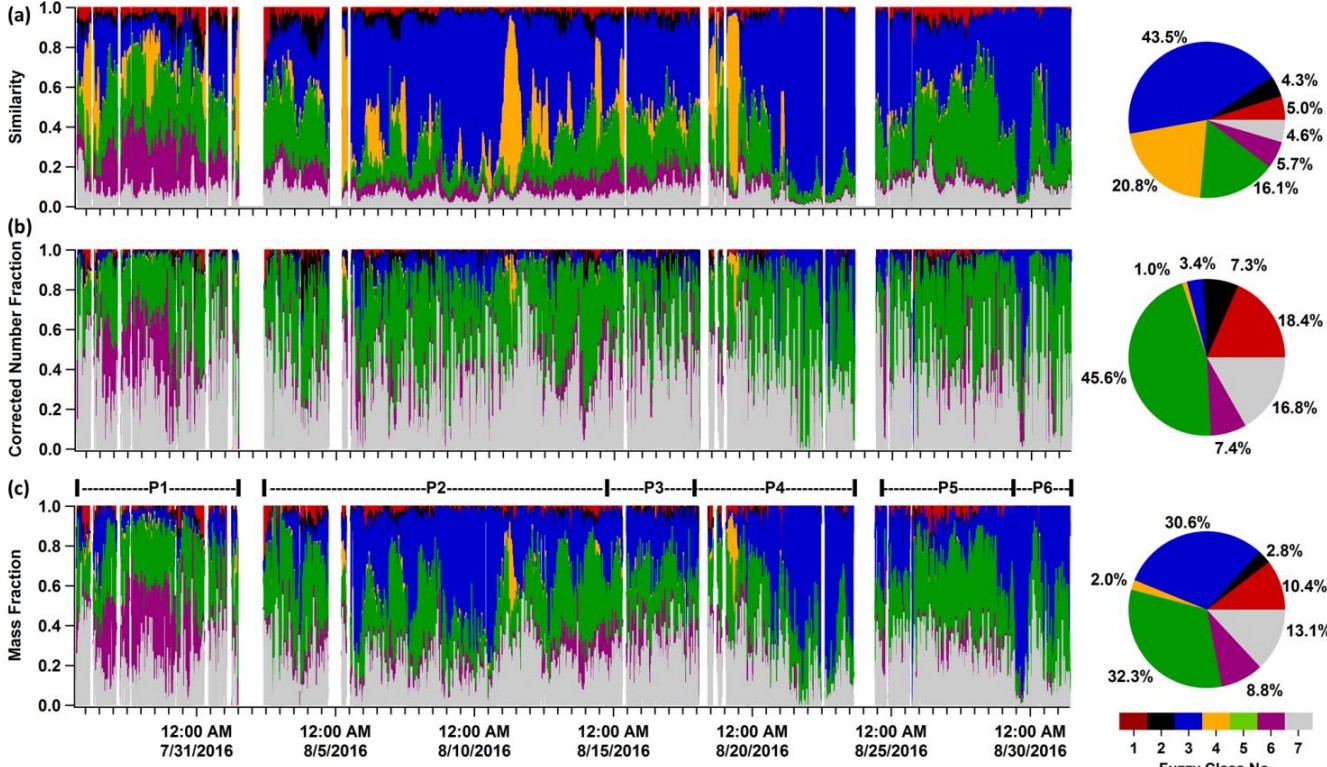

**Figure 4: Time series of the similarity, corrected number fraction, and mass fraction of seven major particle classes and the corresponding pie charts for total fractions. Note that, the correction shown here is based on a chemically or particle class resolved ODE. The seven classes are class 1 "Calcium-Soil"; class 2 "Aged soot"; class3: "Sodium salts"; class 4 "Secondary inorganics-Amine"; class 5 "Biomass burning-Soil"; class 6 "Biomass burning-Organosulfate"; and class 7 "Mixed/aged-Dust". In panel (c), 4 periods have been marked: P1 is Period 1 from 7/26/2016 16:23 to 8/1/2016 11:43; P2 from 8/2/2016 09:43 to 8/14/2016 17:53; P3 from 8/14/2016 18:03 to 8/17/2016 21:03; P4 from 8/17/2016 21:13 to 8/23/2016 15:33; P5 from 8/24/2016 15:03 to 8/29/2016 08:33; P6 from 8/29/2016 08:43 to 8/31/2016 09:13.**

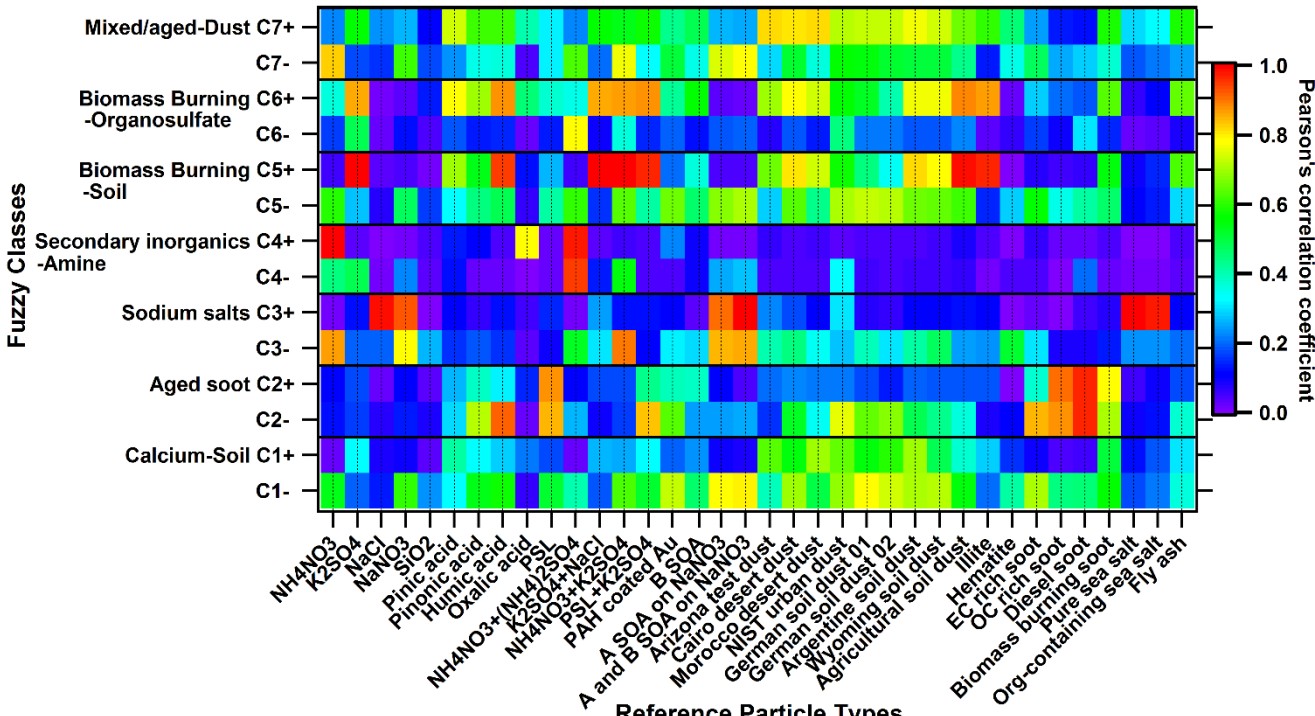

**Figure 5: Correlation diagram of fuzzy classification results (7 classes, C1 to C7) and 36 laboratory-based reference spectra. Correlation results for the positive and negative spectra (e.g., for C1) are in the separated rows (e.g., C1+ and C1-). PAH is short for poly(allylamine hydrochloride), B SOA is short for biogenic SOA (α-pinene SOA in this study), A SOA is short for anthropogenic SOA (toluene SOA in this study), biomass burning soot is the lignocellulosic char from Chestnut wood. Note that, the strong and good correlations mentioned in the paper stand for Pearson's correlation coefficient γ≥ 0.8 and γ≥ 0.6, respectively. The seven classes are class 1 "Calcium-Soil"; class 2 "Aged soot"; class3: "Sodium salts"; class 4 "Secondary inorganics-Amine"; class 5 "Biomass burning-Soil"; class 6 "Biomass burning-Organosulfate"; and class 7 "Mixed/aged-Dust".**

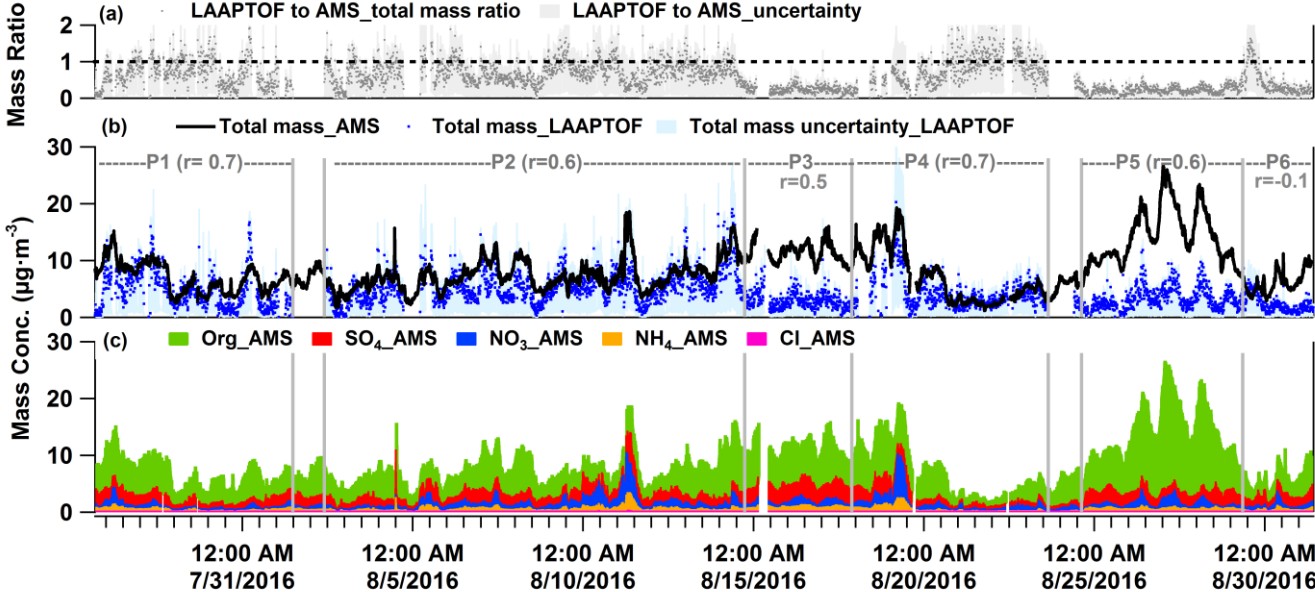

**Figure 6: Time series of (a) total mass ratio of LAAPTOF to AMS data, (b) LAAPTOF total mass and AMS total mass (c) mass concentrations of organic, sulfate, nitrate, and ammonium compounds measured by AMS. In panel (b) r is the Pearson's correlation coefficient between LAAPTOF and AMS results. P1 is Period 1 from 7/26/2016 16:23 to 8/1/2016 11:43; P2 from 8/2/2016 09:43 to 8/14/2016 17:53; P3 from 8/14/2016 18:03 to 8/17/2016 21:03; P4 from 8/17/2016 21:13 to 8/23/2016 15:33; P5 from 8/24/2016 15:03 to 8/29/2016 08:33; P6 from 8/29/2016 08:43 to 8/31/2016 09:13. Zoom in figures for P1, 2, 4, and 5 can be found in Fig. S5, as well as the corresponding scatter plots for LAAPTOF and AMS data comparison.**

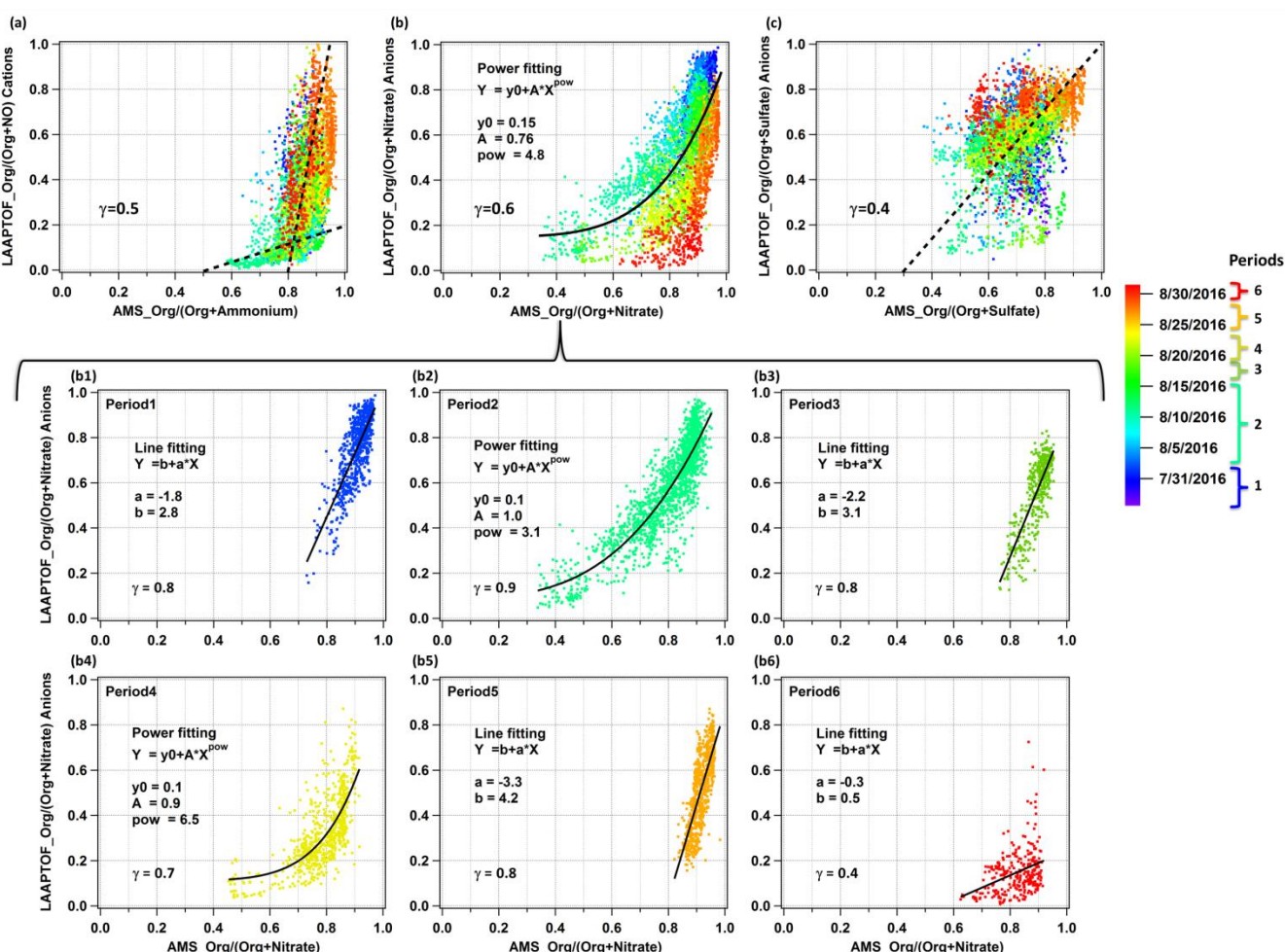

**Figure 7: Comparison of non-refractory compounds measured by LAAPTOF and AMS: (a) LAAPTOF organic cations and NO+ fractions Org/(Org+NO), (b) organic anions and nitrate fractions Org/(Org+Nitrate), (c) organic anions and sulfate fractions Org/(Org+Sulfate) to the corresponding AMS mass fractions. Each point is 10 min averaged data, and there are 4483 points in each scatter plot. Dashed line in panel (a) and (c) are used to guide the eyes, while the curve in panel (b) is from the fitting result. Colour scale is related to the timeline, including periods 1 to 6, same as the ones in Fig.6. Further comparison of Org/(Org+Nitrate) during 6 periods are in the scatter plots (b1) to (b6).**

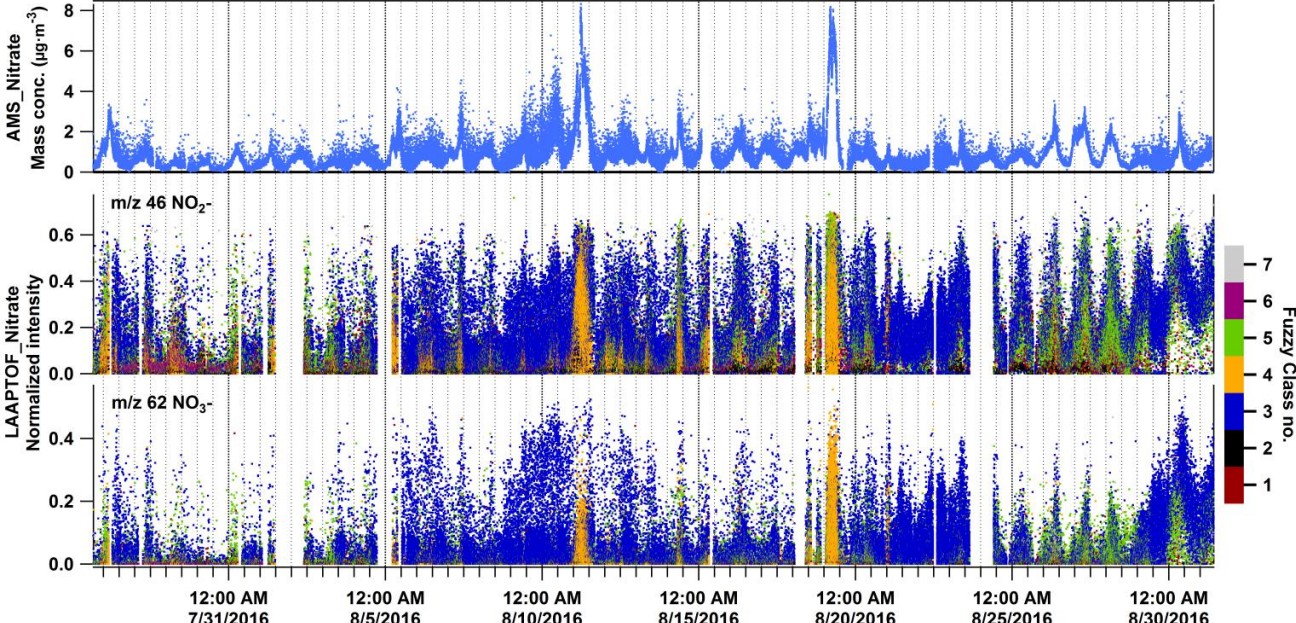

**Figure 8: Time series of nitrates measured by AMS in mass concentration and LAAPTOF in normalized ion intensities, respectively. Normalized intensity refers to the fragment intensity divided by sum of all the ion intensities. Marker peaks for nitrates are at m/z 46 NO$_2^-$ and 62 NO$_3^-$ in LAAPTOF spectra. The seven fuzzy classes are class 1 "Calcium-Soil"; class 2 "Aged soot"; class3: "Sodium salts"; class 4 "Secondary inorganics-Amine"; class 5 "Biomass burning-Soil"; class 6 "Biomass burning-Organosulfate"; and class 7 "Mixed/aged-Dust".**

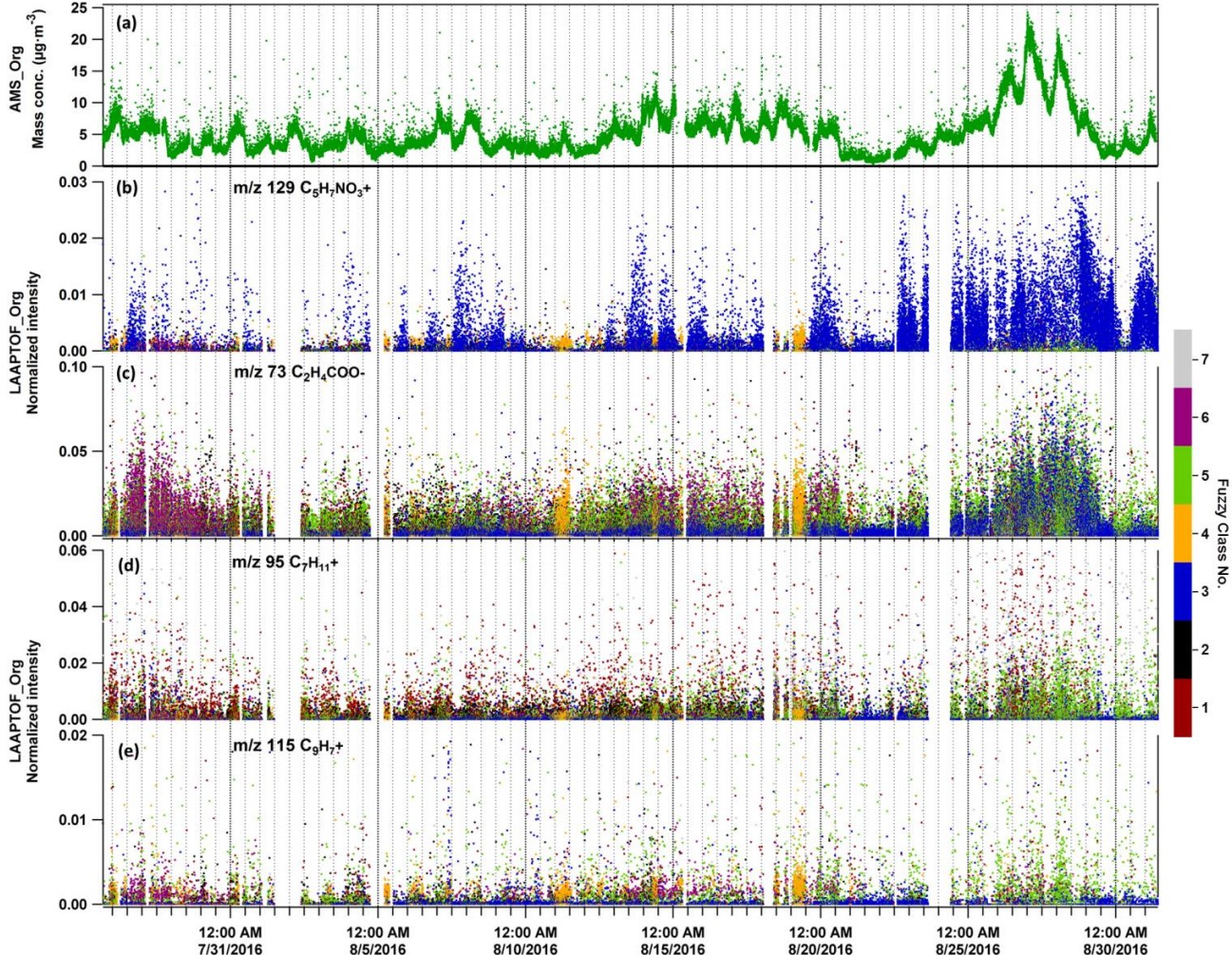

**Figure 9: Time series of organic species measured by AMS in mass concentration and LAAPTOF in normalized ion intensities, respectively. Normalized intensity refers to the fragment intensity divided by sum of all the ion intensities. In LAAPTOF spectra, the peak at m/z 129 $C_5H_7NO_3^+$ is arising from organonitrates, m/z 73 $C_2H_4COO^-$ from organic acids, and m/z 95 $C_7H_{11}^+$ as well as m/z 115 $C_9H_7^+$ are from aromatic compounds. The seven classes are class 1 "Calcium- Soil"; class 2 "Aged soot"; class3: "Sodium salts"; class 4 "Secondary inorganics-Amine"; class 5 "Biomass burning-Soil"; class 6 "Biomass burning-Organosulfate"; and class 7 "Mixed/aged-Dust".**