# Peer review of "Understanding atmospheric aerosol particles with improved particle identification and quantification by single particle mass spectrometry"

_Atmospheric Measurement Techniques, 2018_

## Referee Comment (RC1) · Anonymous Referee #1 · 4 Nov 2018

General comments: This manuscript evaluates the capability of a type of Single particle mass spectrometry (SPMS) to quantify the mass concentration of individual particles, with 6-week field measurement data. Seven major particle classes were concerned through using fuzzy classification, peak area information, and laboratory-based reference spectra. They show the significant difference between the observed particle number fracion and estimated mass contribution. It is interesting that the provided approach could assign the non-refractory compounds measured by AMS to different particle classes measured by the LAAPTOF. The authors also carefully estimate the error associated with the approach. I recommend publication of this manuscript with minor revison.

[Figure]

Specific comments: 1. A discussion on the representative of the field measurement data would be necessary in the revised version. For example, a detail comparison of the identified particle classes with those previously observed in similar region.

2. Healy et al., 2013 has quantitatively determined the mass contribution for each carbonaceous particle classes. Inclusion of this in the introduction and discussion would be necessary for completeness.

3. Lines 57-60: "This provides different sources for the non-refractory species measured by AMS and indicates different sources of aerosol" might be not appropriate. I think a major part of non-refractory species measured by AMS should be secondary.

4. Fig. 2: it is possible to compare the mass concentration of AMS and LAAPTOF in different size range? From Fig. 1, it can be seen significant difference of ODE in difference size range? A comparison of AMS and LAAPTOF in different size range might help reduce the difference.

5. Section 2.2 line 10-15. I do not understand why "A direct class-dependent quantification of particle mass is therefore not possible."

Is it possible that the threshold values set for the positive and negative spectra correlation influence the assignment of individual particles to difficult particle types?

6. Page 4 Line 29 "This leads to an uncertainty of ∼100% in particle mass." is it only for sea salt like particles? How about other particle classes?

7. Page 4 Lines 35-40: The assumption of single density value for each particle classes might introduce large uncertainty. The author should adapt a possible density range through the previous publications and evaluate the uncertainty for each assumption. This would also help reduce the overall difference between the comparison with AMS results.

8. Page 6 Line 25 I think it would be better to include some references for the identification of amines.

[Figure]

References Healy, R.M., Sciare, J., Poulain, L., Crippa, M., Wiedensohler, A., Prevot, A.S.H., Baltensperger, U., Sarda-Esteve, R., McGuire, M.L., Jeong, C.H., McGillicuddy, E., O'Connor, I.P., Sodeau, J.R., Evans, G.J., Wenger, J.C., 2013. Quantitative determination of carbonaceous particle mixing state in Paris using single-particle mass spectrometer and aerosol mass spectrometer measurements. Atmospheric Chemistry And Physics 13, 9479-9496.

---

## Referee Comment (RC2) · Anonymous Referee #2 · 24 Nov 2018

Shen et al describe single-particle mass spectrometry (SPMS) data analysis using LAAPTOF data from a summer 2016 field campaign in rural Germany. While the SPMS data itself appears sound, there are many major technical issues with their analyses, as well as their assertions of originality. Unfortunately, the authors appear to be unaware of the majority of the SPMS literature, which their work would highly benefit from. Please see below for description of major issues, with references to previous literature that I hope will be useful for the authors to place their current work in context and aid in their data analysis and interpretation. I encourage the authors to rethink the framing of their manuscript and, instead of focusing on data analysis methods, consider the science that can be learned from their data itself by examining particle composition

as a function of time and meteorological conditions, for example.

Method development to obtain mass concentrations from SPMS data was previously shown through method development papers by Allen et al. (2000, Environ. Sci. Technol., "Particle detection efficiencies of aerosol time of flight mass spectrometers under ambient sampling conditions"), Fergenson et al. (2001, Analytical Chemistry, "Quantification of ATOFMS data by multivariate methods"), Wenzel et al. (2003, J. Geophys. Res., "Aerosol time-of-flight mass spectrometry during the Atlanta Supersite Experiment: 2. Scaling procedures", Zhao et al. (2005, Analytica Chimica Acta, "Predicting bulk ambient aerosol compositions from ATOFMS data with ART-2a and multivariate analysis"), Allen et al. (2006, Aerosol Sci. Technol., "Instrument busy time and mass measurement using aerosol time-of-flight mass spectrometry"), Bein et al. (2006, Atmos. Environ., "Identification of sources of atmospheric PM at the Pittsburgh Supersite – Part II: Quantitative comparisons of single-particle, particle number, and particle mass measurements"), and Qin et al. (2006, Analytical Chemistry, "Comparison of two methods for obtaining quantitative mass concentrations from aerosol time-of-flight mass spectrometry measurements". The authors "new" method is very similar to the work discussed these older papers, yet these papers are not even cited in the current paper. Many subsequent SPMS papers have used these approaches to provide chemically-resolved mass concentrations: Bhave et al. (2001, Environ. Sci. Technol.), Ault et al. (2009, Environ. Sci. Technol.), Qin et al. (2012, Atmos. Environ.), Healy et al. (2012, Atmos. Chem. Phys.), Healy et al. (2013, Atmos. Chem. Phys.), Gunsch et al. (2018, Atmos. Chem. Phys.), and May et al. (2018, Environ. Sci. Technol. Lett.). I highly suggest that the authors review these previous papers to decide how to move forward with their own work, placing it into the context of previous studies.

The authors assert in the abstract (Page 1, Lines 30-31) that "[their] approach allows for the first time to assign the non-refractory compounds measured by AMS to different particle classes." Similarly, in the conclusions section, it is stated "our study…opens a new way for quantitative information of single particle data, and together with the

complimentary results from bulk measurements by AMS we have shown how a better understanding of the internal and external mixing state of ambient aerosol particles can be achieved." These statements are not accurate, as many SPMS analyses have incorporated bulk aerosol composition data (both off-line impactor and online AMS): Bhave et al (2002, Environ. Sci. Technol.), Middlebrook et al. (2003, J. Geophys. Res.), Spencer & Prather (2006, Aerosol Sci. Technol.), Ferge et al. (2006, Environ. Sci. Technol.), Drewnick et al. (2008, Atmos. Environ.), Dall'Osto et al (2009, Atmos. Chem. Phys.), Pratt et al. (2010, J. Atmos. Sci.), Pratt et al. (2010, J. Geophys. Res.), Pratt et al. (2011, Atmos. Chem. Phys.), Decesari et al. (2011, J. Geophys. Res.), Dall'Osto & Harrison (2012, Atmos. Chem. Phys.), Dall'Osto et al. (2012, Aerosol Sci. Technol.), Dall'Osto et al. (2012, J. Geophys. Res.), Dall'Osto et al. (2013, Atmos. Chem. Phys.), Healy et al. (2013, Atmos. Chem. Phys.), Decesari et al. (2014, Atmos. Chem. Phys.), Gunsch et al. (2018, Atmos. Chem. Phys.), and others. In fact the title of Healy et al. (2013, Atmos. Chem. Phys.) is "Quantitative determination of carbonaceous particle mixing state in Paris using single-particle mass spectrometer and aerosol mass spectrometer measurements". Again, these papers are not cited in the current work and should be considered in their data interpretation and discussion.

Figure 1, which shows the overall detection efficiencies for various particle types, as determined in the laboratory, is useful. However, apparently these data are all already published in Shen et al. (2018, Atmos. Meas. Technol.), unfortunately limiting the originality here. It would be great if additional particle type proxies, based on those observed in the field could be added (e.g. soot, biomass burning). Considering these data and Figure 2 (dominance of soot from 0.2-0.4 um), I encourage the authors to characterize their detection efficiency of soot particles in the laboratory. Also, the authors apply this laboratory-derived ODE to their field data, but it is not discussed whether the ODE was verified in the field, or how reproducible it is in the field. Furthermore, the authors note that using the mean ODE introduces significant uncertainty; this is also why the previously published methods (e.g. Qin et al 2006) determine the detection efficiencies in the field with time. Also, in presenting this curve to LAAPTOF users, it

is important to note that this curve should not be extrapolated to other LAAPTOF or SPMS instruments without a standard to check against (e.g. PSLs). The authors simply note that "alignment and variance in particle-laser interaction lead to uncertainty in ODE" (Page 5, Lines 33-34). This paragraph suggests that this variance is included in the 540% ODE spread for various lab-generated aerosols; however, it should be noted that the ODE dependence on sizing laser and desorption/ionization laser powers and alignments will change with time, especially when the instrument is moved. This is why the previously published methods noted above (e.g. Qin et al. 2006) characterize detection efficiencies in the field. For example, Jeong et al. (2011, Atmos. Chem. Phys.) and Wenzel et al. (2003) show how the hit fraction of particles can change with time during and between field campaigns; this is taken into account in the Wenzel et al. (2003) and Qin et al (2006) method. Also, for consideration of "variance in particle-laser interaction" (Page 5, Line 34) the authors may be interested to review Wenzel & Prather (2004, Rapid. Commun. Mass Spec., "Improvements in ion signal reproducibility obtained using a homogeneous laser beam for on-line laser desorption/ionization of single particles").

Page 5, Lines 3-28: If the authors have chemically-resolved ODEs, why did then choose to apply a mean ODE to all data? If the ODEs are stable relative to a standard (e.g. PSLs), it seems like a strength of the current work that chemically-resolved ODEs, for each particle type, could be applied in the calculation of mass concentrations.

The authors make many subjective statements that can be refuted by previously published literature, such as those mentioned above (and others not listed here). I caution the authors from making such statements. For example, "SPMS is a useful, albeit not fully quantitative tool" (Page 1, Line 14) and "SPMS data analysis has been proven difficult under real world conditions" (Page 2, Lines 2-3).

The authors state that "mass spectroscopic signatures do not necessarily reflect the primary composition of the particles" (page 2, lines 7-8). However, many previous pa-

pers (e.g., Bhave et al. 2001, Environ. Sci. Technol., Reinard et al., 2007, Atmos. Environ., Toner et al. 2008, Atmos. Environ., Pratt & Prather 2009, Environ. Sci. Technol., Healy et al. 2010, Atmos. Chem. Phys., and others) have examined the SPMS source signatures of primary particles and apportioned ambient particles according to these signatures, which is arguably one of the strengths of SPMS.

Page 3, Line 36-37: The authors state that the LAAPTOF has a size range of 70 to 2500 nm; however, Figure 1 shows that the detection efficiencies of particles <400 nm and >1200 nm is extremely low (<1%), making this earlier statement seem misleading.

Section 2.2: The authors apply particle densities to each particle class and assume that all particles are spherical. Instead the authors could consider applying measured individual particle effective densities (reducing assumptions) (e.g. Zelenyuk et al. 2008, Analytical Chem., Spencer et al. 2007, Environ. Sci. Technol., Zhang et al. 2016, Atmos. Chem. Phys.), as has been done in other SPMS studies converting to mass concentrations (e.g. Qin et al. (2012, Atmos. Environ.), Gunsch et al. (2018, Atmos. Chem. Phys.), and May et al. (2018, Environ. Sci. Technol. Lett.).

Section 3.1: This section does not provide any new information in terms of methods/technology; this is simply a description of mass spectra particle types observed during the field study. I also have several concerns about particle type identification, mostly with the respect to the attribution of nearly all particles as "dust-like", which does not have support by the m/z marker ions shown and described, as I note below. In fact, mass spectral markers supporting dust are only shown (in Fig 2) and discussed here for Class 1 (5%, by number) and Class 7 (4.6%, by number), leaving >90% of the particle as non-dust particles. Also, it would be useful to move Figure 4 and its discussion to this section, as that figure is useful and aids with particle type classification.

Particle Class 4: The authors call these "Secondary inorganic and amine like particles" and discuss secondary markers on lines 24-25 (page 6) and larger size (0.5-1 um) (line 27). Yet, the next sentences (lines 26-27) states "class 4 is relatively "clean" with

the fewest peaks, indicating that the particles might be relatively fresh." There is no mass spectral support for these as freshly emitted particles; in fact the authors say "secondary" in the naming of the particle class. This discrepancy must be fixed.

Particle Class 5: The authors call these "Potassium rich and aromatics coated dust-like particles", with m/z 39 (K+), aromatic marker peaks, and m/z 213 (K3SO4+). The authors note that these "particles might originate from biomass burning". No mass spectral support is provided for identification as dust. Based on other SPMS literature of biomass burning studies, I believe this particle class should be labeled as "Biomass Burning".

Particle Class 6: The authors call these "Organosulfate coated dust-like particles", with organosulfate marker ions. Again, no mass spectral marker ions are discussed to support identification as dust. Also, these particles seem very similar to Class 5 (with large K+, m/z 213, etc); could they correspond to more aged biomass burning particles?

Page 5, Line 41: m/z 24 (C2-) is attributed here to organics, which is possible for 193 nm at high laser pulse energy (e.g., Zelenyuk et al. 2009, Int. J. Mass Spec.); however, it is also a common elemental carbon marker peak (e.g., Zelenyuk et al. 2017, Int. J. Engine Res., Spencer et al. 2006, Aerosol Sci. Technol.).

Page 6, Lines 6-8: I am quite confused by this statement. Does this mean that fuzzy classification does not separate individual mass spectra into individual clusters? Or, if this isn't the case, why are the authors using "similarity" to estimate the number fraction of particles in each group? Why not simply count the number of mass spectra in a given group and then divide by the total number of particles sampled? Please clarify. Neutral networking algorithms used previously in SPMS (e.g. Rebotier & Prather 2007, Analytical Chimica Acta) separate individual mass spectra (corresponding to individual particles) into separate clusters such that it is simple to calculate number fractions.

Figure 2: I suggest raising the cut-off intensity for the mass spectra peak areas, as

[Figure]

there is significant noise shown in all mass spectra currently. Some labeled peaks also do not appear to be above the limits of detection; please check. Also, it would be useful to add the particle type names, in addition to the numbers, to the labeling of the mass spectra. Please also clarify what is meant by "background fragments that exist for every particle class"; do the authors mean common ions, or do they mean that there is a chemical background somehow in the mass spectrometer?

Figure 3: Given the large number fraction of EC particles (class 2) in Figure 2, why is this not reflected in Figure 3, especially since the particle number concentration mode should be at less than 0.2 um? I'm concerned that there could be a problem here in the application of the ODE to the "corrected number fractions" shown here. Also, since the authors have taken the time to convert to number and mass concentrations, it would be useful to show to chemically-resolved number and mass concentration time series plots. It would also be useful here and throughout to refer to particle classes chemically (e.g. dust, EC, biomass burning, organic carbon-sulfate) rather than numbers that require the reader to refer regularly back to Section 3.1.

Page 7, Lines 26-27: The authors state here "...there is no well-defined relationship between spectral signal and quantity." I disagree with this statement, as many SPMS (and LDI generally) papers have investigated this relationship, which is governed by ionization energies of species. Based on the statements on Page 2, Lines 20-26, I'm concerned that the authors may have some confusion about LDI, which is known to primarily result in neutral (rather than ion) formation. I encourage the authors to read textbook or review literature on MALDI and LDI (e.g., Zenobi & Knochenmuss 1998, Mass Spec. Rev., "Ion Formation in MALDI Mass Spectrometry"). For example, in the positive ions, typically the largest ions correspond to those with the lowest ionization energies (of those in the sample). SPMS data analysis methods to account for LDI matrix effects are discussed by Hatch et al (2014, Aerosol Sci. Technol.). The authors do mention in the introduction that SPMS relative sensitivity factors that account for differences in ionization energies are discussed by Gross et al. (2000, Analytical Chem.).

Woods et al. (2001, Analytical Chem., "Quantitative detection of aromatic compounds in single aerosol particle mass spectrometry") is another reference. Thomson et al. (1997, Aerosol Sci. Technol., "Thresholds for laser-induced ion formation from aerosols in a vacuum using ultraviolet and vacuum-ultraviolet laser wavelengths"), Thomson & Murphy (1993, Applied Optics, "Laser-induced ion formation thresholds of aerosol particles in a vacuum"), and Reinard & Johnson (2007, J. Am. Soc. Mass Spec.) will likely also be useful to the authors. Other papers discussing relationships between species quantities and SPMS ion signals are provided above in the comments about SMPS quantification and comparisons with bulk measurements.

Sections 3.2 & 3.3: My comments about this section are primarily summarized above in my notes about previous work producing mass concentrations from SPMS data. I want to add here two additional comments. 1) Given the differences in typical detection efficiencies between the LAAPTOF and AMS, one would not expect good correlations without examining only the size range of overlap, which I would encourage the authors to do. Figure 5 should be revised accordingly. I also encourage the authors to look at previous SPMS-AMS comparisons (see comments above) for greater interpretation of their results and also for additional ways to conduct this analysis that have previously been successful. For individual ions (Page 8, Lines 22-34), I suggest the authors look at Hatch et al. (2014, Aerosol Sci. Technol.) and Healy et al (2013, Atmos. Chem. Phys., "Quantitative determination of carbonaceous particle mixing state in Paris using single-particle mass spectrometer and aerosol mass spectrometer measurements"). 2) The authors note that sulfate salts may have been missed during P3. As such I suggest the authors consider the work of Wenzel et al. (2003, J. Geophys. Res.), who developed a method to identify and quantify "missed" particles, consistent with pure ammonium sulfate.

Figures 6-8: Due to matrix effects in LDI, peak areas of a given species will depend on the full matrix of a particle, such that an ammonium peak area for a given quantity would be expected to be different on a dust particle vs an organic carbon particle. As

[Figure]

such, it is not advised to compare peak areas across all particle types together. Please see Hatch et al. (2014, Aerosol Sci. Technol.). Also see Healy et al. (2013, Atmos. Chem. Phys.) for suggestions of how to compare SPMS and AMS data, as they found agreement when the data are handled properly. Bhave et al. (2002, Environ. Sci. Technol.) may also be useful, as they compared ammonium and nitrate SPMS data to bulk filter data. For organics in particular, the authors may consider Spencer & Prather 2006 (Aerosol. Sci. Technol., "Using ATOFMS to determine OC/EC mass fractions in particles") and Ferge et al. (2006, Environ. Sci. Technol.).

---

## Author Response (AR1)

*Author's response to the reviews:*

*We thank the reviewers for their helpful comments to improve the quality of our manuscript. Our point-to-point replies to the individual comments are in italics, marked by R. as follows:*

**Referee #1 comments:** General comments: This manuscript evaluates the capability of a type of Single particle mass spectrometry (SPMS) to quantify the mass concentration of individual particles, with 6-week field measurement data. Seven major particle classes were concerned through using fuzzy classification, peak area information, and laboratory-based reference spectra. They show the significant difference between the observed particle number fraction and estimated mass contribution. It is interesting that the provided approach could assign the non-refractory compounds measured by AMS to different particle classes measured by the LAAPTOF. The authors also carefully estimate the error associated with the approach. I recommend publication of this manuscript with minor revision.

Specific Comments:

1. A discussion on the representative of the field measurement data would be necessary in the revised version. For example, a detail comparison of the identified particle classes with those previously observed in similar region.

*R1: We have found some previous studies in the similar region, and added them in **section 3.1** "Identification of particle classes and the internal mixing", as follows:*

*1ˢᵗ paragraph in this section "…Similar species were previously identified off-line in the same region (Faude and Goschnick, 1997; Goschnick et al., 1994)."*

*4ᵗʰ paragraph: "…In fact, previous studies identified soil dust as the particle type dominating the coarse particles sampled in the same region (Faude and Goschnick, 1997; Goschnick et al., 1994). Goschnick et al. (1994) found a core-shell structure in both submicron and coarse particles collected north of the Karlsruhe city of Karlsruhe in the upper Rhine valley. This supports our hypothesis.…"*

2. Healy et al., 2013 has quantitatively determined the mass contribution for each carbonaceous particle classes. Inclusion of this in the introduction and discussion would be necessary for completeness. (References Healy, R.M., Sciare, J., Poulain, L., Crippa, M., Wiedensohler, A., Prevot, A.S.H., Baltensperger, U., Sarda-Esteve, R., McGuire, M.L., Jeong, C.H., McGillicuddy, E., O'Connor, I.P., Sodeau, J.R., Evans, G.J., Wenger, J.C., 2013. Quantitative determination of carbonaceous particle mixing state in Paris using single-particle mass spectrometer and aerosol mass spectrometer measurements. Atmospheric Chemistry And Physics 13, 9479-9496.)

*R2: We have added this reference in introduction, method and discussion sections of the revised manuscript.*

3. Lines 57-60: "This provides different sources for the non-refractory species measured by AMS and indicates different sources of aerosol" might be not appropriate. I think a major part of non-refractory species measured by AMS should be secondary.

*R3: Indeed, major fractions of non-refractory species measured by AMS are secondary. Our points in that sentence are:(1) since the external mixing varied, namely, the dominating particle types varied, the sources for the non-refractory species might be different. The sources herein are stand for the particles containing non-refractory species; (2) varying mixing state also indicates different sources of aerosol particles. The sources in this case are the origins of particles.*

4. Fig. 2: it is possible to compare the mass concentration of AMS and LAAPTOF in different size range? From Fig. 1, it can be seen significant difference of ODE in difference size range? A comparison of AMS and LAAPTOF in different size range might help reduce the difference.

*R4: The ODE for LAAPTOF is significantly size-dependent. However, the LAAPTOF results were corrected by using a size-dependent ODE. Both instruments are equipped with a similar $PM_{2.5}$ aerodynamic lens, which allows focusing the particles between 70 and 2500 nm vacuum aerodynamic diameter for both. However, LAAPTOF can only detect particles by light scattering, which are larger than about 200 nm mobility equivalent diameter (cf. Fig. 1). Therefore, the discrepancy between their results could only be due to particles in this size range, which typically has only a minor influence on the mass concentrations of the refractory components. AMS and LAAPTOF results differ mainly for periods when sulfate and organic show high concentrations, due to the fact that the LAAPTOF is not sensitive to some sulfate salts, e.g., pure ammonium sulfate, and pure organic species. The overall difference can be reduced by using chemically-resolved effective densities and ODEs (please refer to R7).*

5. Section 2.2 line 10-15. I do not understand why "A direct class-dependent quantification of particle mass is therefore not possible." Is it possible that the threshold values set for the positive and negative spectra correlation influence the assignment of individual particles to difficult particle types?

*R5: (1) The classification method embedded in the Igor LAAPTOF data analysis software is Fuzzy c-means clustering, which*
*allows a single particle to belong to multiple classes (Reitz et al., 2016). Fractions of individual particles are assigned to*
*different classes, according to the similarity (for example, if we set the number of classes to two, one particle would have x%*
*similarity to class1 and 1-x% similarity to class 2). Such similarity information is only available for the whole data set*
*rather than a single particle. For a single particle, we only have the information of the corresponding measurement time, its*
*$d_{va}$ and the bipolar mass spectra, but no class information. Therefore, we used the fuzzy c-mean resulting representative*
*class spectra as a reference to identify and count the number of single particles belonging to the individual classes.*

*We have revised the sentence to make it clearer for the readers, as follows:*

*"Thus, we can obtain similarity information for the whole data set rather than a single particle. One drawback is that the*
*individual particles are not directly assigned to individual particle classes, which hinders a direct class-dependent*
*quantification of particle mass."*

*(2) The threshold values set for the positive and negative spectra correlation will influence the assignment of individual*
*particles to different particle classes. Therefore, we tuned the thresholds until we obtained a time series of particle counts,*
*which have a good ($\gamma > 0.6$) correlation with fuzzy results (cf. Table S1 and Fig. S1).*

6. Page 4 Line 29 "This leads to an uncertainty of ~100% in particle mass." is it only for sea salt like particles? How about other particle classes?

*R6: We have deleted this sentence and added a more general explanation of the uncertainty associated with particle shape*
*later in **section 2.2**, as follows:*

*"The aforementioned assumptions and the related uncertainties in particle mass are summarised as follows: 1) ambient*
*particles are spherical with a shape factor $\chi=1$. However, several ambient particle types are non-spherical with a shape*
*factor $\chi$ not equal to 1, e.g., $\chi_{NaCl} = 1.02-1.26$ (Wang et al., 2010) and $\chi_{NH_4NO_3} = 0.8$ (Williams et al., 2013). This can cause*
*uncertainties of 26% and 20% for the particle diameter and 100% and 50% for the particle mass of sodium chloride like and*
*ammonium nitrate like particles, respectively. For soot like particles, the shape caused uncertainty could be even larger, due*
*to their aggregate structures. Such an uncertainty is difficult to reduce, since we do not have particle shape information for*
*individual particles. However, using effective densities may at least partially compensate some of the particle shape related*
*uncertainties. 2)..."*

7. Page 4 Lines 35-40: The assumption of single density value for each particle classes might introduce large uncertainty. The author should adapt a possible density range through the previous publications and evaluate the uncertainty for each assumption. This would also help reduce the overall difference between the comparisons with AMS results.

*R7: We think it is best to apply chemically or particle class resolved effective densities as suggested by Referee#2. Apart*
*from that, we have also applied chemically or particle class resolved ODE values. This reduced the overall difference in the*
*comparisons with AMS results. We have updated added more discussions in the main manuscript and supporting*
*information, as follows:*

*Section 2.2 (3$^{rd}$ paragraph)*

*"It should be noted that in some previous studies, the particle shapes were also assumed as spherical and uniform particle*
*densities ranging from ~1.2 to 1.9 g cm$^{-3}$ were applied for total aerosol particle mass quantification (Allen et al., 2006; Allen*
*et al., 2000; Ault et al., 2009; Gemayel et al., 2017; Healy et al., 2013; Healy et al., 2012; Jeong et al., 2011; Wenzel et al.,*
*2003; Zhou et al., 2016). In our study, we have determined an average density of 1.5 ± 0.3 g cm$^{-3}$ for all ambient particles,*
*based on a comparison between $d_{va}$ measured by AMS and $d_m$ measured by SMPS. However, the density for different types of*
*ambient particles varies, especially for fresh ones (Qin et al., 2006). Particle densities varied during the campaign (Fig. S2)*
*and the representative mass spectra of different particle classes indicate chemical inhomogeneity. In order to reduce the*
*uncertainty induced by the assumption of a uniform density, we assigned specific effective densities (derived from $d_{va}/d_m$)*
*from literature data to each particle class. A density of 2.2 g cm$^{-3}$ was used for calcium nitrate rich particles (Zelenyuk et al.,*
*2005), 1.25 g cm$^{-3}$ for aged soot rich in ECOC-sulfate (Moffet et al., 2008; Spencer et al., 2007) , 2.1 g cm$^{-3}$ for sodium salts*
*(Moffet et al., 2008; Zelenyuk et al., 2005), 1.7 g cm$^{-3}$ for secondary inorganic rich particles (Zelenyuk et al., 2005; Zelenyuk*
*et al., 2008), 2.0 g cm$^{-3}$ for aged biomass burning particles (Moffet et al., 2008), 2.6 g cm$^{-3}$ for dust like particles (Bergametti*

*and Forêt, 2014; Hill et al., 2016). These densities were used for the individual particles of each class without size dependence. Similar chemically-resolved densities have also been used in some previous studies (Gunsch et al., 2018; May et al., 2018; Qin et al., 2006; Qin et al., 2012)."*

*Supporting Information*

[Figure]

*Figure S2: Time series of effective densities derived from comparison between AMS-$d_{va}$ and SMPS-$d_m$.*

8. Page 6 Line 25 I think it would be better to include some references for the identification of amines.

*R8: We have added some references to support our identification of amines. The sentence has been revised in **section 3.1** to:*

*"In addition, it features marker peaks for amines at m/z 58 $C_2H_5NHCH_2^+$, 59 $(CH_3)_3N^+$, 86 $(C_2H_5)_2NCH_2^+$, 88*
10 *$(C_2H_5)_2NO/C_3H_6NO_2^+$, 118 $(C_2H_5)_2NCH_2^+$, which were also identified by SPMS in the other field and lab studies (Angelino et al., 2001; Köllner et al., 2017; Lin et al., 2017; Roth et al., 2016; Schmidt et al., 2017)."*

**Referee #2 comments:** Shen et al describe single-particle mass spectrometry (SPMS) data analysis using LAAPTOF data from a summer 2016 field campaign in rural Germany. While the SPMS data itself appears sound, there are many major technical issues with their analyses, as well as their assertions of originality. Unfortunately, the authors appear to be unaware of the majority of the SPMS literature, which their work would highly benefit from. Please see below for description of major issues, with references to previous literature that I hope will be useful for the authors to place their current work in context and aid in their data analysis and interpretation. I encourage the authors to rethink the framing of their manuscript and, instead of focusing on data analysis methods, consider the science that can be learned from their data itself by examining particle composition as a function of time and meteorological conditions, for example.

*R: We admit that we missed to cite and discuss several relevant publications and we really appreciate the constructive comments by reviewer #2 pointing to the weaknesses of our manuscript and showing ways for improvement. After carefully considering the reviewer's suggestion to shift the scope of the manuscript from the analysis method to the scientific application, we decided to improve the current manuscript highlighting its original points, which we still consider valuable not only to the LAAPTOF user community. In particular we:*

*1) Removed subjective statements throughout the manuscript*
*2) Demonstrated the stability of the LAAPTOF overall detection efficiency (ODE) during field deployment*
*3) Determined ODE for more particle classes allowing now chemically or particle class resolved correction for ODE values*
*4) Discussed the differences between our quantification method and those in previous SPMS studies*

*Please see the detailed revisions in our replies to the specific comments below.*

1. Method development to obtain mass concentrations from SPMS data was previously shown through method development papers by Allen et al. (2000, Environ. Sci. Technol.,"Particle detection efficiencies of aerosol time of flight mass spectrometers under ambient sampling conditions"), Fergenson et al. (2001, Analytical Chemistry, "Quantification of ATOFMS data by multivariate methods"), Wenzel et al. (2003, J. Geophys. Res., "Aerosol time-of-flight mass spectrometry during the Atlanta Supersite Experiment:2. Scaling procedures", Zhao et al. (2005, Analytica Chimica Acta, "Predicting bulk ambient aerosol compositions from ATOFMS data with ART-2a and multivariate analysis"), Allen et al. (2006, Aerosol Sci. Technol., "Instrument busy time and mass measurement using aerosol time-of-flight mass spectrometry"), Bein et al. (2006, Atmos. Environ., "Identification of sources of atmospheric PM at the Pittsburgh Supersite– Part II: Quantitative comparisons of single-particle, particle number, and particle mass measurements"), and Qin et al. (2006, Analytical Chemistry, "Comparison of two methods for obtaining quantitative mass concentrations from aerosol time-of-flight mass spectrometry measurements". The authors "new" method is very similar to the work discussed these older papers, yet these papers are not even cited in the current paper. Many subsequent SPMS papers have used these approaches to provide chemically-resolved mass concentrations: Bhave et al. (2001, Environ. Sci. Technol.), Ault et al. (2009, Environ. Sci. Technol.), Qin et al. (2012, Atmos. Environ.), Healy et al. (2012, Atmos. Chem. Phys.), Healy et al. (2013, Atmos. Chem. Phys.), Gunsch et al. (2018, Atmos. Chem. Phys.), and May et al. (2018, Environ. Sci. Technol. Lett.). I highly suggest that the authors review these previous papers to decide how to move forward with their own work, placing it into the context of previous studies.

*R1: Our ODE values are based on laboratory measurements of reference particles, while most previous studies determined their sensitivities by comparison of single particle data to data from reference instruments, both obtained in the field. The field-based scaling approaches (field-based ODE) allows converting particle number to mass and shows good agreement with the reference instrument and other independent quantitative aerosol particle measurements as well. However, field-based ODE relies on the availability of a reference instrument and their corrections are often class independent. Our approach uses particle class dependent ODE values and doses not rely on the availability of a reference instrument in the field, which is a strength of our laboratory-based method. Our approach aims to determine to total particle mass for the different particle classes but is not intended to determine mass concentrations for specific particle compounds like sulfate or nitrate. However, we also studied special ion intensities or their ratios compared to AMS mass concentrations and found useful correlations especially for the fraction of org/(org+nitrate). This will be applied for source apportionment in an upcoming publication.*

*We included most of the recommended publications in the introduction and method sections, and discussed them as shown below:*

*Section 1. Introduction (4th paragraph)*

[revised manuscript text omitted]

**Section 3.3 (1st paragraph)**

*"Considering the different capabilities of LAAPTOF and AMS, we did not apply the relative sensitivity factors (RSF) method (Healy et al., 2013; Jeong et al., 2011). We analysed ourOur LAAPTOF and AMS data independently and compared them thereafter. For LAAPTOF data, we used relative ion intensities (each ion peak intensity is normalised to the sum of all or selected ion signals. Positive and negative ions were analysed separately), similar to the relative peak area (RPA) method suggested by Hatch et al. (2014)."*

2. The authors assert in the abstract (Page 1, Lines 30-31) that "[their] approach allows for the first time to assign the non-refractory compounds measured by AMS to different particle classes." Similarly, in the conclusions section, it is stated "our study…opens a new way for quantitative information of single particle data, and together with the complimentary results from bulk measurements by AMS we have shown how a better understanding of the internal and external mixing state of ambient aerosol particles can be achieved." These statements are not accurate, as many SPMS analyses have incorporated bulk aerosol composition data (both off-line impactor and online AMS): Bhave et al (2002, Environ. Sci. Technol.), Middlebrook et al. (2003, J. Geophys. Res.), Spencer & Prather (2006, Aerosol Sci. Technol.), Ferge et al. (2006, Environ. Sci. Technol.), Drewnick et al. (2008, Atmos. Environ.), Dall'Osto et al (2009, Atmos. Chem. Phys.), Pratt et al. (2010, J. Atmos. Sci.), Pratt et al. (2010, J. Geophys. Res.), Pratt et al. (2011, Atmos. Chem. Phys.), Decesari et al. (2011, J. Geophys. Res.), Dall'Osto & Harrison (2012, Atmos. Chem. Phys.), Dall'Osto et al. (2012, Aerosol Sci. Technol.), Dall'Osto et al. (2012, J. Geophys. Res.), Dall'Osto et al. (2013, Atmos. Chem. Phys.), Healy et al. (2013, Atmos. Chem. Phys.), Decesari et al. (2014, Atmos. Chem. Phys.), Gunsch et al. (2018, Atmos. Chem. Phys.), and others. In fact the title of Healy et al. (2013, Atmos. Chem. Phys.) is "Quantitative determination of carbonaceous particle mixing state in Paris using single-particle mass spectrometer and aerosol mass spectrometer measurements". Again, these papers are not cited in the current work and should be considered in their data interpretation and discussion.

*R2: We included the previous publications in different sections of the manuscript and pointed out how our approach differs and is valuable not only to the LAAPTOF user community.*

**Section 1. Introduction (4th paragraph):** *Please refer to our answer (R1) to previous comments*

**Section 1. Introduction (5th paragraph)**

*"Many previous studies have also compared single particle classes and bulk species (Dall'Osto et al., 2012; Dall'Osto and Harrison, 2012; Dall'Osto et al., 2009; Dall'Osto et al., 2013; Decesari et al., 2014; Decesari et al., 2011; Drewnick et al., 2008; Gunsch et al., 2018; Pratt et al., 2010; Pratt et al., 2011; Pratt and Prather, 2012). Some studies compared ion intensities from single particle data (Bhave et al., 2002) or specific ion ratios, such as nitrate/sulfate (Middlebrook et al., 2003), OC/EC (Spencer and Prather, 2006), and EC/(EC+OC) (Ferge et al., 2006), carbonaceous/(carbonaceous+sulfate) (Murphy et al., 2006) with the other bulk data. Hatch et al. (2014) used m/z 36 $C_3^+$ as a pseudo-internal standard to normalize the secondary inorganic and organic peak areas in organic rich particles, resulting in good correlation with the independent AMS measurements. Similarly, Ahern et al. (2016) used the peak area ratio of organic matter marker at m/z 28 $CO^+$ to EC markers ($C_{2-5}^+$) to account for laser shot-to-shot variability, and demonstrated a linear relationship between normalized organic intensity and secondary organic aerosol (SOA) coating thickness on soot particles. A normalized or relative peak areas (RPAs) method was suggested by Hatch et al. (2014) to account for shot-to-shot variability of laser intensities. Although the LDI matrix effects cannot be completely overcome by the aforementioned method, some examples for good comparisons between single particle and bulk measurements were shown."*

*(2) Indeed, our study is not the first to compare to bulk composition data. One of the most relevant recent work was done by Healy et al. (2013) showing that SPMS-derived mass concentrations of organics, ammonium, nitrate and sulfate are comparable with AMS results. However, they used AMS data to generate particle class independent relative sensitivity factor (RSF) for organics, ammonium, nitrate and sulfate. Such RSF may vary in different particle types, and thus also vary during individual measurement campaigns and for different locations. In our study, SPMS data and AMS data are analysed independently and only compared thereafter. Hence, our method can potentially be applied to deduce mass concentrations also if no AMS data is available. In addition, we found specific relationships of LAAPTOF ion intensities and AMS mass concentrations for non-refractory compounds, especially for the fraction of org/(org+nitrate), which has not been reported in previous studies. We showed the originality of our method in **Section. 1.** (last paragraph). Please refer to our answer (R1) to previous comments.*

*(3) The inaccurate statements have been revised as follows:*

*Abstract*

*"Furthermore, our approach allows assigning the non-refractory compounds measured by AMS to different particle classes."*

*Section 4 (Last paragraph)*

*"In spite of significant uncertainties stemming from several assumptions and instrumental aspects, our study provides a good example for identification and quantitative interpretation of single particle data. Together with the complimentary results from bulk measurements by AMS, we have shown how a better understanding of the internal and external mixing state of ambient aerosol particles can be achieved."*

3 (1): Figure 1, which shows the overall detection efficiencies for various particle types, as determined in the laboratory, is useful. However, apparently these data are all already published in Shen et al. (2018, Atmos. Meas. Technol.), unfortunately limiting the originality here. (2) It would be great if additional particle type proxies, based on those observed in the field could be added (e.g. soot, biomass burning). Considering these data and Figure 2 (dominance of soot from 0.2-0.4 um), I encourage the authors to characterize their detection efficiency of soot particles in the laboratory. (3) Also, the authors apply this laboratory-derived ODE to their field data, but it is not discussed whether the ODE was verified in the field, or how reproducible it is in the field.

*R3 (1): Actually, Figure 1 included additional overall detection efficiency (ODE) data compared to our previous publication (Figure 2 in Shen et al., 2018) such as organics, ammonium sulfate, soil dust, and sea salt particles.*

*(2) Following your suggestion we have added also ODE values in the revised version for additional sizes and particle types, such as soot. Soot particles from incomplete combustion of propane were generated with a propane burner (RSG miniCAST; Jing Ltd.), and then injected into and sampled from a stainless steel cylinder of ~0.2 $m^3$ volume. It turns out that the ODE for soot particles are on the extrapolated mean ODE curve. In addition, we have added ODE values for $SiO_2$ particles (800 and 1200 nm $d_m$) as dust proxy particles. The updated Fig. 1 is shown below:*

[Figure]

*Figure 1: Overall detection efficiency of LAAPTOF for different types of particles as a function of the mobility diameter ($d_m$), adapted from Shen et al. (2018) and extended. Dashed lines are fitting curves for maximum, mean and minimum values of ODE. For other organic particles (green), ODE at 400 nm is the data from secondary organic aerosol (SOA) particles from α-pinene ozonolysis, ODE at 500 nm is the data from humic acid, and ODE at 800 nm is data from humic acid (1.9 ± 0.3%), oxalic acid (0.3 ± 0.1%), pinic acid (1.6 ± 0.1%), and cis-pinonic acid (1.9 ± 0.7%). SOA particles were formed in the Aerosol Preparation and Characterization (APC) chamber and then transferred into the AIDA chamber. Agricultural soil dust (brown symbol) were dispersed by a rotating brush generator and injected via cyclones into the AIDA chamber. Sea salt particles (purple) were also sampled from the AIDA chamber. Soot particles from incomplete combustion of propane were generated with a propane burner (RSG miniCAST; Jing Ltd.), and then injected into and sampled from a stainless steel cylinder of ~0.2 $m^3$ volume. $SiO_2$ particles were directly sampled from the headspace of their reservoirs. The other aerosol particles shown in this figure were generated from a nebulizer and size-selected by a DMA. Note that there is uncertainty with respect to particle size due to the particle generation method. The nebulized and DMA sized samples have relative smaller standard deviation (SD) from Gaussian fitting to the measured particle sizes. PSL size has the smallest size SD (averaged value is 20 nm) and the corresponding relative SD (RSD = SD divided by the corresponding size) is ~6%, since the original samples are with certain sizes. The other nebulized samples have standard deviations ranging from 70 to 120 nm SD and 3 to 23% RSD. Particles sampled from AIDA chamber have much bigger size SD: ~70 nm for SOA (17% RSD), ~100 nm for agricultural soil dust (~83% RSD) and ~180 nm for sea salt particles (~34% RSD). Considering this*

*uncertainty, we have chosen size segment of 100 nm (±50 nm) for correction, e.g., particles with size of 450 to 550 nm will use the ODE at 500 nm particle number correction.*

*(3) During our field measurements we did calibrations of the LAAPTOF with PSL particles of 400, 500, 700, and 800 nm diameter resulting in ODE values with no significant difference compared to the ODE values determined in the laboratory, as shown in the figure below. This finding reflects the good stability of the LAAPTOF performance in the temperature controlled container. Actually, once the LAAPTOF adjustments were optimized after transport no further adjustments were necessary during the 6 weeks of the campaign.*

[Figure]

***Figure:** ODE values measured during the campaign (green markers) compared to values measured in the laboratory.*

*The stability of the LAAPTOF ODE values is mentioned in the revised version as follows:*

***Section 2.2 (2ⁿᵈ last paragraph)***

*"During our field measurements we did calibrations of the LAAPTOF with PSL particles of 400, 500, 700, and 800 nm $d_m$ resulting in ODE values with no significant difference compared to the ODE values determined in the laboratory. This finding reflects the good stability of the LAAPTOF performance in the temperature controlled container. Actually, once the LAAPTOF adjustments were optimized after transport no further adjustments were necessary during the 6 weeks of the campaign."*

3 (4): Furthermore, the authors note that using the mean ODE introduces significant uncertainty; this is also why the previously published methods (e.g. Qin et al 2006) determine the detection efficiencies in the field with time. Also, in presenting this curve to LAAPTOF users, it is important to note that this curve should not be extrapolated to other LAAPTOF or SPMS instruments without a standard to check against (e.g. PSLs). The authors simply note that "alignment and variance in particle-laser interaction lead to uncertainty in ODE" (Page 5, Lines 33-34). This paragraph suggests that this variance is included in the 540% ODE spread for various lab-generated aerosols; however, it should be noted that the ODE dependence on sizing laser and desorption/ionization laser powers and alignments will change with time, especially when the instrument is moved. This is why the previously published methods noted above (e.g. Qin et al. 2006) characterize detection efficiencies in the field. For example, Jeong et al. (2011, Atmos. Chem. Phys.) and Wenzel et al. (2003) show how the hit fraction of particles can change with time during and between field campaigns; this is taken into account in the Wenzel et al. (2003) and Qin et al (2006) method. Also, for consideration of "variance in particle laser interaction" (Page 5, Line 34) the authors may be interested to review Wenzel & Prather (2004, Rapid. Commun. Mass Spec., "Improvements in ion signal reproducibility obtained using a homogeneous laser beam for on-line laser desorption/ionization of single particles").

*R3 (4): Laboratory-based ODE and the field-based ODE have their advantages and disadvantages. The field-based scaling approaches for quantification of single particle have been developed and being upgraded with hit rate correction and composition-dependent density correction, which now allows converting particle number to mass and shows good agreement with the reference instrument (as one may expect). Such that one can quantitatively interpret the mixing state of the ambient aerosol particles by SPMS, which is the strength of field-based scaling/ODE. However, these methods rely on the availability of a reference instrument and their corrections are mainly class independent (e.g. Wentzel et al., 2003; Qin et al., 2006) even though upgraded versions aimed to correct chemical biases by using hit rate thresholds, below which a new missed particle type will be added to the total hit). Our approach uses particle class dependent ODE values and is not relaying on the availability of a reference instrument in the field.*

*As shown above and as discussed by Shen et al., 2018 the ODE of LAAPTOF in the field was very stable over a period of 6 weeks.*

*We agree that it is important to note that the ODE curve applied herein should not be extrapolated to other LAAPTOF or SPMS instruments without a standard to check against e.g. PSL particles. We have added this statement in the revised version.*

*The corresponding content has been revised as follows:*

*Section 2.2 (2ⁿᵈ last paragraph)*

*"2) instrumental aspects such as alignment and variance in particle-laser interaction lead to uncertainty in ODE. They are included in the uncertainties given in Fig. 1 for which repeated measurements after various alignments were used. The fluctuations of particle-laser interactions can be reduced by using a homogeneous laser desorption and ionization beam (Wenzel and Prather, 2004) or delayed ion extraction. (Li et al., 2018; Vera et al., 2005; Wiley and Mclaren, 1955). Note that we used the same sizing laser and desorption/ionization laser pulse energy (4 mJ) in the field as those used for generating ODE, and aligned the instrument in the field with the similar procedures as we did in the lab. During our field measurements we did calibrations of the LAAPTOF with PSL particles of 400, 500, 700, and 800 nm $d_m$ resulting in ODE values with no significant difference compared to the ODE values determined in the laboratory. This finding reflects the good stability of the LAAPTOF performance in the temperature controlled container. Actually, once the LAAPTOF adjustments were optimized after transport no further adjustments were necessary during the 6 weeks of the campaign. Moreover, it is important to note that the ODE curve applied herein should not be extrapolated to other LAAPTOF or SPMS instruments without a standard check against e.g. PSL particles."*

4. Page 5, Lines 3-28: If the authors have chemically-resolved ODEs, why did then choose to apply a mean ODE to all data? If the ODEs are stable relative to a standard (e.g. PSLs), it seems like a strength of the current work that chemically-resolved ODEs, for each particle type, could be applied in the calculation of mass concentrations.

*R4: The reason for choosing a mean ODE was that it is difficult to assign a specific ODE to individual particle classes. Particle classes of ambient aerosol particles are often complex mixtures for which we do not always have the corresponding laboratory reference. However, after measuring several ODE data for more reference particles we determined size and chemically or particle class resolved ODE values. Application of the chemically-resolved ODEs, as well as chemically-resolved effective densities for each class results in a better agreement between LAAPTOF and AMS results as shown in the figure below. However, some discrepancies still remain especially for specific time periods e.g. P3 and P5 with high mass fractions of organics. Although LAAPTOF data shows a good correlation with the AMS data e.g. for period P5, it obviously misses a large mass fraction of most likely smaller organic particles. This may be due to an insufficient representation of this kind of organic rich particles in the particles classes identified initially. Even using reference spectra of organic rich particles it was not possible to indentify a number of those particles sufficient to close this gap.*

[Figure]

*Figure: Time series of total mass concentration measured by AMS and LAAPTOF total mass concentration estimated based on chemically-resolved densities and for different ODE values.*

*We have updated the corresponding figures (Fig. 3, Fig. 5, Fig. S2, and Fig. S3 have been changed to new Fig. 4, Fig. 6, Fig. S5, and Fig. S6, respectively) and added some explanations in the revised version, as follows:*

[Figure]

*Figure 4: Time series of the similarity, corrected number fraction, and mass fraction of seven major particle classes and the corresponding pie charts for total fractions. Note that, the correction shown here is based on a chemically or particle class resolved ODE. The seven classes are class 1 "Calcium-Soil"; class 2 "Aged soot"; class3: "Sodium salts"; class 4 "Secondary inorganics-Amine"; class 5 "Biomass burning-Soil"; class 6 "Biomass burning-Organosulfate"; and class 7 "Mixed/aged-Dust".*

[Figure]

***Figure 6:*** *Time series of (a) total mass ratio of LAAPTOF to AMS data, (b) LAAPTOF total mass and AMS total mass (c) mass concentrations of organic, sulfate, nitrate, and ammonium compounds measured by AMS. In panel (b) r is the Pearson's correlation coefficient between LAAPTOF and AMS results. P1 is Period 1 from 7/26/2016 16:23 to 8/1/2016 11:43; P2 from 8/2/2016 09:43 to 8/14/2016 17:53; P3 from 8/14/2016 18:03 to 8/17/2016 21:03; P4 from 8/17/2016 21:13 to 8/23/2016 15:33; P5 from 8/24/2016 15:03 to 8/29/2016 08:33; P6 from 8/29/2016 08:43 to 8/31/2016 09:13. Zoom in figures for P1, 2, 4, and 5 can be found in Fig. S5, as well as the corresponding scatter plots for LAAPTOF and AMS data comparison.*

[Figure]

**Figure S5:** *Comparison of mass concentration results between LAAPTOF and AMS in four periods. r represents for Pearson's correlation coefficient. Period 1 is from 7/26/2016 16:23 to 8/1/2016 11:43; P2 from 8/2/2016 09:43 to 8/14/2016 17:53; P4 from 8/17/2016 21:13:00 to 8/23/2016 15:33; P5 from 8/24/2016 15:03 to 8/29/2016 08:33.*

[Figure]

**Figure S6:** *Chemical resolved size distributions for the particles measured by AMS during organics rich period (P5).*

*Section.2.2 (above equation 4)*

*"Therefore, we used reference particle ODE values to estimate the size dependent ODE values for the particle classes observed in the field as follows. ODE values for ammonium nitrate and sodium chloride were used to fit ODE curves for secondary inorganic rich and sodium salt like particles, respectively. The mean ODE values from all reference particles was used for the class of aged soot particles since it showed best agreement with the reference soot particles (cf. Fig. 1). The minimum ODE curve from all reference particles was used for all dust like particle classes."*

*Section.3.2 (2ⁿᵈ and 3ʳᵈ paragraphs)*

*"It turns out that the total mass of the particles measured by LAAPTOF is 7±3% (with maximum ODE), 16±6% (mean ODE), 60±24% (minimum ODE) and 45±16% (23–68% with chemically-resolved ODE) of the total AMS mass depending on the measurement periods. Despite of this relative large differences in the average mass concentrations of LAAPTOF and AMS they show much better agreement in total mass and also good correlations during specific periods (P), such as P1, 2, 4, and 5 (cf. Fig 6 and Fig. S5), covering ~85% of the measurement time. Hence, the large differences in the average mass concentrations are caused by larger deviations during some relatively short periods or events. Considering that AMS can only measure non-refractory compounds, the good correlation between AMS and LAAPTOF gives us a hint that the species measured by AMS may mainly originate from the particles of complex mixtures of both refractory and non-refractory species. It is worth noting that weakest correlation (γ=-0.1) is observed in P6 when LAAPTOF measured the high fraction of sodium salts particles (especially on August 29ᵗʰ). Specifically, from 9:00 to 23:53 on August 29ᵗʰ, LAAPTOF and AMS tended to be slightly anti-correlated (γ=-0.3), due to a burst of sodium chloride rich particles, which are refractory and thus AMS is unable to measure. Sodium chloride is a possible sub-class of sodium salts particles and will be discussed in a separate study.*

*As shown in Fig. 6 (a), the mass ratio of LAAPTOF to AMS has its lower values during lower value in P3 and P5 when the AMS organic mass concentration is higher than in most of the other periods. Although LAAPTOF data shows a good correlation with the AMS data e.g. for period P5, it obviously misses a large mass fraction of most likely smaller organic particles. The corresponding chemically-resolved size distributions of particles measured by AMS are given in Fig. S6. This may be due to an insufficient representation of this kind of organic rich particles in the particles classes identified initially. Even using reference spectra of organic particles it was not possible to identify a number of those particles sufficient to close this gap. In addition, during the whole campaign the sulfate mass fraction measured by AMS is largest in P3 (cf. Fig. 6c). However, the LAAPTOF is not sensitive to some sulfate salts, e.g., pure ammonium sulfate (Shen et al., 2018), thus it is likely that such particles were dominating in P3, which resulted in a weaker correlation between these two instruments. Relatively pure ammonium sulfate was also suggested to be a "missing" particle type in the other SPMS field studies (Erisman et al., 2001; Stolzenburg and Hering, 2000; Wenzel et al., 2003) and (Thomson et al., 1997) showed in a laboratory study that pure ammonium sulfate particles were difficult to measure using LDI at various wavelengths."*

5. The authors make many subjective statements that can be refuted by previously published literature, such as those mentioned above (and others not listed here). I caution the authors from making such statements. For example, "SPMS is a useful, albeit not fully quantitative tool" (Page 1, Line 14) and "SPMS data analysis has been proven difficult under real world conditions" (Page 2, Lines 2-3). The authors state that "mass spectroscopic signatures do not necessarily reflect the primary composition of the particles" (page 2, lines 7-8). However, many previous papers (e.g., Bhave et al. 2001, Environ. Sci. Technol., Reinard et al., 2007, Atmos. Environ., Toner et al. 2008, Atmos. Environ., Pratt & Prather 2009, Environ. Sci. Technol., Healy et al. 2010, Atmos. Chem. Phys., and others) have examined the SPMS source signatures of primary particles and apportioned ambient particles according to these signatures, which is arguably one of the strengths of SPMS.

*R5: We agree with your comment and tried to remove or rephrase all rather subjective statements, e.g. as follows:*

*"SPMS is a useful, albeit not fully quantitative tool" has been changed to "SPMS is a widely used tool"*

*"SPMS data analysis has been proven difficult under real world conditions." has been changed to "there are still challenging issues related to large amounts of SPMS data analysis."*

*"Mass spectroscopic signatures do not necessarily reflect the primary composition of the particles" has been changed to "some mass spectroscopic signature peaks do not necessarily reflect the primary composition of the particles."*

*We agree with you about the strengths of SPMS for apportionment of the ambient particles according to the spectroscopic signatures. However, there are still some signatures, which cannot be well distinguished. For example, potassium and organics can both contribute to m/z 39. As mentioned in a resent SPMS study (Christopoulos et al., 2018), different primary*

*aerosol particles may have similar marker peaks, e.g., fly ash, mineral dust, and biological aerosol particles can all have phosphate makers (Zawadowicz et al., 2017) Therefore, many studies used specific ratios to refine the signatures, as well as reference spectra in our previous work.*

6. Page 3, Line 36-37: The authors state that the LAAPTOF has a size range of 70 to 2500 nm; however, Figure 1 shows that the detection efficiencies of particles <400 nm and >1200 nm is extremely low (<1%), making this earlier statement seem misleading.

*R6: We agree that this was misleading. Therefore, we have changed the corresponding section as follows:*

*"In brief, aerosols are sampled with a flowrate of ~80 $cm^3$ $min^{-1}$ via an aerodynamic lens, focusing and accelerating particles in a size range between 70 nm and 2500 nm $d_{va}$. Afterwards, they pass through the detection chamber with two diode laser beams ($\lambda$ = 405 nm). Particles smaller than 200 nm and larger than 2 $\mu m$ are difficult to detect, due to weak light scattering by the smaller particles and due to a larger particle beam divergence for the larger particles."*

7. Section 2.2: The authors apply particle densities to each particle class and assume that all particles are spherical. Instead the authors could consider applying measured individual particle effective densities (reducing assumptions) (e.g. Zelenyuk et al. 2008, Analytical Chem., Spencer et al. 2007, Environ. Sci. Technol., Zhang et al. 2016, Atmos. Chem. Phys.), as has been done in other SPMS studies converting to mass concentrations (e.g. Qin et al. (2012, Atmos. Environ.), Gunsch et al. (2018, Atmos. Chem. Phys.), and May et al. (2018, Environ. Sci. Technol. Lett.).

*R7: We have applied effective densities from the literature and explained this in the method section, as follows:*

*Section 2.2 (3rd paragraph)*

*"In order to reduce the uncertainty induced by the assumption of a uniform density, we assigned specific effective densities (derived from $d_{va}/d_m$) from literature data to each particle class. A density of 2.2 g $cm^{-3}$ was used for calcium nitrate rich particles (Zelenyuk et al., 2005), 1.25 g $cm^{-3}$ for aged soot rich in ECOC-sulfate (Moffet et al., 2008; Spencer et al., 2007) , 2.1 g $cm^{-3}$ for sodium salts (Moffet et al., 2008; Zelenyuk et al., 2005), 1.7 g $cm^{-3}$ for secondary inorganic rich particles (Zelenyuk et al., 2005; Zelenyuk et al., 2008), 2.0 g $cm^{-3}$ for aged biomass burning particles (Moffet et al., 2008), 2.6 g $cm^{-3}$ for dust like particles (Bergametti and Forêt, 2014; Hill et al., 2016). These densities were used for the individual particles of each class without size dependence. Similar chemically-resolved densities have also been used in some previous studies (Gunsch et al., 2018; May et al., 2018; Qin et al., 2006; Qin et al., 2012)."*

8. Section 3.1: (1) This section does not provide any new information in terms of methods/technology; this is simply a description of mass spectra particle types observed during the field study. I also have several concerns about particle type identification, mostly with the respect to the attribution of nearly all particles as "dust-like", which does not have support by the m/z marker ions shown and described, as I note below. In fact, mass spectral markers supporting dust are only shown (in Fig 2) and discussed here for Class 1 (5%, by number) and Class 7 (4.6%, by number), leaving >90% of the particle as non-dust particles. Also, it would be useful to move Figure 4 and its discussion to this section, as that figure is useful and aids with particle type classification.

*R8 (1): Considering your suggestions we have revised the particle class labels as listed in the table below:*

*Table 1: Particle class numbers, names, and labels.*

| Class No. | Name | Label |
|---|---|---|
| 1 | Calcium rich and soil dust like particles | Calcium-Soil |
| 2 | Aged soot like particles | Aged soot |
| 3 | Sodium salts like particles | Sodium salts |
| 4 | Secondary inorganics rich and amine containing particles | Secondary inorganics-Amine |
| 5 | Aged biomass burning and soil dust like particles | Biomass burning-Soil |
| 6 | Aged biomass burning and organosulfate containing particles | Biomass burning-Organosulfate |
| 7 | Mixed/aged and dust like particles | Mixed/aged-Dust |

*In this section, we did use Fig 4 (Correlation diagram of fuzzy representative spectra and 36 laboratory-based reference spectra; new Fig 5 in the revised version) to discuss particle type identification (page 6 line10 and page 7 line 15 in the manuscript submitted. For example, among the 7 classes, we attributed class 1, 5, 6 and 7 as "dust-like", based on the correlation diagram (cf. new Fig.5). We cannot rule out especially the soil dust contributions although there are not obvious m/z marker ions showing. The weaker signal may be caused by a core-shell structure of the particles. In fact, previous studies identified soil dust as the major particle type dominating the coarse particles sampled in the same region of the upper Rhine valley (Faude and Goschnick, 1997; Goschnick et al., 1994). Goschnick et al. (1994) found a core-shell structure in both submicron and coarse particles collected North of Karlsruhe city.*

8 (2). Particle Class 4: The authors call these "Secondary inorganic and amine like particles" and discuss secondary markers on lines 24-25 (page 6) and larger size (0.5-1 um) (line 27). Yet, the next sentences (lines 26-27) states "class 4 is relatively "clean" with the fewest peaks, indicating that the particles might be relatively fresh." There is no mass spectral support for these as freshly emitted particles; in fact the authors say "secondary" in the naming of the particle class. This discrepancy must be fixed.

*R8 (2): Indeed, this was misleading. We think these particles are rather young secondary particles, formed not very long ago, as they obviously had no time to uptake other species. We have changed this in the revised manuscript, as follows:*

*"Among all the representative mass spectra for seven particle classes, class 4 is relatively 'clean' with the fewest peaks (cf. Fig. 2 and Fig. S3), indicating that these particles did not have had the time to uptake other components. Hence, most likely they were formed not very long ago by conversion of their precursors."*

8 (3). Particle Class 5: The authors call these "Potassium rich and aromatics coated dust like particles", with m/z 39 (K+), aromatic marker peaks, and m/z 213 (K3SO4+). The authors note that these "particles might originate from biomass burning". No mass spectral support is provided for identification as dust. Based on other SPMS literature of biomass burning studies, I believe this particle class should be labeled as "Biomass Burning".

Particle Class 6: The authors call these "Organosulfate coated dust-like particles", with organosulfate marker ions. Again, no mass spectral marker ions are discussed to support identification as dust. Also, these particles seem very similar to Class 5 (with large K+, m/z 213, etc); could they correspond to more aged biomass burning particles?

*R8 (3): As mentioned above, the reason for naming them as dust like particles were based on the correlation diagram, showing good correlation between them and the dust particles, especially for class 5. However, we agree that it is more reasonable to assign them to biomass burning particles, due to the strong maker ions of potassium mixed with sulfate. Given the other features, such as strong nitrate marker and the good correlation with soil dust for class 5, and organosulfate markers for class 6, we have labelled both of them as aged biomass burning particles. We have changed their labels to "Biomass burning-Soil" and "Biomass burning-Organosulfate" and discussed this in the revised manuscript, as follows:*

*Section 3.1 (4th paragraph)*

*"Note that we also attributed this class as soil dust like based on the correlation diagram (Fig. 5), although there are no obvious marker ions visible. It is correlated well (γ≥0.6) with reference spectra of dust particles, especially agricultural soil dust. The weak spectral signal might due to a core-shell structure of the particles. In fact, previous studies identified soil dust as the particle type dominating the coarse particles sampled in the same region (Faude and Goschnick, 1997; Goschnick et al., 1994). Goschnick et al. (1994) found a core-shell structure in both submicron and coarse particles collected north of the Karlsruhe city of Karlsruhe in the upper Rhine valley. This supports our hypothesis. In addition, similar as class 3, class 5 also has two modes in its size distribution centred at about 500 and 800 nm $d_{va}$. Such potential sub-classes will be further analysed in the future."*

8 (4). Page 5, Line 41: m/z 24 (C2-) is attributed here to organics, which is possible for 193 nm at high laser pulse energy (e.g., Zelenyuk et al. 2009, Int. J. Mass Spec.); however, it is also a common elemental carbon marker peak (e.g., Zelenyuk et al. 2017, Int. J. Engine Res., Spencer et al. 2006, Aerosol Sci. Technol.).

*R8 (4): Yes, you are right. We have pointed this out in the revised version, as follows:*

*"Besides, m/z 24 $C_2^-$ could also be related to elemental carbon (EC). In this case, m/z 24⁻ should actually show a higher intensity than m/z 26⁻, and further EC markers ($C_n^\pm$) should show up as well."*

9. Page 6, Lines 6-8: I am quite confused by this statement. Does this mean that fuzzy classification does not separate individual mass spectra into individual clusters? Or, if this isn't the case, why are the authors using "similarity" to estimate the number fraction of particles in each group? Why not simply count the number of mass spectra in a given group and then

divide by the total number of particles sampled? Please clarify. Neutral networking algorithms used previously in SPMS (e.g. Rebotier & Prather 2007, Analytical Chimica Acta) separate individual mass spectra (corresponding to individual particles) into separate clusters such that it is simple to calculate number fractions.

*R9: Actually, fuzzy c-means classification does not separate individual mass spectra into individual clusters. Instead, it classifies the spectra according to their similarities, allowing one spectrum (particle) to belong to different particle classes! This is explained in the method section. In the revised manuscript, we have added one paragraph in the introduction section to clarify this classification method, as follows:*

*Section 1 (2$^{rd}$ paragraph)*

*"Particle type identification, i.e., the assignment of every detected particle to one out of a set of particle types, which are either predefined or deduced from the experimental data, is perhaps one of the most critical issues. Different data classification methods, e.g., fuzzy k-means clustering algorithm, fuzzy c-means (modification of k-means), ART-2a neural network, hierarchical clustering algorithms, and machine learning algorithms are applied to reduce the complexity and highlight the core information of mass spectrometric data (Reitz, et al., 2016; Christopoulos et al., 2018). Reitz et al. (2016) reviewed commonly used data classification methods in SPMS studies and pointed out the advantage of the fuzzy c-means clustering approach, which allows individual particle to belong to different particle classes according to spectral similarities. One recent classification approach applied machine learning algorithms and successfully distinguished SOA, mineral and soil dust, as well as biological aerosols based on a known a priori data set (Christopoulos et al., 2018). In this study we used the fuzzy c-means clustering approach which is embedded in the data analysis Igor software for our laser ablation aerosol particle time-of-flight mass spectrometer (LAAPTOF, AeroMegt GmbH). Based on the data classification, averaged or representative mass spectra of different particle classes can be obtained."*

*Furthermore we have added the following sentence to **section 2.2**:*

*"Thus, we can obtain similarity information for the whole data set rather than a single particle. One drawback is that the individual particles are not directly assigned to individual particle classes, which hinders a direct class-dependent quantification of particle mass."*

10. Figure 2: I suggest raising the cut-off intensity for the mass spectra peak areas, as there is significant noise shown in all mass spectra currently. Some labeled peaks also do not appear to be above the limits of detection; please check. Also, it would be useful to add the particle type names, in addition to the numbers, to the labeling of the mass spectra. Please also clarify what is meant by "background fragments that exist for every particle class"; do the authors mean common ions, or do they mean that there is a chemical background somehow in the mass spectrometer?

*R10: We have raised the cut-off intensity and removed some labelled peaks not above the detection limit, and added the particle type names. The background fragments are the common ions observed in every particle class. We have updated Figure 2 and clarified this in the caption.*

11. Figure 3: (1) Given the large number fraction of EC particles (class 2) in Figure 2, why is this not reflected in Figure 3, especially since the particle number concentration mode should be at less than 0.2 um? I'm concerned that there could be a problem here in the application of the ODE to the "corrected number fractions" shown here. (2) Also, since the authors have taken the time to convert to number and mass concentrations, it would be useful to show to chemically-resolved number and mass concentration time series plots. (3) It would also be useful here and throughout to refer to particle classes chemically (e.g. dust, EC, biomass burning, organic carbon-sulfate) rather than numbers that require the reader to refer regularly back to Section 3.1.

*R11: (1) Although the number fraction of aged soot particles (class 2) is large in the smaller size range between 200-400 nm d$_{va}$ their contribution to the total number counts is only 4.3% (Figure 3a) or 7.3% (Figure 3b) after correction of the number fraction. As shown in the figure below, the total number of particles counted for particle sizes below 500 nm is much smaller than that for particles with sizes above 500 nm diameter. Therefore, the number fraction of the aged soot particles which dominate the small particles is only a minor fraction of the total number of all particles which is dominated by the larger particles. To illustrate this we have made a new Fig. 3 for **section 3.1** combining the size dependent number fractions and the total number counts measured during the campaign.*

[Figure]

*Figure 3: (a) Size resolved number fraction for seven particle classes measured during the field campaign TRAM01, based on fuzzy classification according to fuzzy c-means clustering algorithm. (b) Overall size distribution for the particles measured by LAAPTOF during the whole campaign.*

*Furthermore, we have added the following text to **section 3.2**:*

*"Please note that the aged soot particles (class 2), which dominate the number fraction for particles below 400 nm in the fuzzy c-means analysis comprise only a minor fraction of the total number counts in Figure 4 because the total particle number is dominated by particles larger than 500 nm (cf. Figure 3b)."*

*(2) We have added chemically-resolved number and mass concentration time series in the revised supporting information as Fig. S4, as follows:*

[Figure]

*Figure S4: Time series of the particle number, corrected number, and mass concentration of seven major particle classes and the corresponding pie charts for total fractions. 7 fuzzy classes are class 1 "Calcium-Soil"; class 2 "Aged soot"; class3: "Sodium salts"; class 4 "Secondary inorganics-Amine"; class 5 "Biomass-Soil"; class 6 "Biomass-Organosulfate"; and class 7: "Mixed/aged-Dust". This figure is similar as Fig. 3, except with the absolute values. Another panel (b2) was added in order to better visualize the time series of class 3 and 4, since their number fraction is small after correction.*

*(3) Wherever space allowed we have replaced particle class numbers by their names or labels.*

12. Page 7, Lines 26-27: The authors state here "…there is no well-defined relationship between spectral signal and quantity." I disagree with this statement, as many SPMS (and LDI generally) papers have investigated this relationship, which is governed by ionization energies of species. Based on the statements on Page 2, Lines 20-26, I'm concerned that the authors

may have some confusion about LDI, which is known to primarily result in neutral (rather than ion) formation. I encourage the authors to read textbook or review literature on MALDI and LDI (e.g., Zenobi & Knochenmuss 1998, Mass Spec. Rev., "Ion Formation in MALDI Mass Spectrometry"). For example, in the positive ions, typically the largest ions correspond to those with the lowest ionization energies (of those in the sample). SPMS data analysis methods to account for LDI matrix effects are discussed by Hatch et al (2014, Aerosol Sci. Technol.). The authors do mention in the introduction that SPMS relative sensitivity factors that account for differences in ionization energies are discussed by Gross et al. (2000, Analytical Chem.). Woods et al. (2001, Analytical Chem., "Quantitative detection of aromatic compounds in single aerosol particle mass spectrometry") is another reference. Thomson et al. (1997, Aerosol Sci. Technol., "Thresholds for laser-induced ion formation from aerosols in a vacuum using ultraviolet and vacuum-ultraviolet laser wavelengths"), Thomson & Murphy (1993, Applied Optics, "Laser-induced ion formation thresholds of aerosol particles in a vacuum"), and Reinard & Johnson (2007, J. Am. Soc. Mass Spec.) will likely also be useful to the authors. Other papers discussing relationships between species quantities and SPMS ion signals are provided above in the comments about SMPS quantification and comparisons with bulk measurements.

*R12: (1) Indeed, several SPMS papers (mainly ATOFMS) have investigated the relationship between spectral signal and quantity, such as the examples you mentioned. However, we don't think that relationships between spectral signal and quantity are well-defined especially considering field observations. What can be achieve is maybe demonstrated best by Gross et al., 2000. For LAAPTOF no systematic work to determine this relationship has been done yet. In fact, our statement you pointed out only refers to the LAAPTOF instrument. We have clarified this in the revised version, as follows:*

**Section 3.1 (2ⁿᵈ last paragraph)**

*"We emphasize here that the expression "rich" as used in this study only indicates a strong signal in the mass-spectra rather than a large fraction in mass, since there is no well-defined relationship between LAAPTOF spectral signal and the corresponding quantity. The sensitivities of this instrument to different species have to be established in the future."*

*(2) We revised our statement on Page 2, Lines 20-26 (former version), in order to make it more clearly:*

**Section 1 (4ᵗʰ paragraph)**

*"...because laser ablation only allows an a priori unknown fraction (neutral species) of the single particle to be vaporized/desorbed and then ionized (Murphy, 2007; Reinard and Johnston, 2008). In addition, matrix effects may obscure the particle composition (Gemayel et al., 2017; Gross et al., 2000; Hatch et al., 2014)."*

13. Sections 3.2 & 3.3: My comments about this section are primarily summarized above in my notes about previous work producing mass concentrations from SPMS data. I want to add here two additional comments.

(1) Given the differences in typical detection efficiencies between the LAAPTOF and AMS, one would not expect good correlations without examining only the size range of overlap, which I would encourage the authors to do. Figure 5 should be revised accordingly. (2) I also encourage the authors to look at previous SPMS-AMS comparisons (see comments above) for greater interpretation of their results and also for additional ways to conduct this analysis that have previously been successful. (3) For individual ions (Page 8, Lines 22-34), I suggest the authors look at Hatch et al. (2014, Aerosol Sci. Technol.) and Healy et al (2013, Atmos. Chem. Phys., "Quantitative determination of carbonaceous particle mixing state in Paris using single-particle mass spectrometer and aerosol mass spectrometer measurements"). (4) The authors note that sulfate salts may have been missed during P3. As such I suggest the authors consider the work of Wenzel et al. (2003, J. Geophys. Res.), who developed a method to identify and quantify "missed" particles, consistent with pure ammonium sulfate.

*R13: (1) Both instruments are equipped with a similar PM$_{2.5}$ aerodynamic lens, which allows focusing the particles between 70 and 2500 nm vacuum aerodynamic diameter for both. However, LAAPTOF can only detect particles by light scattering, which are larger than about 200 nm mobility equivalent diameter (cf. Fig. 1). Therefore, the discrepancy between their results could only be due to particles in this size range, which typically has only a minor influence on the mass concentrations of the refractory components.*

*We have added the following sentences to **sections 2.1 and 3.2**, respectively:*

*"In brief, aerosols are sampled with a flowrate of ~80 cm$^3$ min$^{-1}$ via an aerodynamic lens, focusing and accelerating particles in a size range between 70 nm and 2500 nm d$_{va}$. Afterwards, they pass through the detection chamber with two diode laser beams (λ = 405 nm). Particles smaller than 200 nm and larger than 2 μm are difficult to detect, due to weak light scattering by the smaller particles and due to a larger particle beam divergence for the larger particles."*

*"After correction of the number counts and estimation of the mass concentrations, we can compare the LAAPTOF result with the quantitative instruments such as AMS in the overlapping size range of 200 to 2500 nm $d_{va}$. A correction for the particles in the size range between 70–200 nm considering mass concentrations may be negligible since they typically contribute only a minor mass fraction."*

*(2) Please refer also to our answers to previous comments. The main approach used in previous SPMS-AMS comparisons is the field-based scaling approach, which relies on the availability of the reference instrument and typically assumes a particle class independent detection efficiency, as well as a class independent relative sensitivity factor (RSF) for different particulate species. Our laboratory-based method does not require a reference instrument during the field measurement and accounts for particle class size dependent detection efficiencies. It is evident that both methods have different advantages and disadvantages. To reflect this we have added the following to the introduction (please refer to our answers to previous comments) and conclusions sections, as follows:*

*Section 4 (2$^{nd}$ paragraph)*

*"...we applied a quantification method for single particles, employing size and particle class/chemically-resolved overall detection efficiencies (ODE) for this instrument. In contrast to methods used in previous SPMS studies, our approach is laboratory-based and doses not rely on the availability of a reference instrument in the field."*

*(3) We have added these references to the introduction and discussion sections.*

*(4) We have included the work of Wentzel et al., 2003 to support our discussions on missing (sulfate) particles in **Section.3.2** (3$^{rd}$ paragraph). Please refer to our answer (R4) to previous comments.*

14. Figures 6-8: (1) Due to matrix effects in LDI, peak areas of a given species will depend on the full matrix of a particle, such that an ammonium peak area for a given quantity would be expected to be different on a dust particle vs an organic carbon particle. As such, it is not advised to compare peak areas across all particle types together. Please see Hatch et al. (2014, Aerosol Sci. Technol.). Also see Healy et al. (2013, Atmos. Chem. Phys.) for suggestions of how to compare SPMS and AMS data, as they found agreement when the data are handled properly. (2) Bhave et al. (2002, Environ. Sci. Technol.) may also be useful, as they compared ammonium and nitrate SPMS data to bulk filter data. For organics in particular, the authors may consider Spencer & Prather 2006 (Aerosol. Sci. Technol., "Using ATOFMS to determine OC/EC mass fractions in particles") and Ferge et *al. (2006, Environ. Sci. Technol.).*

*R14 (1): Indeed, the matrix effects have a strong impact on the peak areas and it is not advised to compare peak areas across all particle types. To correct for the LDI matrix effect, Hatch et al., 2014, used m/z 36 $C_3^+$ as a pseudo-internal standard to normalize the secondary inorganic and organic peak areas in organic rich particles, resulting in good correlation with the independent AMS measurements. Hatch et al. (2014) suggested another method named normalized or relative peak areas (RPAs) to account for LDI artifacts, such as shot-to-shot variability of laser intensities. However, matrix effects cannot be completely overcome by pseudo-internal standards or a RPAs method.*

*Actually, instead of absolute peak area/ion intensities, we used relative ion intensities to correlate LAAPTOF and AMS data (each ion peak intensity is normalized to the sum of all or selected ion signals. Positive and negative ions were analysed separately). Variations of these correlations are caused by varying particle classes. This is actually similar to the method used by Hatch et al. (2014).*

*We have shown that specific ratios such as org/(org+nitrate) are useful to determine the relationships of LAAPTOF ion intensity and AMS mass concentration (cf. Fig. 6 and Fig. S4; which are modified and changed to new Fig. 7 and Fig. S7), which will be applied for source apportionment in an upcoming publication, and to estimate mass concentrations in future SPMS studies.*

*Actually, partially employing the LDI matrix effects the time series of relative intensities of maker peaks (Fig. 7 and 8; new Fig. 8 and 9) allow at least for preliminary assignments of the bulk species from AMS to different particle types. This should be useful in further source appointment.*

*(2) For comparison of ammonium, nitrate, and organic signals, we have cited Bhave et al. (2002), Spencer & Prather 2006, and Ferge et al. (2006) in the introduction.*

*The corresponding revisions are in **Section 1**. Introduction (5$^{th}$ paragraph): please refer to our answers (R2) to previous comments; **Section 3.3** (1$^{st}$ paragraph): please refer to our answers (R1); and in **Section 3.4** (2$^{nd}$ paragraph), as follows:*

[revised manuscript text omitted]

Many previous studies have also compared single particle classes and bulk species (Dall'Osto et al., 2012; Dall'Osto and Harrison, 2012; Dall'Osto et al., 2009; Dall'Osto et al., 2013; Decesari et al., 2014; Decesari et al., 2011; Drewnick et al., 2008; Gunsch et al., 2018; Pratt et al., 2010; Pratt et al., 2011; Pratt and Prather, 2012). Some studies compared ion intensities from single particle data (Bhave et al., 2002) or specific ion ratios, such as nitrate/sulfate (Middlebrook et al., 2003), OC/EC (Spencer and Prather, 2006), and EC/(EC+OC) (Ferge et al., 2006), carbonaceous/(carbonaceous+sulfate) (ZhouMurphy et al., 20106). Combining AMS, optical particle counter (OPC), and SPMS data, Gemayel et al. (2017) also quantified the fragments aforementioned in size-segregated atmospheric aerosols measured by SPMS in the field. In addition with the other bulk data. Hatch et al. (2014) used m/z 36 $C_3^+$ as a pseudo-internal standard to normalize the secondary inorganic and organic peak areas in organic rich particles, resulting in good correlation with the independent AMS measurements. Similarly, Ahern et al. (2016) used the peak area ratio of organic matter marker at m/z 28 $CO^+$ to EC markers ($C_{2-5}^+$) to account for laser shot-to-shot variability, and demonstrated a linear relationship between normalized organic intensity and secondary organic aerosol (SOA) coating thickness on soot particles. To the best of our knowledge, all the previous SPMS quantification methods focused on some components of the aerosol particles, and most of them are based on comparison with reference instrumentsA 
[revised manuscript text omitted]
 averaged particle densities (~1.6 to 1.9 g cm$^{-3}$) based on the comparison between d$_{va}$ and d$_m$ were applied for total aerosol particle mass quantification (Gemayel et al., 2017; Jeong et al., 2011; Zhou et al., 2016). The density for different types of ambient particles varies, which will be shown in the following text. In order to reduce the uncertainty induced by the assumption of a uniform density, we assigned one density to each particle class and this density was used for the individual particles of each class. As discussed in the following Sect. 3.1, 7 major particle classes have been identified. Class 1, 5, and 6 are dust like particles and class 7 contains more mixed particles which also show good correlation to dust reference particles, thus we assumed the same density for class 1, 5, 6, and 7 as for dust, which is about 2.6 g cm$^{-3}$ (Bergametti and Forêt, 2014). Class 2 particles are like aged soot for which we use a density of 1.8 g cm$^{-3}$ as recommended by (Bond et al., 2013). Class 3 is sodium salts like particles, with relatively more sodium nitrate related markers, thus we assumed the density of sodium nitrate (2.3 g cm$^{-3}$). Class 4 consists of relative fresh particles with strongest correlation to the mixture of ammonium sulfate (1.77 g cm$^{-3}$) and nitrate (1.73 g cm$^{-3}$), thus we assume the density as 1.75 g cm$^{-3}$.~~ In our study, we have determined an average density of 1.5 ± 0.3 g cm$^{-3}$ for all ambient particles, based on a comparison between $d_{va}$ measured by AMS and $d_m$ measured by SMPS. However, the density for different types of ambient particles varies, especially for fresh ones (Qin et al., 2006). Particle densities varied during the campaign (Fig. S2) and the representative mass spectra of different particle classes indicate chemical inhomogeneity. In order to reduce the uncertainty induced by the assumption of a uniform density, we assigned specific effective densities (derived from $d_{va}/d_m$) from literature data to each particle class. A density of 2.2 g cm$^{-3}$ was used for calcium nitrate rich particles (Zelenyuk et al., 2005), 1.25 g cm$^{-3}$ for aged soot rich in ECOC-sulfate (Moffet et al., 2008b; Spencer et al., 2007) , 2.1 g cm$^{-3}$ for sodium salts (Moffet et al., 2008b; Zelenyuk et al., 2005), 1.7 g cm$^{-3}$ for secondary inorganic rich particles (Zelenyuk et al., 2005; Zelenyuk et al., 2008), 2.0 g cm$^{-3}$ for aged biomass burning particles (Moffet et al., 2008b), 2.6 g cm$^{-3}$ for dust like particles (Bergametti and Forêt, 2014; Hill et al., 2016). These densities were used for the individual particles of each class without size dependence. Similar chemically-resolved densities have also been used in some previous studies (Gunsch et al., 2018; May et al., 2018; Qin et al., 2006; Qin et al., 2012).

Furthermore, the single particle identification allows for correcting the particle number counts by using the overall detection efficiency  ODE which depends strongly on particle size and type (Shen et al., 2018). Given the fact that ambient aerosol particles are complex mixtures, it is difficult to obtain a specific ODE for each particle class.  For

simplicity and in order to account for different types of ambient particles, we averaged the ODE determined for ammonium nitrate, sodium chloride, PSL particles, and some other particles, e.g., agricultural soil dust, sea salt, organic acids, as well as secondary organic aerosol particles measured in the lab. The mean ODE with uncertainties as a function of particle size are shown in Fig. 1. Using a mean ODE will obviously lead to some bias. For example, if we apply ODE mean values to all the ambient particles, the number of ammonium nitrate rich particles will be overestimated due to the higher ODE of ammonium nitrate, while the ammonium sulfate rich, sea salt particles, and some organic rich particles will be underestimated. Therefore, we used reference particle ODE values to estimate the size dependent ODE values for the particle classes observed in the field as follows. ODE values for ammonium nitrate and sodium chloride were used to fit ODE curves for secondary inorganic rich and sodium salt like particles, respectively. The mean ODE values from all reference particles was used for the class of aged soot particles since it showed best agreement with the reference soot particles (cf. Fig. 1). The minimum ODE curve from all reference particles was used for all dust like particle classes. The equations for correction and calculation of mass concentration are as follows:

$$counts_{corrected} = 1/\mathrm{ODE_{d_m}} \qquad\qquad\qquad\qquad\qquad \mathrm{ODE}_{size\ and\ chemically-resolved}$$

(4)

$$mass_{corrected} = counts_{corrected} \times m_p \qquad\qquad\qquad\qquad\qquad\qquad\qquad (5)$$

mass concentration = Total mass/(sample flowrate × time)  (6)

where $\mathrm{ODE_{d_m}}$ is the mean ODE that depends on $d_m$; $counts_{corrected}$ and $mass_{corrected}$ are the corrected particle number counts and mass at each time point; the sample flowrate is ~80 cm$^{-3}$ min$^{-1}$. Using Eq. (4) to (6) we can calculate the corrected number  and mass fractions .

The aforementioned assumptions and the related uncertainties in particle mass are summarised as follows: 1) ambient particles are spherical with a shape factor χ=1. However, several ambient particle types are non-spherical with a shape factor χ not equal to 1, e.g., $\chi_{NaCl}$= 1.02−1.26 (Wang et al., 2010) and $\chi_{NH_4NO_3}$= 0.8 (Williams et al., 2013). This can cause uncertainties of 26% and 20% for the particle diameter and 100% and 50% for the particle mass of sodium chloride like and ammonium nitrate like particles, respectively. For soot like particles, the shape caused uncertainty could be even larger, due to their aggregate structures. Such an uncertainty is difficult to reduce, since we do not have particle shape information for individual particles. However, using effective densities may at least partially compensate some of the particle shape related uncertainties. 2) Particles in the same class have the same density, which is likely to vary and lead to an uncertainty hard to estimate. 3) The variability of the ODE values (cf. Fig. 1) depends on particle size and type. It reaches values ranging from ±100% for 200 nm particles to ±170% for 800 nm size particles.

Hence, the overall uncertainty in particle mass according to the assumptions is ~300% with the ODE caused uncertainty being dominant. This is because: 1) ambient particles are more complex than particles generated in the laboratory, e.g., concerning morphology or optical properties. These factors have a strong impact on ODE ~~The aforementioned assumptions and the related uncertainties in particle mass are summarised as follows: 1) ambient particles are spherical. This leads to an uncertainty of ~100% in particle mass; 2) particles in the same class have the same density. This leads to an uncertainty of ~4% in particle mass for ammonium salt particles with assumed density (ρ_as) of 1.75 g cm^-3, which is the averaged value from ammonium sulfate and nitrate with densities of 1.77 g cm^-3 and 1.73 g cm^-3, respectively (Weast, 1987), ~25% for sodium salts particles with ρ_as = 2.36 g cm^-3, averaged value from sodium nitrate, chloride, and sulfate with densities of 2.26, 2.17, and 2.67 g cm^-3, respectively (Weast, 1987), and much bigger uncertainty for soot particles with ρ_as = 1.8 g cm^-3, due to their densities raging from < 1 to ~2 as a results of their aggregate structuresBond3; 3) the ODE is the same for all particles of the same size. This leads to an uncertainty of 500% in particle mass due to the variability of ODE values (Fig. 1). Obviously, this will lead to some bias. For example, if we apply ODE mean values to all the ambient particles, the number of ammonium nitrate rich particles might be overestimated due to the higher ODE of ammonium~~

nitrate, while the ammonium sulfate rich, sea salt particles, and some organic rich such as organic acids rich particles might be underestimated.

Hence, the overall uncertainty in particle mass according to the assumptions is ~540% with the ODE caused uncertainty being dominant. ; 2) instrumental aspects such as alignment and variance in particle-laser interaction lead to uncertainty in ODE. They are included in the uncertainties given in Fig. 1 for which repeated measurements after various alignments were used. The fluctuations of particle-laser interactions can be reduced by using a homogeneous laser desorption and ionization beam (Wenzel and Prather, 2004) or delayed ion extraction. (Li et al., 2018; Vera et al., 2005; Wiley and Mclaren, 1955). Note that we used the same sizing laser and desorption/ionization laser pulse energy (4 mJ) in the field as those used for generating ODE, and aligned the instrument in the field with the similar procedures as we did in the lab. During our field measurements we did calibrations of the LAAPTOF with PSL particles of 400, 500, 700, and 800 nm $d_m$ resulting in ODE values with no significant difference compared to the ODE values determined in the laboratory. This finding reflects the good stability of the LAAPTOF performance in the temperature controlled container. Actually, once the LAAPTOF adjustments were optimized after transport no further adjustments were necessary during the 6 weeks of the campaign. Moreover, it is important to note that the ODE curve applied herein should not be extrapolated to other LAAPTOF or SPMS instruments without a standard check against e.g. PSL particles.

[revised manuscript text omitted]

---

## Referee Report (RR1)

**Understanding atmospheric aerosol particles with improved particle identification and quantification by single particle mass spectrometry**

By Shen et al.

General:

This paper characterizes a new approach to scaling single particle mass spec data to mass concentrations then compares the mass to the AMS. This work is of interest to the atmospheric chemistry and aerosol mass spectrometry community. This paper should be considered for publication after the following revisions:

Major Comments:
1. The concept of the ODE needs to be clarified in the work. Several studies [*Allen et al.*, 2000; *Dall'Osto et al.*, 2006; *Gross et al.*, 2000] have identified the need to account for the transmission efficiency of the nozzle and skimmers through a scaling factor and the ability of the wavelength of the LDI laser to be absorbed by the laser and induce the ionization and detection of individual chemical compounds through a RSF. Does the ODE account for both physical and chemical factors? This was not clear in the manuscript. In particular, the authors address the issue of converting dm to dva; however, the issue of scaling factors are never explicitly addressed.
2. More detail of the laboratory studies performed needs to be included in the paper. More detail is given in the figure caption for figure 1 then in the methods. For instance, how were ODEs from laboratory particles translated to ODEs for ambient particles? How were ODEs averaged or "mixed" across particle types like the biomass burning-soil particles which presumably behave like both dust and organic particles?
3. Matrix effects need to be more explicitly addressed in this work. For instance, the ODE for compounds like NaCl and ammonium sulfate, which are poorly detected by SPMS, can be increased by the inclusion of a compound or substance that is well-detected by SPMS. How did the authors account for matrix effects and the fact that ambient particles have a larger variety of chemical compounds that may show higher ODEs than their laboratory counterparts?
4. I suggest that the authors re-write section 3.1 to more explicitly include how the ambient particles were "fit" or "assigned" to different laboratory mimics and to show how this affected the ODEs in a figure. This would be more useful then showing the spectra of the particle classes.

Specific Comments:
Abstract:
1. The goal of this work, if I understand correctly, is to scale the SPMS data to mass without the need of a reference instrument. If I understand this correctly, then this goal should be more clearly stated.

Introduction:
1. I suggest that the introduction be reorganized to explicitly address the transmission biases associated with the nozzle and skimmers and the need for a scaling factor, then biases associated with the ionization and detection of individual chemical compounds and the need for a RSF, then how comparison of SPMS data to a reference instrument has been used to overcome these limitations.

Methods:
1. The authors should discuss any biases associated with scaling to just an SMPS, which cannot size the supermicron particles detected by the SPMS.
2. Page 5, lines 38-39, in addition to Shen et al, please cite [*Allen et al.*, 2000; *Dall'Osto et al.*, 2006; *Qin et al.*, 2006]
3. Section 2.2, too much emphasis is placed on the shape corrections. It would be much more useful to know how scaling factors and RSFs were addressed with the ODEs.
4. There is only one sentence on the laboratory studies used to create the ODEs applied to the field data. This data is critically important and, as stated in the abstract, is a unique aspect of this study. The laboratory studies need their own section in the paper with details of what was studied, how it was studied, how the ODEs were experimentally determined, if RH affected the ODEs, and detail of how Figure 1 was generated.
5. I am surprised the ODEs look so similar for the different particles types in Figure 1. Do the ODEs also account for RSFs?

Results:
1. While interesting, the particle types discussed in section 3.1 don't seem to be the main focus. I suggest reducing the discussion of the particle types and placing most of this text in the SI. Instead, focus on what the ODEs were for ambient particles compared to the laboratory particles since that is the novel aspect of this work.
2. Figure 2 has a lot of text, I suggest reducing to just a few characteristic ion peaks.
3. More citations are needed in section 3.1 on previous work showing these particle types including, but not limited to [*Ault et al.*, 2010; *Gard et al.*, 1998; *Gaston et al.*, 2011; *Gaston et al.*, 2013; *Pratt et al.*, 2009; *Pratt and Prather*, 2009; *Qin et al.*, 2012; *Silva and Prather*, 2000]
4. The "sodium salts" appear to be sea salts. Why not call them sea salts (both fresh and aged) instead?
5. Page 8, line 2, missing this citation [*Pratt et al.*, 2009]
6. Page 8, line 18, cite [*Pratt and Prather*, 2009] for particle coatings that can mask ion peaks.
7. Page 8, line 36, denote the ion peaks for sodium, zinc, copper, etc
8. Section 3.2 first paragraph seems to imply that the ODEs are scaling factors only meaning they only account for size and not RSFs. Is this accurate?
9. Section 3.2, how were the SPMS particle classes compared to the mass concentrations from the AMS? Several ion peaks for the SPMS were listed for comparison with the AMS, but how were the individual compounds scaled to mass? Were just SPMS ion peaks compared to AMS mass concentrations?
10. Page 11, line 26, what is the "compound donor class"?

References:

Allen, J. O., D. P. Fergenson, E. E. Gard, L. S. Hughes, B. D. Morrical, M. J. Kleeman, D. S. Gross, M. E. Galli, K. A. Prather, and G. R. Cass (2000), Particle detection efficiencies of aerosol time of flight mass spectrometers under ambient sampling conditions, *Environmental Science & Technology*, *34*(1), 211-217.

Ault, A. P., C. J. Gaston, Y. Wang, G. Dominguez, M. H. Thiemens, and K. A. Prather (2010), Characterization of the single particle mixing state of individual ship plume events measured at the Port of Los Angeles, *Environmental Science & Technology*, *44*(6), 1954-1961.

Dall'Osto, M., R. M. Harrison, D. C. S. Beddows, E. J. Freney, M. R. Heal, and R. J. Donovan (2006), Single-particle detection efficiencies of aerosol time-of-flight mass spectrometry during the North Atlantic marine boundary layer experiment, *Environmental Science & Technology*, *40*(16), 5029-5035.

Gard, E. E., M. J. Kleeman, D. S. Gross, L. S. Hughes, J. O. Allen, B. D. Morrical, D. P. Fergenson, T. Dienes, M. E. Galli, R. J. Johnson, G. R. Cass, and K. A. Prather (1998), Direct observation of heterogeneous chemistry in the atmosphere, *Science*, *279*(5354), 1184-1187.

Gaston, C. J., H. Furutani, S. A. Guazzotti, K. R. Coffee, T. S. Bates, P. K. Quinn, L. I. Aluwihare, B. G. Mitchell, and K. A. Prather (2011), Unique ocean-derived particles serve as a proxy for changes in ocean chemistry, *Journal of Geophysical Research-Atmospheres*, *116*, D18310, doi:18310.11029/12010JD015289.

Gaston, C. J., P. K. Quinn, T. S. Bates, J. B. Gilman, D. M. Bon, W. C. Kuster, and K. A. Prather (2013), The impact of shipping, agricultural, and urban emissions on single particle chemistry observed aboard the R/V Atlantis during CalNex, *Journal of Geophysical Research-Atmospheres*, *118*, doi:10.1002/jgrd.50427.

Gross, D. S., M. E. Galli, P. J. Silva, and K. A. Prather (2000), Relative sensitivity factors for alkali metal and ammonium cations in single particle aerosol time-of-flight mass spectra, *Analytical Chemistry*, *72*(2), 416-422.

Pratt, K. A., L. E. Hatch, and K. A. Prather (2009), Seasonal volatility dependence of ambient particle phase amines, *Environmental Science & Technology*, *43*(14), 5276-5281.

Pratt, K. A., and K. A. Prather (2009), Real-time, single-particle volatility, size, and chemical composition measurements of aged urban aerosols, *Environmental Science & Technology*, *43*(21), 8276-8282.

Qin, X., K. A. Pratt, L. G. Shields, S. M. Toner, and K. A. Prather (2012), Seasonal comparisons of single-particle chemical mixing state in Riverside, CA, *Atmospheric Environment*, *59*, 587-596.

Qin, X. Y., P. V. Bhave, and K. A. Prather (2006), Comparison of two methods for obtaining quantitative mass concentrations from aerosol time-of-flight mass spectrometry measurements, *Analytical Chemistry*, *78*(17), 6169-6178.

Silva, P. J., and K. A. Prather (2000), Interpretation of mass spectra from organic compounds in aerosol time-of-flight mass spectrometry, *Analytical Chemistry*, *72*(15), 3553-3562.

---

## Author Response (AR2)

*Author's response to the 2nd round reviews:*

We gratefully thank the reviewers for their helpful comments to improve the quality of our manuscript. Reviewers' comments are in black. Our point-to-point replies marked by "R" are in blue. Changes to manuscript (version 2) text are in green.

**Referee #3 comments:**

5 **General remarks:**

1. Section 2.2 describes how the spectra were grouped using fuzzy clustering, creating a soft assignment of each spectrum to one or more cluster centers, and then assigned to each cluster center using correlation coefficients. Unless I missed something, this seems to be a way of arriving at a hard assignment for each spectrum to one of the classes. Subsequent data analysis seems to be based on this hard assignment. Have the authors checked whether hard clustering ("fuzzyfier" = 1) would arrive at the

10 same number fractions and cluster centers? It would be nice if the physical (or mathematical) meaning of the similarity fraction and the value extracted from fuzzy (rather than hard) clustering specifically for this study were explained more clearly.

R: The smallest possible fuzzyfier number in the LAAPTOF software fuzzy c-means algorithm is 1.01. Although it closes to 1, we are still not able to obtain the information of individual particles like by a fuzzy k-means approach (fuzzyfier=1 each spectrum to one class). Therefore, we have developed a method to do a "hard assignment" based on representative spectra

15 obtained by the fuzzy-c means analysis. This is described explicitly in the 2nd paragraph of section 2.2. We have added there the following explanation for the mathematical meaning of the similarity fraction and the value extracted from fuzzy c-means clustering specially for this study.

"The similarity metric is Euclidian distance between the spectral data vectors and a cluster centre (Hinz et al., 1999; Reitz et al., 2016). In our study, fuzzy clustering derived fraction for each particle class is the degree of similarity between aerosol

20 particles in one particular class, rather than a number percentage."

2. What were the criteria by which the measurement period was split into those periods of different lengths? Period 4 seems a bit arbitrarily defined (having no other data to go by than the mass spectrometer results shown in the figures). Were those periods meteorologically different from each other? Also, were the data of the instruments on pg. 27, lines 25 – 29, used at all in this study?

25 R: We used two criteria to determine the different time periods: 1) a period should have a stable correlation between LAAPTOF and AMS total mass; 2) a period should contain special events or dominating particle classes observed by LAAPTOF and/or AMS (except for the residue period 6), i.e.,

P1-"Biomass burning- organosulfate" like particles burst measured by LAAPTOF

P2-Secondary inorganic rich measured by LAAPTOF and AMS

30 P3-Sulfate and organic rich measured by AMS

P4-Secondary inorganic rich measured by LAAPTOF and AMS

P5-Organic rich measured by AMS

Partially these periods are related to certain meteorological conditions but these were not selection criteria. We have clarified the criteria in the revised manuscript in the 2nd paragraph of section 3.2, as follows:

35 "Two criteria were used to select characteristic time periods: a period should have a stable correlation between LAAPTOF and AMS total mass; and a period should contain special events or dominating particle classes observed by LAAPTOF and/or AMS (cf. Fig 4 c and Fig. 6)."

Although we did not use data of all the instruments listed on pg. 27, lines 25 – 29, we think it is worth mentioned them all since we used e.g. the meteorological data and $PM_{2.5}$ from the FIDAS OPC and will use these data in also in an upcoming publication. Therefore, and since it is not consuming to much space we decided to keep the instrument description as it is.

**Minor edits:**

1. pg. 28, line 12: more accurate wording would be: the single particle is detected successively (not "coincidently") by the two detection lasers, and its flight time (not diameter) is recorded and then converted to size. Also, the mass spectrometer itself, I believe, has better than unit mass resolution. The total system may be limited to unit mass resolution, so this should be worded accurately.

R: To take this into account we have revised these sentences in the last paragraph of section 2.1 to: "Once a single particle is coincidently detected successively by both of the detection lasers, its aerodynamic size is determined and recorded based on its flight time of flight, and an excimer laser pulse ($\lambda$ = 193 nm) is fired for a one step desorption/ionization of the refractory and non-refractory species of the particle. The resulting cations and anions are analysed by a bipolar time-of-flight mass spectrometer resulting it mass spectra with unit mass resolution."

2. pg. 28, line 18: perhaps a short outline of the procedure would be good here.

R: We have added the outline in the 1st paragraph of section 2.2, as follows:

"In brief, spectral data is classified by a fuzzy $c$-means clustering algorithm embedded in the LAAPTOF Data Analysis Igor software (Version 1.0.2, AeroMegt GmbH) to find the major particle classes. Afterwards, we can obtain particle class resolved size ($d_{va}$) distribution and the representative spectra, which will be correlated with laboratory-based reference spectra. The resulting correlations together with marker peaks (characteristic peaks arising from the corresponding species and some typical peak ratios (e.g., isotopic ratio of potassium) are used to identify the particle classes. Here, we extend this approach to quantify particle class mass contributions using a large ambient sample as test case."

3. pg. 35, line 33: Was this found for the LAAPTOF or a different SPMS?

R: This was found for a different SPMS, PALMS (Particle Analysis by Laser Mass Spectrometry). We have clarified this in the 1st paragraph of section 3.3, as follows:

"This was also found by Murphy et al. (2006) for another single particle mass spectrometer, PALMS, which also uses an excimer laser with the same wavelength for ionization as that in the LAAPTOF."

**Presentation/Figures:**

Figure 1: This is an enormous caption. Perhaps some of the information could be put into a table or into the supplement (or both)?

R: This is a quite important figure for this manuscript as it is the basis for our mass quantification with LAAPTOF. Therefore, we consider it justified to keep this figure and the detailed caption together in the main manuscript.

Figure 2: The labels in the spectra are extremely hard to read, besides being a bit reader-unfriendly with its color-coded numbers that one has to read up about in the caption. The peak numbering is also a bit excessive. I'd suggest including a table that shows the signature peaks (m/z and name, if known) for each class, as well as those peaks present in each class, and reducing the number of labels (especially for unidentified peaks) in the graphs to a manageable number.

R: As suggested, we have reduced the number of labels. All the background fragments are only kept in the overall averaged spectrum. Furthermore, we have added the characteristic signature peaks for each particle class in Table 1. The revised figure and table are shown as below:

[Figure]

**Table 1: Particle class numbers, names, labels and corresponding signature ion peaks.**

| Class number: name (label) | Signature ion peaks (cations and anions are marked in red and blue, respectively) |
|---|---|
| **Class 1:** Calcium rich and soil dust like particles (Calcium-Soil) | 23 $Na^+$, 40 $Ca^+$, 56 $CaO/Fe^+$, 57 $CaOH^+$, 64/66 $Zn^+$, 65 $Cu^+$, 75 $CaCl^+$, 96 $Ca_2O^+$, 112 $(CaO)_2^+$, 138 $Ba^+$, 154 $BaO^+$, 206-208 $Pb^+$ |
| **Class 2:** Aged soot like particles (Aged soot) | 12n $Cn^+$, 206-208 $Pb^+$ 12n $Cn^-$; sulfate (32 $S^-$, 64 $SO_2^-$, 80 $SO_3^-$, 81 $HSO_3^-$, 97 $HSO_4^-$, 177 $SO_3HSO_4^-$, 195 $HSO_4H_2SO_4^-$) |
| **Class 3:** Sodium salts like particles (Sodium salts) | 23 $Na^+$, 39 $NaO/K^+$, 40 $Ca^+$, 46 $Na_2^+$, 62 $Na_2O^+$, 63 $Na_2OH^+$, 81/83 $Na_2Cl^+$, 92 $Na_2NO_2^+$, 108 $Na_2NO_3^+$, 129 $C_5H_7NO_3^+$, 141 $Na_3Cl_2^+$, 149 $C_4H_7O_2NO_3^+$, 165 $Na_3SO_4^+$, 181 $C_4H_7O_4NO_3^+$, 206-208 $Pb^+$ 35/37 $Cl^-$, 93/95 $NaCl_2^-$, 111 $NaCl_2H_2O^-$, 115 $Na(NO_2)_2^-$, 119 $NaSO_4/AlSiO_4^-$, 120 $NaClNO_3^-$, 131 $NaNO_2NO_3^-$, 147 $Na(NO_3)_2^-$, 151/153 $Na_2Cl_3^-$, 177 $NaClNaSO_4^-/SO_3HSO_4^-$ |
| **Class 4:** Secondary inorganics rich and amine containing particles (Secondary inorganics-Amine) | ammonium and amine (18 $NH_4^+$, 27 $C_2H_3/CHN^+$, 28 $CO/CH_2N^+$, 30 $NO^+$, 43 $C_3H_7/C_2H_3O/CHNO^+$, 58 $C_2H_5NHCH_2^+$, Amine 59 $(CH_3)_3N^+$, 86 $(C_2H_5)_2NCH_2^+$, 88 $(C_2H_5)_2NO/C_3H_6NO_2^+$, 118 $(C_2H_5)_2NCH_2^+$) nitrate (46 $NO_2^-$, 62 $NO_3^-$); sulfate |
| **Class 5:** Aged biomass burning and soil dust like particles (Biomass burning-Soil) | 39 $K/C_3H_3^+$, 41 $K/C_3H_5^+$, 43 $C_3H_7/C_2H_3O^+$, 50 $C_4H_2^+$, 53 $C_4H_5^+$, 55 $C_4H_4/C_3H_3O^+$, 63 $C_5H_3^+$, 77 $C_6H_5^+$, 85 $C_7H^+$, 91 $C_7H_7^+$, 95 $C_7H_{11}^+$, 104 $C_8H_8^+$, 115 $C_9H_7^+$, 138 $Ba^+$, 154 $BaO^+$, 175 $K_2HSO_4^+$, 206-208 $Pb^+$, 213 $K_3SO_4^+$ sulfate |
| **Class 6:** Aged biomass burning and organosulfate containing particles (Biomass burning-Organosulfate) | positive signature peaks feature biomass-burning very similar as given for class 5 organosulfate (141 $C_2H_5O_8O_4^-$, 155 $C_2H_3O_2SO_4^-$, 215 $C_5H_{11}O_3SO_4^-$) |
| **Class 7:** Mixed/aged and dust like particles (Mixed/aged-Dust) | contains almost all the signature peaks from the other classes |

Note that "rich" used in the names stands for the strong spectral signal rather than the real mass fraction.

Figure 4: It might be useful to mark the periods from Figure 5 in graph 3 c).

R: We have marked the periods in Figure 4 (c) as shown below:

[Figure]

5    Figure 5: It would be nice to repeat the class labels on the y-axis.

R: We have done this for Fig 5 as shown below:

[Figure]

**Referee #4 comments:**

**General:** This paper characterizes a new approach to scaling single particle mass spec data to mass concentrations then compares the mass to the AMS. This work is of interest to the atmospheric chemistry and aerosol mass spectrometry community. This paper should be considered for publication after the following revisions:

5   **Major Comments:**

1. The concept of the ODE needs to be clarified in the work. Several studies [Allen et al., 2000; Dall'Osto et al., 2006; Gross et al., 2000] have identified the need to account for the transmission efficiency of the nozzle and skimmers through a scaling factor and the ability of the wavelength of the LDI laser to be absorbed by the laser and induce the ionization and detection of individual chemical compounds through a RSF. Does the ODE account for both physical and chemical factors? This was not

10   clear in the manuscript. In particular, the authors address the issue of converting dm to dva; however, the issue of scaling factors are never explicitly addressed.

R: Thank you for pointing this out. The laboratory-derived ODE used in this study accounts for both physical and chemical factors (e.g. including the transmission of the aerodynamic lens, ADL) and it is defined by the following equations:

ODE=SE $\times$ HR $\times$ 100%

15   SE=$N_d/N_0 \times 100\%$ (transmission efficiency of ADL is included)

HR=$N_s/N_d \times 100\%$ (ionization efficiency is included)

$N_0$= $C_n \times$ flowrate $\times$ time

where $N_d$ is the number of particles detected by light scattering, $N_0$ is the number of particles in front of the ADL, $N_s$ the number of bipolar spectra, $C_n$ is the particle number concentration (cm$^{-3}$) measured by a CPC in front of the ADL and the

20   flowrate is the LAAPTOF sample flowrate. This has been described in detail in our previous publication (Shen et al., 2018).

It should be noted that, regarding the chemical factors, our ODE is related to particle types as used as reference rather than individual chemical compounds. We did not determine relative sensitivity factors (RSF) for individual chemical compounds.

In section 2.2 (4th paragraph) we have added the following explanations:

"In a previous publication, we defined ODE as the number of bipolar mass spectra obtained from the total number of particles

25   in the sampled air, described how to generate the laboratory-derived ODE and discussed the factors influencing ODE in detail (Shen et al., 2018). Our ODE accounts for both physical and chemical factors (e.g., particle size and types shown in Fig. 1). However, we did not determine relative sensitivity factors for individual chemical compounds."

2. More detail of the laboratory studies performed needs to be included in the paper. More detail is given in the figure caption for figure 1 than in the methods. For instance, how were ODEs from laboratory particles translated to ODEs for ambient

30   particles? How were ODEs averaged or "mixed" across particle types like the biomass burning-soil particles which presumably behave like both dust and organic particles?

R: 1) The laboratory approach to generate ODEs has been described in detail in our previous publication (Shen et al., 2018). Therefore, we did not repeat all of it in this study. Some points are clarified in our answer to the previous comment. Figure 1 was adapted from Shen et al. (2018), but we added more particle types and curves showing the ODE uncertainty range.

35   Therefore, we extended the figure caption significantly. However, we consider it useful to have this information closely associated with this important figure.

2) We have determined ODEs for several particle types, from particles consisting of pure compounds to the complex mixtures including major ambient particle types, such as dust, sea salt, soot, and SOA. We chose ODEs of reference particle types that are similar as the ambient particle classes to represent ODEs for the ambient particles. Regarding biomass burning-soil particles

and the other dust like particles, we assumed that their detection is dominated by the dust core as it significantly influences the light scattering (size) and the particle beam divergence (shape). As mentioned in section 3.1, previous studies in the same region as our study identified soil dust as the particle type dominating the coarse particles (Faude and Goschnick, 1997; Goschnick et al., 1994). Furthermore, they identified core-shell particle structures (Goschnick et al. (1994).

5   To clarify this issue, we have reformulated the 4$^{th}$ paragraph in section 2.2, as follows:

 "As shown in Fig. 1, we have determined ODEs for several particle types, from particles consisting of pure compounds to the more realistic ones including major ambient particle types (cf. Fig. 1). For simplicity and in order to account for different types of ambient particles, we averaged the ODE determined for ammonium nitrate, sodium chloride, PSL particles, and some other particles, e.g., agricultural soil dust, sea salt, organic acids, as well as secondary organic aerosol particles measured in the lab.

10  The mean ODE with uncertainties as a function of particle size ($d_m$) are shown in Fig. 1. However, using a mean ODE will obviously lead to some bias. For example, if we apply ODE mean values to all the ambient particles, the number of ammonium nitrate rich particles will be overestimated due to the higher ODE of ammonium nitrate, while the ammonium sulfate rich, sea salt particles, and some organic rich particles will be underestimated. Therefore, we used reference particle ODE values to estimate the size dependent ODE values for the particle classes observed in the field as follows. ODE values for ammonium

15  nitrate and sodium chloride were used to fit ODE curves for secondary inorganic rich and sodium salt like particles, respectively. The mean ODE values from all reference particles was used for the class of aged soot particles since it showed best agreement with the reference soot particles (cf. Fig. 1). For the same reason, the minimum ODE curve from all reference particles was used for all dust like particle classes. It should be noted, that dust like particles were often mixed with other species such as organics (e.g. biomass burning-soil particles; cf. section 3.1), and that they likely have dust-core shell structures

20  (Goschnick et al., 1994). We assume that their detection is dominated by the dust core as it significantly influences the light scattering (size) and the particle beam divergence (shape)."

It should be feasible to determine in the future studies ODEs for complex ambient particle types by either generating them in simulation chambers or selecting them during field measurements under controlled conditions (selected size, controlled temperature and RH). We have rephrased the corresponding contents in section 4 conclusions (3$^{rd}$ paragraph), as follows:

25  "Considering reduced quantification uncertainties, systematic measurements on different types of standard samples, as well as real ambient samples (size selected) under controlled environmental conditions (temperature and relative humidity) are still needed to obtain more comprehensive sensitivities for LAAPTOF."

3. Matrix effects need to be more explicitly addressed in this work. For instance, the ODE for compounds like NaCl and ammonium sulfate, which are poorly detected by SPMS, can be increased by the inclusion of a compound or substance that is

30  well-detected by SPMS. How did the authors account for matrix effects and the fact that ambient particles have a larger variety of chemical compounds that may show higher ODEs than their laboratory counterparts?

R: If ammonium sulfate is internally mixed with ammonium nitrate, LAAPTOF can detect both of them, which has been verified in our laboratory and the matrix effect has also been discussed in our previous study (Shen et al., 2018). As mentioned above, we assigned reference particle types with known ODE as surrogate for the ambient particle classes observed by

35  LAAPTOF. Since most of the particle classes consist of mixtures of poorly detectable types with better detectable types this seems to partially compensate for the limitation of LAAPTOF to detected certain particle types as evident by the relative good comparison of LAAPTOF and AMS mass concentrations. If the ODEs for ambient particles would differ very much from our assumptions the comparison e.g. with results from the AMS measurements should be worse. For over 85% of the measurement time LAAPTOF and AMS show good correlations regarding total particle mass concentrations. Nonetheless, we are aware of

40  the weakness of LAAPTOF to detect certain particle types (e.g. ammonium sulfate or organics) and the remaining large uncertainties of our approach, which are stated in section 2.2.

To clarify this issue, we have added a new paragraph (5th) and reformulated 2nd last paragraph in section 2.2, as follows:

Section 2.2 (5th paragraph):

"The chemically-resolved ODE could also bring some bias due to complex particle matrix. For instance, if ammonium sulfate is internally mixed with ammonium nitrate, LAAPTOF can detect both of them with good efficiency. This has been verified in our laboratory and the matrix effect has been discussed in our previous study (Shen et al., 2018). As shown in Fig. 1, ODEs for ammonium nitrate are at a higher level, while ODEs for sodium chloride are relatively low. This could lead to an underestimation and overestimation of secondary inorganic rich and sodium salts particles, respectively. ODEs from reference particles with low detection efficiency were applied to dust like particles. This may lead to an overestimation of their concentration if they are mixed with better detectable species. Mean ODEs values were applied to soot particles which may lead to an overestimation if they were e.g. coated. This is because even non-absorbing species e.g., organics can refract light towards the absorbing black carbon core, increasing light absorption (Ackerman and Toon, 1981). Since most of the particle classes consist of mixtures of the poorly detectable types with better detectable types this seems to partially compensate for the limitation of LAAPTOF to detected certain particle types as evident by comparison with the AMS mass concentrations (cf. section 3.2)."

Section 2.2 (2nd last paragraph):

"…the aforementioned particle matrix effects may cause higher or lower ODEs than their surrogates generated in the laboratory. In addition, the more complex morphology and various optical properties of ambient particles can have a strong impact on their ODE (Shen et al., 2018). ….In order to evaluate our quantification approach, we will compare the particle mass estimated based on single particle measurements with AMS total mass in section 3.2"

4. I suggest that the authors re-write section 3.1 to more explicitly include how the ambient particles were "fit" or "assigned" to different laboratory mimics and to show how this affected the ODEs in a figure. This would be more useful then showing the spectra of the particle classes.

R: We consider it important to show the mass spectra related to the different particle types identified. This is the basis to assign ODE values of different laboratory mimics to them. This assignment is now explained in more detail in section 2.2 (see previous comment). Therefore, it is not necessary to repeat this in section 3.1. Furthermore, we have modified the first sentence in section 3.2 as follows:

"In this section, we estimate mass concentrations of the particle classes observed in the field. This is based on the particle identification discussed above as well as the assignment of appropriate ODE values of surrogate reference particles and on several assumptions on particle density and shape (cf. Sect. 2.2)."

**Specific Comments:**

**Abstract:**

1. The goal of this work, if I understand correctly, is to scale the SPMS data to mass without the need of a reference instrument. If I understand this correctly, then this goal should be more clearly stated.

R: We have updated the abstract with more emphasis on this goal. The revised sentence is as follows:

"With the precise particle identification and well characterized laboratory-derived overall detection efficiency (ODE) for this instrument, particle similarity can be transferred into corrected number and mass fractions without a need of a reference instrument in the field."

**Introduction:**

1. I suggest that the introduction be reorganized to explicitly address the transmission biases associated with the nozzle and skimmers and the need for a scaling factor, then biases associated with the ionization and detection of individual chemical compounds and the need for a RSF, then how comparison of SPMS data to a reference instrument has been used to overcome these limitations.

R: We consider it not necessary to discuss the potential biases associated with nozzles, skimmers, ionization, and detection more explicitly as in the current version of the introduction since we use laboratory generated reference particles to determine the overall detection efficiency, which is including all the individual processes. Part of this discussion is also included in our previous publications (Shen et al., 2018; Ramisetty et al., 2018). Therefore, we would like to keep the current introduction structure.

**Methods:**

1. The authors should discuss any biases associated with scaling to just an SMPS, which cannot size the supermicron particles detected by the SPMS.

R: In our laboratory aerosol particles are sized by SMPS, by optical particle counters (e.g. WELAS, Palas), and by aerodynamic particle counters (APS, TSI) covering particle sizes between 4 nm and 100 µm. For reference particles measured from our AIDA simulation chamber typically, a combination of particle sizers was used to determine the size distributions.

As shown Figure 1, we determined ODE values for mobility equivalent particle sizes ($d_m$) ranging from 300 nm to 1 µm (submicron $d_m$). Above 1 µm we measured only 1.2 µm silica as well as 1.6 and 2 µm PSL particles. The ODE decreases significantly for larger particles, because of increasing particle beam divergence. We assume ODEs for supermicron particles to follow the decreasing trend illustrated in Fig. 1.

Please note that LAAPTOF cannot measure particles larger than a vacuum aerodynamic diameter ($d_{va}$) of 2.5 µm, which corresponds to a mobility equivalent diameter ($d_m$) of 1.0 to 1.5 µm assuming effective particle densities of 1.7 to 2.6 g cm$^{-3}$ for different ambient particle classes, respectively. Hence, a large fraction of the ambient particles measured by LAAPTOF could be number-corrected by using our laboratory-derived ODEs.

We have clarified this in 6$^{th}$ paragraph of section 2.2, as follows:

"As shown Fig. 1, we determined ODE values for mobility equivalent particle sizes ($d_m$) ranging from 300 nm to 1 µm. The ODE decreases significantly for larger particles, because of increasing particle beam divergence. We assume ODEs for supermicron particles to follow the decreasing trend illustrated in Fig. 1. Please note that LAAPTOF cannot measure particles larger than a $d_{va}$ of 2.5 µm, which corresponds to a $d_m$ of 1.0 to 1.5 µm assuming effective particle densities of 1.7 to 2.6 g cm$^{-3}$ for different ambient particle classes, respectively. Hence, a large fraction of the ambient particles measured by LAAPTOF could be number-corrected by using our laboratory-derived ODEs."

We have also added a few sentences in the 1$^{st}$ paragraph of section 3.2, as follows:

"Please note that both AMS and LAAPTOF cannot measure particles larger than 2.5 µm, which can be analysed by FIDAS. FIDAS data showed that PM$_{2.5}$ accounted for majority mass of the total aerosol particles sampled through TSP inlet (PM$_{2.5}$ =73% of PM$_{10}$ and 64% of PM$_{total}$, respectively). In this study, we only focus on PM$_{2.5}$ particles."

2. Page 5, lines 38-39, in addition to Shen et al, please cite [Allen et al., 2000; Dall'Osto et al., 2006; Qin et al., 2006]

R: We have cited these references in the revised manuscript.

3. Section 2.2, too much emphasis is placed on the shape corrections. It would be much more useful to know how scaling factors and RSFs were addressed with the ODEs.

R: The approach to obtain laboratory-derived ODE has been described in detail in our previous work (Shen et al., 2018). Therefore, in this study we focused on corrections by using updated ODEs with new particle types in the laboratory and well identified particle classes in the field. Such corrections are not straightforward and thus deserve sufficient explanation.

4. There is only one sentence on the laboratory studies used to create the ODEs applied to the field data. This data is critically important and, as stated in the abstract, is a unique aspect of this study. The laboratory studies need their own section in the paper with details of what was studied, how it was studied, how the ODEs were experimentally determined, if RH affected the ODEs, and detail of how Figure 1 was generated.

R: Please refer to our answers to the Major Comments #1 and #2 as well as the extended section 2.2.

5. I am surprised the ODEs look so similar for the different particles types in Figure 1. Do the ODEs also account for RSFs?

R: The ODE values shown in Figure 1 may look similar because of the logarithmic scale. The ODEs account for particle size and type, but not RSFs regarding specific chemical compounds. We have clarified this in the revised manuscript. Please refer to our answers to Major Comments #1.

**Results:**

1. While interesting, the particle types discussed in section 3.1 don't seem to be the main focus. I suggest reducing the discussion of the particle types and placing most of this text in the SI. Instead, focus on what the ODEs were for ambient particles compared to the laboratory particles since that is the novel aspect of this work.

R: Particle identification is one important prerequisite for quantification. Only this allows the suitable assignment of surrogate reference particle types and their ODEs. Therefore, we consider it necessary to keep the current discussion in the manuscript while the other question is addressed in the extended section 2.2 of the revised manuscript. Please refer also to our answer to Major Comment #4.

2. Figure 2 has a lot of text, I suggest reducing to just a few characteristic ion peaks.

R: We agree. We have reduced the number of labels in Fig 2. Please refer also to our answers to Reviewer #2's comment on Presentation/Figures.

3. More citations are needed in section 3.1 on previous work showing these particle types including, but not limited to [Ault et al., 2010; Gard et al., 1998; Gaston et al., 2011; Gaston et al., 2013; Pratt et al., 2009; Pratt and Prather, 2009; Qin et al., 2012; Silva and Prather, 2000]

R: We have cited these in section 3.1 and a few more citations. including (Dall'Osto et al., 2016; Jeong et al., 2011; May et al., 2018; Middlebrook et al., 2003; Roth et al., 2016; Schmidt et al., 2017; Zelenyuk et al., 2017)

4. The "sodium salts" appear to be sea salts. Why not call them sea salts (both fresh and aged) instead?

R: As mentioned in this section, this class contains sub-classes, e.g., sodium chloride and sodium nitrate/sulfate, respectively, which are likely to be fresh and aged sea salts. However, the measurement site is relatively far away from the sea (e.g., North Atlantic Ocean is ~800 km away). Therefore, we need more evidence, such as back trajectory analysis or other transport modelling, to prove that this class is really fresh and/or aged sea salt. Until this model analysis is completed, we would like to use the more general term "sodium salts". We have clarified this in the 2nd paragraph of section 3.1

"This hints to possible sub-classes assignments, which are likely to be fresh and aged sea salts. However, the measurement site is relatively far away from the sea (e.g., North Atlantic Ocean is ~800 km away). Therefore, we need more evidence, such

as back trajectory analysis or other transport modelling, to prove that this class is really fresh and/or aged sea salt. This will be discussed in a separate study."

5. Page 8, line 2, missing this citation [Pratt et al., 2009]

R: We have added this citation in the revised manuscript.

5   6. Page 8, line 18, cite [Pratt and Prather, 2009] for particle coatings that can mask ion peaks.

R: We have cited this reference in the revised manuscript.

7. Page 8, line 36, denote the ion peaks for sodium, zinc, copper, etc

R: We have denoted the ion peaks in the 6th paragraph of section 3.1, as follows:

"…some other metals related signatures including m/z 23 $Na^+$, 64/66 $Zn^+$, 65 $Cu^+$, 138 $Ba^+$, 154 $BaO^+$ and 206–208 $Pb^+$."

10   8. Section 3.2 first paragraph seems to imply that the ODEs are scaling factors only meaning they only account for size and not RSFs. Is this accurate?

R: Our ODEs account for particle size and type, but not RSFs regarding specific chemical compounds. We have clarified this in the revised manuscript especially in section 2.2. Please refer also to our answers to Major Comments #1:

9. Section 3.2, how were the SPMS particle classes compared to the mass concentrations from the AMS? Several ion peaks
15   for the SPMS were listed for comparison with the AMS, but how were the individual compounds scaled to mass? Were just SPMS ion peaks compared to AMS mass concentrations?

R: In section 3.2, we only compare the total particle mass between SPMS and AMS but we did not scale the individual particulate compounds, and did not use SPMS ion peaks for comparison. However, in section 3.3, we used the SPMS relative ion intensities and the special ion ratios compared to AMS mass concentrations, and found specific relationships of LAAPTOF
20   ion intensities ratios and AMS mass concentration especially for the fraction of org/(org+nitrate).

10. Page 11, line 26, what is the "compound donor class"?

R: Refer to terminology in chemistry, such as "proton donor", we used "compound donor" to represent the particle class that contributes to the specific compound. We have rephrased this sentence in the 1st paragraph of section 3.4, as follows:

"In order to find out the dominant particle class/classes contributing to/donating a certain non-refractory compound measured
25   by AMS (namely compound-donor particle class/classes)"

References suggested by the reviewer:

Allen, J. O., D. P. Fergenson, E. E. Gard, L. S. Hughes, B. D. Morrical, M. J. Kleeman, D. S. Gross, M. E. Galli, K. A. Prather, and G. R. Cass (2000), Particle detection efficiencies of aerosol time of flight mass spectrometers under ambient sampling conditions, Environmental Science & Technology, 34(1), 211-217.

30   Ault, A. P., C. J. Gaston, Y. Wang, G. Dominguez, M. H. Thiemens, and K. A. Prather (2010), Characterization of the single particle mixing state of individual ship plume events measured at the Port of Los Angeles, Environmental Science & Technology, 44(6), 1954-1961.

Dall'Osto, M., R. M. Harrison, D. C. S. Beddows, E. J. Freney, M. R. Heal, and R. J. Donovan (2006), Single-particle detection efficiencies of aerosol time-of-flight mass spectrometry during the North Atlantic marine boundary layer experiment,
35   Environmental Science & Technology, 40(16), 5029-5035.

Gard, E. E., M. J. Kleeman, D. S. Gross, L. S. Hughes, J. O. Allen, B. D. Morrical, D. P. Fergenson, T. Dienes, M. E. Galli, R. J. Johnson, G. R. Cass, and K. A. Prather (1998), Direct observation of heterogeneous chemistry in the atmosphere, Science, 279(5354), 1184-1187.

Gaston, C. J., H. Furutani, S. A. Guazzotti, K. R. Coffee, T. S. Bates, P. K. Quinn, L. I. Aluwihare, B. G. Mitchell, and K. A.
40   Prather (2011), Unique ocean-derived particles serve as a proxy for changes in ocean chemistry, Journal of Geophysical Research-Atmospheres, 116, D18310, doi:18310.11029/12010JD015289.

Gaston, C. J., P. K. Quinn, T. S. Bates, J. B. Gilman, D. M. Bon, W. C. Kuster, and K. A. Prather (2013), The impact of shipping, agricultural, and urban emissions on single particle chemistry observed aboard the R/V Atlantis during CalNex, Journal of Geophysical Research-Atmospheres, 118, doi:10.1002/jgrd.50427.

Gross, D. S., M. E. Galli, P. J. Silva, and K. A. Prather (2000), Relative sensitivity factors for alkali metal and ammonium cations in single particle aerosol time-of-flight mass spectra, Analytical Chemistry, 72(2), 416-422.

Pratt, K. A., L. E. Hatch, and K. A. Prather (2009), Seasonal volatility dependence of ambient particle phase amines, Environmental Science & Technology, 43(14), 5276-5281.

Pratt, K. A., and K. A. Prather (2009), Real-time, single-particle volatility, size, and chemical composition measurements of aged urban aerosols, Environmental Science & Technology, 43(21), 8276-8282.

Qin, X., K. A. Pratt, L. G. Shields, S. M. Toner, and K. A. Prather (2012), Seasonal comparisons of single-particle chemical mixing state in Riverside, CA, Atmospheric Environment, 59, 587-596.

Qin, X. Y., P. V. Bhave, and K. A. Prather (2006), Comparison of two methods for obtaining quantitative mass concentrations from aerosol time-of-flight mass spectrometry measurements, Analytical Chemistry, 78(17), 6169-6178.

Silva, P. J., and K. A. Prather (2000), Interpretation of mass spectra from organic compounds in aerosol time-of-flight mass spectrometry, Analytical Chemistry, 72(15), 3553-3562.

Additional references added:

Ackerman, T. P. and Toon, O. B.: Absorption of visible radiation in atmosphere containing mixtures of absorbing and non-absorbing particles, Appl Optics, 20, 3661–3668, 1981.

Dall'Osto, M., Beddows, D. C. S., McGillicuddy, E. J., Esser-Gietl, J. K., Harrison, R. M., and Wenger, J. C.: On the simultaneous deployment of two single-particle mass spectrometers at an urban background and a roadside site during SAPUSS, Atmos Chem Phys, 16, 9693–9710, 2016.

Faude, F. and Goschnick, J.: XPS, SIMS and SNMS applied to a combined analysis of aerosol particles from a region of considerable air pollution in the upper Rhine valley, Fresen J Anal Chem, 358, 67–72, 1997.

Goschnick, J., Schuricht, J., and Ache, H. J.: Depth-structure of airborne microparticles sampled downwind from the city of Karlsruhe in the river Rhine Valley, Fresen J Anal Chem, 350, 426–430, 1994.

Jeong, C. H., McGuire, M. L., Godri, K. J., Slowik, J. G., Rehbein, P. J. G., and Evans, G. J.: Quantification of aerosol chemical composition using continuous single particle measurements, Atmos Chem Phys, 11, 7027–7044, 2011.

May, N. W., Gunsch, M. J., Olson, N. E., Bondy, A. L., Kirpes, R. M., Bertman, S. B., China, S., Laskin, A., Hopke, P. K., Ault, A. P., and Pratt, K. A.: Unexpected contributions of sea spray and lake spray aerosol to inland particulate matter, Environ Sci Tech Let, 5, 405–412, 2018.

Middlebrook, A. M., Murphy, D. M., Lee, S. H., Thomson, D. S., Prather, K. A., Wenzel, R. J., Liu, D. Y., Phares, D. J., Rhoads, K. P., Wexler, A. S., Johnston, M. V., Jimenez, J. L., Jayne, J. T., Worsnop, D. R., Yourshaw, I., Seinfeld, J. H., and Flagan, R. C.: A comparison of particle mass spectrometers during the 1999 Atlanta Supersite Project, J Geophys Res-Atmos, 108, doi:10.1029/2001jd000660, 2003.

Roth, A., Schneider, J., Klimach, T., Mertes, S., van Pinxteren, D., Herrmann, H., and Borrmann, S.: Aerosol properties, source identification, and cloud processing in orographic clouds measured by single particle mass spectrometry on a central European mountain site during HCCT-2010, Atmos Chem Phys, 16, 505–524, 2016.

Schmidt, S., Schneider, J., Klimach, T., Mertes, S., Schenk, L. P., Curtius, J., and Borrmann, S.: Online single particle analysis of ice particle residuals from mountain-top mixed-phase clouds using laboratory derived particle type assignment, Atmos Chem Phys, 17, 575–594, 2017.

Shen, X. L., Ramisetty, R., Mohr, C., Huang, W., Leisner, T., and Saathoff, H.: Laser ablation aerosol particle time-of-flight mass spectrometer (LAAPTOF): performance, reference spectra and classification of atmospheric samples, Atmos Meas Tech, 11, 2325–2343, 2018.

Zelenyuk, A., Wilson, J., Imre, D., Stewart, M., Muntean, G., Storey, J., Prikhodko, V., Lewis, S., Eibl, M., and Parks, J.: Detailed characterization of particulate matter emitted by lean-burn gasoline direct injection engine, Int J Engine Res, 18, 560–572, 2017.

Best regards,

Xiaoli Shen and co-authors

[revised manuscript text omitted]
30    350, 426–430, 1994.

Gross, D. S., Gälli, M. E., Silva, P. J., and Prather, K. A.: Relative sensitivity factors for alkali metal and ammonium cations in single particle aerosol time-of-flight mass spectra, Anal Chem, 72, 416–422, 2000.

Gunsch, M. J., May, N. W., Wen, M., Bottenus, C. L. H., Gardner, D. J., VanReken, T. M., Bertman, S. B., Hopke, P. K., Ault, A. P., and Pratt, K. A.: Ubiquitous influence of wildfire emissions and secondary organic aerosol on summertime
35    atmospheric aerosol in the forested Great Lakes region, Atmospheric Chemistry and Physics, 18, 3701–3715, 2018.

Hagemann, R., Corsmeier, U., Kottmeier, C., Rinke, R., Wieser, A., and Vogel, B.: Spatial variability of particle number concentrations and NOx in the Karlsruhe (Germany) area obtained with the. mobile laboratory 'AERO-TRAM', Atmos Environ, 94, 341–352, 2014.

Hatch, L. E., Creamean, J. M., Ault, A. P., Surratt, J. D., Chan, M. N., Seinfeld, J. H., Edgerton, E. S., Su, Y. X., and Prather,
40    K. A.: Measurements of Isoprene Derived Organosulfatesisoprene-derived organosulfates in Ambient Aerosolsambient aerosols by Aerosol Timeaerosol time-of-Flight Mass Spectrometryflight mass spectrometry-Part 2: Temporal Variabilitytemporal variability and Formation Mechanismsformation 
[revised manuscript text omitted]